# Identification of Nonparametric Dynamic Causal Model and Latent Process for Climate Analysis

## Abstract

The study of learning causal structure with latent variables has advanced the understanding of the world by uncovering causal relationships and latent factors. However, in real-world scenarios, such as those in climate systems, causal relationships are often nonparametric, dynamic, and exist among both observed variables and latent variables. These challenges motivate us to consider a general setting in which causal relations are nonparametric and unrestricted in their occurrence, which is unconventional to current methods. To solve this problem, with the aid of *3-measurement* in temporal structure, we theoretically show that both latent variables and processes can be identified up to minor indeterminacy under mild assumptions. Furthermore, we demonstrate that the observed causal structure is identifiable if there is *generation variability*, roughly speaking, the latent variables induce sufficient variations in generating the noise terms, by the established *functional equivalence*. The primary idea of this framework is to learn causal representations from causally-related observations, and subsequently address this problem as a task of general nonlinear causal discovery. Based on these theoretical insights, we develop an estimation approach simultaneously learning both the observed causal structure, latent representation, and latent Markov network. Experimental results in simulation studies validate the theoretical foundations and demonstrate the effectiveness of the proposed methodology. In the climate data experiments, we show that it offers a powerful and in-depth understanding of the climate system.

## 1 Introduction

In real-world observations, such as video data, temperature distributions, and economic investigations, are often partially observed. Estimating latent variable causal graphs from these observations is particularly challenging, as the latent variables are in general not identifiable or unique due to the possibility of undergoing nontrivial transformations (Hyvärinen & Pajunen, 1999), even when the independent factors of variation are known (Locatello et al., 2019). Several studies have aimed to uncover causally related latent variables in specific cases. For instance, (Silva et al., 2006) identify latent variables in linear-Gaussian models using Tetrad conditions (Spearman, 1928), while the generalized independent noise (GIN) condition (Xie et al., 2020) has been proposed to estimate linear, non-Gaussian latent variable causal graphs. Recent work employs rank constraints to identify hierarchical latent structures (Huang et al., 2022). However, these approaches are constrained by linear relations and require specific types of structural assumptions.

Furthermore, nonlinear Independent Component Analysis (ICA) has established identifiability results using auxiliary variables (Hyvarinen & Morioka, 2016; 2017; Hyvärinen et al., 2023), showing that independent factors can be recovered up to a certain transformation of the underlying latent variables under appropriate assumptions. In contrast, (Zheng et al., 2022; Zheng & Zhang, 2023) develop identifiability results without auxiliary variables, relying instead on a sparse structure in the generating process, while Gresele et al. (2021) impose restrictions on the function classes of generating process. The framework of nonlinear ICA has further been extended to incorporate temporal structures (Hyvarinen & Morioka, 2017; Yao et al., 2022; Lachapelle et al., 2024), utilizing time-lagged dependencies in temporal data to model dynamic causal mechanisms. Consequently,

identifying latent structures has become a prominent research focus (Schölkopf et al., 2021). For instance, (Lippe et al., 2022) propose methods for identifying latent variables in both time-lagged and instantaneous contexts, (Yao et al., 2023) address settings with partially observed variables, (Buchholz et al., 2024) explore interventional data under linear mixing assumptions, and (Zhang et al., 2024) develop a general framework for identifying latent structures using multiple distributions. However, most prior work assumes that the generating processes from sources to observations are deterministic, except for a few studies that consider linear additive noise (Khemakhem et al., 2020; Hälvä et al., 2021; Gassiat et al., 2020)—let alone scenarios where causal relations exist among observations. Moreover, all of the aforementioned work is constrained by the assumption of an invertible and deterministic generating function, which is often considered infeasible in many real-world scenarios.

A typical setting where such methodological assumptions are too restrictive is the climate system (Rolnick et al., 2022; Lucarini et al., 2014). With high-level latent variables dynamically changing and influencing observations (e.g., temperature, humidity) and causal structure (e.g., wind system), this presents a challenging problem that necessitates developing a solution to uncover these complex relationships. To address it, we focus on a temporal causal structure involving dynamic latent variables driving a latent causal process and causally-related observed variables, with all connections represented through general nonlinear functions. The final objective is to identify these factors under minimal indeterminacy. For this challenging task, we address three fundamental questions: *(i)* What unique attributes of latent variables can be recovered from the *causally-related* observed variables with the aid of temporal structures? *(ii)* How to do causal discovery in the presence of latent variables, without relying on conventional restrictions? *(iii)* How can empirically identify both the latent processes and the observed causal structure simultaneously? Begging these questions, our main contributions are mainly three-fold:

1. We theoretically establish the conditions required for achieving the identifiability of latent variables from causally-related observations in 3-measurements model.

2. We establish functional equivalence between a nonlinear Structural Equation Model (SEM) and a nonlinear ICA model, and provide identifiability guarantees of observed causal Directed Acyclic Graph (DAG) in favor of it.

3. Building on these theoretical foundations, we present a comprehensive estimation framework that, to the best of our knowledge, is the first to simultaneously tackle causal representation learning and causal discovery. Extensive experiments on diverse synthetic and real-world datasets validate the effectiveness of our theory and methodology.

## 2  PROBLEM SETUP

**Causality in climate system.**
We begin by motivating our research from the viewpoint of the climate system. As shown in Fig 1, the high-level climate variables—sunshine, CO2 (Stips et al., 2016), ocean currents (Paulson & Simpson, 1981), and rainfall (Chen & Wang, 1995)—are not directly observed. These variables evolve together through a temporal causal process, changing gradually and exhibiting causal relationships across both instantaneous and time-lagged interactions (Runge et al., 2019; Runge, 2020). This dynamic interplay reshapes their influence on temperature, human activities, and wind patterns. Moreover, regional temperatures interact via wind-driven heat transfer, which is influenced by latent variables

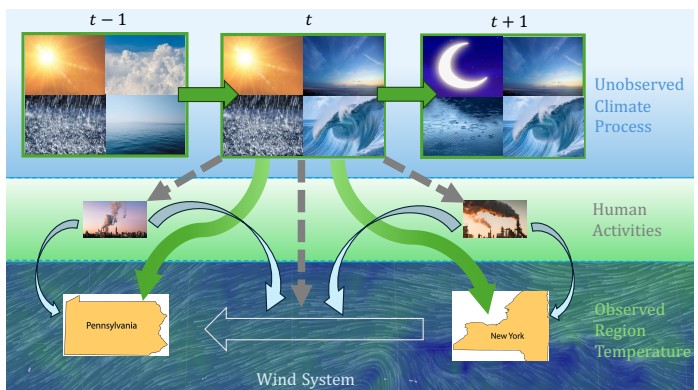

Figure 1: **Systematic analysis of the climate system.** Arrows indicate the causal relationships.

and human activities (Vautard et al., 2019). Given these properties, we know that the climate is a forced and dissipative nonlinear system featuring non-trivial dynamics of a vast range of spatial and temporal scales (Lucarini et al., 2014; Rolnick et al., 2022). Thus traditional causal models fail to represent this. To address this issue, we formally define the 3-measurement model, and describe how observed variables and latent variables are causally-related in data generating process by a structural equation model (SEM) (Spirtes et al., 2001; Pearl, 2009) in a general manner.

**Definition 2.1 (3-Measurement Model)** $\mathbf{Z} = \{\mathbf{z}_{t-1}, \mathbf{z}_t, \mathbf{z}_{t+1}\}$ *represents latent variables in three distinct states, where each state is indexed by its respective time step, we discretize it as* $t \in \mathcal{T} = \{2, \ldots, T-1\}$ *and* $T \geq 3$. *These latent states mutually influence one another. Similarly,* $\mathbf{X} = \{\mathbf{x}_{t-1}, \mathbf{x}_t, \mathbf{x}_{t+1}\}$ *are observed variables that directly measure* $\mathbf{z}_{t-1}, \mathbf{z}_t, \mathbf{z}_{t+1}$ *using the same generating functions g, while* $\mathbf{x}_{t-1}$ *and* $\mathbf{x}_{t+1}$ *provide indirect measurements of* $\mathbf{z}_t$. *Let* $\mathcal{X} \subseteq \mathbb{R}^{d_x}$ *denotes the range of* $\mathbf{x}_t$, *and* $\mathcal{Z} \subseteq \mathbb{R}^{d_z}$ *denote that of* $\mathbf{z}_t$, *where* $d_z \leq d_x$. *The model is defined by the following properties:*

- *The transformation within* $\mathbf{z}_{t-1}, \mathbf{z}_t, \mathbf{z}_{t+1}$ *is not measure-preserving.*

- *Joint density of* $\mathbf{x}_{t-1}, \mathbf{x}_t, \mathbf{x}_{t+1}, \mathbf{z}_t$ *is a product measure w.r.t. the Lebesgue measure on* $\mathcal{X}_{t-1} \times \mathcal{X}_t \times \mathcal{X}_{t+1} \times \mathcal{Z}_t$, *and a dominating measure* $\mu$ *is defined on* $\mathcal{Z}_t$.

- $\mathbf{x}_{t-1}, \mathbf{x}_t$ *and* $\mathbf{x}_{t+1}$ *are conditional indepedent given* $\mathbf{z}_t$.

- *The distribution over* $(\mathbf{X}, \mathbf{Z})$ *is Markov and faithful to a directed acyclic graph (DAG).*

**Explanation.** The example in Fig. 2 defines $\mathbf{x}_{t-1}, \mathbf{x}_t, \mathbf{x}_{t+1}$ as 3 different measurements of $\mathbf{z}_t$ within a temporal structure, since the property of conditional independence given above explicitly specifies that they possess unique information provided from $\mathbf{z}_t$. Conditional independence is standard in causality (Pearl, 2009; Spirtes et al., 2001; Schölkopf et al., 2021) for learning unobservable factors. 3-measurement model can apply beyond the climate data, as detailed in Appendix D.

**Intuition behind 3-measurement.** As shown in Fig. 2, climate time-series data satisfied the 3-measurement model, which supports the enough information for identification of latent variables, analogous to the needs in sufficient number of environments: domain changes in multiple distribution (Hyvärinen et al., 2023; Hyvarinen et al., 2019; Khemakhem et al., 2020; Zhang et al., 2024), sufficient variability in temporal structure (Yao et al., 2021; 2022; Chen et al., 2024), a sufficient number of pure children (Silva et al., 2012; Kong et al., 2023; Ng et al.; Huang et al., 2022). The assumption of 3-measurement implies the *minimum information* required by Hu & Schennach (2008). If more than 3 measurements are available for the same latent variables, the additional measurements, which also carry information about the latent variables, typically enhance their recovery.

**Notation.** We conceptualize the data-generating process through the lens of our perspective on the climate system. It consists of observed variables $\mathbf{x}_t := (x_{t,i})_{i \in \mathcal{I}}$ with index set $\mathcal{I} = \{1, 2, \ldots, d_x\}$ and their causal graph $\mathcal{G}_{x_t}$. Additionally, there are latent variables $\mathbf{z}_t := (z_{t,j})_{j \in \mathcal{J}}$ indexed by $\mathcal{J} = \{1, 2, \ldots, d_z\}$ with the corresponding instantaneous causal graph $\mathcal{G}_{z_t}$. We assume no selection effects, so samples are drawn i.i.d. from the distribution. Let $\mathbf{pa}(\cdot)$ denote the parent variables, $\mathbf{pa}_O(\cdot)$ represent the parent variables in the observed space, and $\mathbf{pa}_L(\cdot)$ indicate the parent variables in the latent space. Notably, $\mathbf{pa}_L(z_{t,j})$ includes latent variables in current time step $t$ and previous time step $t-1$ that are parents of $z_{t,j}$. Specifically, we assume no time-lagged causal relationships in the observed space, and that future states cannot influence the past (Freeman, 1983). However, our methodology can generally be extended to these scenarios; for a discussion, see Appendix G.

**Definition 2.2 (Data Generating Process)** *For $x_{t,i} \in \mathbf{x}_t$ and $z_{t,j} \in \mathbf{z}_t$, the data generating process is defined as follows:*

$$x_{t,i} = g_i(\mathbf{pa}_O(x_{t,i}), \mathbf{pa}_L(x_{t,i}), s_{t,i}); \quad z_{t,j} = f_j(\mathbf{pa}_L(z_{t,j}), \epsilon_{z_{t,j}}); \quad s_{t,i} = g_{s_i}(\mathbf{z}_t, \epsilon_{x_{t,i}}), \quad (1)$$

*where $g_i$ and $f_j$ are differentiable functions, and $\epsilon_{z_{t,j}} \sim p_{\epsilon_{z_j}}$ are independent noise terms. The noise $s_{t,i}$ depending on $\mathbf{z}_t$ is modeled by a nonlinear generation from $\mathbf{z}_t$ and independent noise $\epsilon_{x_{t,i}} \sim p_{\epsilon_{x_i}}$.*

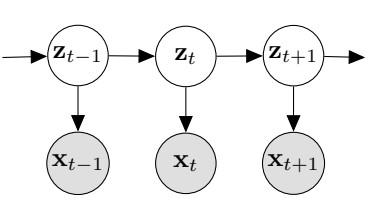

Figure 2: **3-measurement model.** A example of temporal structure with latent Markov process. $\mathbf{x}_t$ is the directed (dominating) measurement of $\mathbf{z}_t$, and $\mathbf{x}_{t-1}$, $\mathbf{x}_{t+1}$ represent indirect measurements.

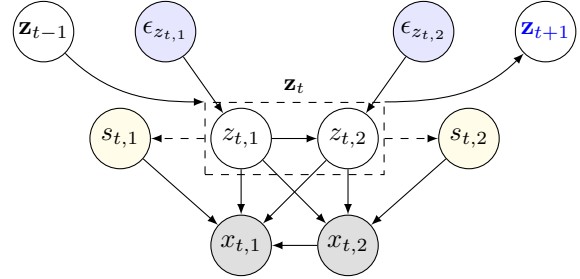

Figure 3: **Data generating process.** Within time-index $t$, there are two types of noise: $\epsilon_{z_{t,i}}$, which denotes independent noise, and $s_{t,i}$, which denotes noise on observations that is dependent on $\mathbf{z}_t$. The dashed rectangle represent $\mathbf{z}_t$, the block of latent variables.

We present a graphical depiction of the data-generating process in Fig. 3. The causal relations within $\mathbf{x}_t$ highlight a fundamental divergence from previous works in causal representation learning (Schölkopf et al., 2021), serving as both a challenge and a point of potential breakthroughs.

Extending our analysis of climate systems, the formulation for generating $x_{t,i}$ in the function space is significantly more general than existing formulations employed in function-based causal discovery methods. These include the post-nonlinear model (Zhang & Hyvarinen, 2012), nonlinear additive models (Ng et al., 2022; Rolland et al., 2022; Hoyer et al., 2008), nonlinear additive models with latent confounders (Zhang et al., 2012; Ng et al.), and models with changing causal mechanisms and nonstationarity (Huang et al., 2020; 2019; Hyvarinen & Morioka, 2016; 2017). Our approach, however, extends beyond these, allowing for different subsets of latent variables and parameters to dynamically control causal edges and generated effects. To illustrate this, we provide a simplified subcase of the defined generating process using a linear additive equation:

$$x_{t,i} = g(\{z_{t,j} \mid z_{t,j} \in \mathbf{pa}_L(x_{t,i})\}) + \sum_{x_{t,j} \in \mathbf{pa}_O(x_{t,i})} b_{i,j}(\mathbf{z}_t, s_{t,i}) \cdot x_{t,j} + s_{t,i}, \tag{2}$$

which captures variations in causal mechanisms driven by latent variables and uncertainties in a nonlinear manner, a scenario often observed in complex climate systems (Rolnick et al., 2022).

## 3 MAIN RESULTS

We present theoretical results demonstrating that the underlying causal variables can be recovered, up to an invertible transformation, from causally-related observations. Specifically, under sparsity constraints on the latent variable graph, we establish the identifiability of latent causal structure up to minor indeterminacies.

### 3.1 PHASE I: IDENTIFYING LATENT VARIABLES FROM CAUSALLY-RELATED OBSERVATIONS

**Definition 3.1** *(Linear Operator) Consider two random variables $a$ and $b$ with support $\mathcal{A}$ and $\mathcal{B}$, the linear operator $L_{b|a}$ is defined as a mapping from a function $f_a$ in some function space $\mathcal{F}(\mathcal{A})$ onto the function $L_{b|a} \circ f_a$ in some function space $\mathcal{F}(\mathcal{B})$,*

$$\mathcal{F}(\mathcal{A}) \to \mathcal{F}(\mathcal{B}): \quad f_b = L_{b|a} \circ f_a = \int_{\mathcal{A}} g_{b|a}(\cdot \mid a) f_a(a) da. \tag{3}$$

We consider it is bounded by the $L^p$-norm, a comprehensive set of all absolutely integrable functions supported on $\mathcal{A}$ (endowed with $\int_{\mathcal{A}} |f(a)| \, d\mu(x) < \infty$, where $\mu$ is a measure on a $\sigma$-field in $\mathcal{A}$), which is sufficiently indicated by an integral operator.

**Theorem 3.2 (Monoblock Identifiability)** *Suppose observed variables and hidden variables follow the data-generating process in Def. 2.2, observations matches the true joint distribution of $\{\mathbf{x}_{t-1}, \mathbf{x}_t, \mathbf{x}_{t+1}\}$, and*

    *(i) The joint distribution of $(\mathbf{X}, \mathbf{Z})$ and their all marginal and conditional densities are bounded and continuous.*

    *(ii) The linear operators $L_{x_{t+1}|z_t}$ and $L_{x_{t-1}|x_{t+1}}$ are injective for bounded function space.*

    *(iii) For all $\mathbf{z}_t, \mathbf{z}'_t \in \mathcal{Z}_t$ $(\mathbf{z}_t \neq \mathbf{z}'_t)$, the set $\{\mathbf{x}_t : p(\mathbf{x}_t|\mathbf{z}_t) \neq p(\mathbf{x}_t|\mathbf{z}'_t)\}$ has positive probability.*

*Suppose that we have learned $(\hat{g}, \hat{f}, p(\hat{\mathbf{z}}_t))$[1] to achieve Eq. 2.2, then we have*

$$\hat{\mathbf{z}}_t = h_z(\mathbf{z}_t) \tag{4}$$

*where $h_z : \mathbb{R}^{d_z} \to \mathbb{R}^{d_z}$ is an invertible function.*

**Proof sketch.** A proof is given in Appendix B.1. The intuition is that, with the information from 3 measurements that can completely recover the latent space, some density functions can be identified by constructing a unique spectral decomposition, up to an indeterminacy in the function space.

**Discussion of assumptions** Assumption (i) is a moderate assumption for ensuring computable distribution supporting the subsequent spectral decomposition. Assumption (ii) enables us to take inverses of certain linear operators. Intuitively, an operator $L_{b|a}$ will be injective if there is enough variation in the density of $b$ for different values of $a$. Special cases of this assumption have been considered in (Newey & Powell, 2003; Mattner, 1993). Under least variation, e.g., $b = a + \epsilon$, it still can be satisfied if Fourier transform of $\epsilon$ is everywhere nonvanishing (Mattner, 1993). In general, it is worth noting that injectivity assumptions are quite weak and commonly made in the literature on nonparametric identification (Hu & Schennach, 2008; Carroll et al., 2010; Hu & Shum, 2012). In our case, this condition could be further relaxed in Cor. 3.9, and interestingly, happen to hold the same view as sufficient variability 3.8 in the following section. Assumption (iii) is much weaker and distinct from monotonicity. For instance, let $\mathcal{M}[p(\mathbf{x}_t \mid \mathbf{z}_t)]$ be monotonic with respect to $\mathbf{z}_t$, where $\mathcal{M} : \mathcal{P}(\mathcal{X}_t) \to \mathbb{R}$ can represent any operator locating a distribution, such as expectation $\mathbb{E}[\cdot]$. Given that $\mathcal{X}_t$ spans an infinite space, the existence of an adequate $\mathcal{M}$ is generally possible.

We show that this result is compatible with existing identifiability results across multiple settings in Appendix D. Monoblock identifiability is sufficient to support the step of causal discovery in Section 3.2. However, without additional constraints, they offer limited insight into the latent causal structure. To address this, we enforce sparsity on the latent Markov network, as proposed in (Zhang et al., 2024; Zheng et al., 2023).

**Theorem 3.3 (Identifiability of Latent Markov Network (See Appendix B.2))**

Once monoblock identifiability *(i)* $\hat{\mathbf{z}}_t = h_z(\mathbf{z}_t)$ is established, leveraging two properties of latent space—namely, *(ii)* the sparsity in the latent Markov network, *(iii)* $z_{t,i} \perp\!\!\!\perp z_{t,j} \mid \mathbf{z}_{t-1}, \mathbf{z}_{t/[i,j]}$ if $z_{t,i}, z_{t,j}$ $(i \neq j)$ are not adjacent—it smoothly links up the advanced identification results on causal representation learning (Zhang et al., 2024; Li et al., 2024) under *sufficient variability* assumption. Due to page limitation, we elaborate the theoretical analysis in Appendix B.2.

## 3.2 Phase II: Identifying Observed Causal DAG in Presence of Latent Variables

In this section, we present an identifiable solution for addressing the problem of general nonlinear causal discovery with latent variables. First, we demonstrate how to transform the SEM into a specific nonlinear ICA at the functional level, with the aim of making this problem analytic tractable.

**Additional notation.** We define the partial Jacobian matrices on $g$ and $g_m$ below. For all $(i, j) \in \mathcal{I} \times \mathcal{I}$:

$$[\mathbf{J}_{g_m}(\mathbf{s}_t)]_{i,j} = \frac{\partial x_{t,i}}{\partial s_{t,j}}, \quad [\mathbf{J}_g(\mathbf{x}_t, \mathbf{s}_t)]_{i,j} = \begin{cases} \frac{\partial x_{t,i}}{\partial x_{t,j}}, & i \neq j \\ \frac{\partial x_{t,i}}{\partial s_{t,j}}, & i = j \end{cases}, \quad [\mathbf{J}_g(\mathbf{x}_t)]_{i,j} = \begin{cases} \frac{\partial x_{t,i}}{\partial x_{t,j}}, & i \neq j \\ 0, & i = j \end{cases}, \tag{5}$$

and $\mathbf{D}_{g_m}(\mathbf{s}_t) = \mathrm{diag}(\frac{\partial x_{t,1}}{\partial s_{t,1}}, \frac{\partial x_{t,2}}{\partial s_{t,2}}, \ldots, \frac{\partial x_{t,d_x}}{\partial s_{t,d_x}})$, $\mathbf{I}_{d_x}$ is the identity matrix in $\mathbb{R}^{d_x \times d_x}$. Specifically, $\mathbf{J}_{g_m}(\mathbf{s}_t)$ represents the derivative of the function in the mixing process (right hand side of 6) from

---

[1] We use the hat symbol to denote the estimated variables and functions.

$\mathbf{s}_t$ to $\mathbf{x}_t$. Note that $\mathbf{J}_g(\mathbf{x}_t)$ implies the causal adjacency in the nonlinear SEM (left hand side of 6), if the assumption below hold true.

**Assumption 3.4 (Functional Faithfulness)** *Causal relations among observed variables are represented by the support set of Jacobian matrix* $\mathbf{J}_g(\mathbf{x}_t)$.

**Explanation.** Functional faithfulness implies *edge minimality* in causal graphs, analogous to the interpretation of structural minimality discussed in Peters et al. (2017) (Remark 6.6) and minimality in Zhang (2013). More discussion about this assumption could be found in the Appendix C.

**Theorem 3.5 (Nonlinear SEM $\iff$ Nonlinear ICA)** *Suppose Assumption 3.4 holds true, then*

a) *(Existence of Equivalent ICA) There exists a function $g_{m_i}$ which is partial differential to $s_{t,i}$ and* $\mathbf{x}_t$, *making*

$$x_{t,i} = g_i(\mathbf{pa}_O(x_{t,i}), \mathbf{pa}_L(x_{t,i}), s_{t,i}) \quad and \quad x_{t,i} = g_{m_i}(\mathbf{an}_L(x_{t,i}), s_{t,i}) \tag{6}$$

*describing the **same data-generating process** as in SEM form (Eq. 1), where $\mathbf{an}_L(\cdot)$ denotes variable's ancestors in $\mathbf{z}_t$.*

b) *(Functional Equivalence) Consider the nonlinear SEM form (left) and nonlinear ICA form (right) described in Eq. 6, the following equation always holds:*

$$\mathbf{J}_g(\mathbf{x}_t)\mathbf{J}_{g_m}(\mathbf{s}_t) = \mathbf{J}_{g_m}(\mathbf{s}_t) - \mathbf{D}_{g_m}(\mathbf{s}_t). \tag{7}$$

**Proof sketch.** See Appendix B.3 for a proof. This proof leverages the properties of DAG to trace the flow of information from the observed variables to the latent variables $\mathbf{z}_t$ (used as side information) and the independent noise term $\epsilon_{x_t}$ in SEM, and ultimately establish a connection to nonlinear ICA.

**Impact of building equivalence.** With assuming causal sufficiency, (Shimizu et al., 2006) connects the causal adjacency matrix $B$ to mixing matrix $(I - B)^{-1}$ of linear ICA, the works by (Monti et al., 2020; Reizinger et al., 2023) attempt to relate nonlinear ICA and SEM in non i.i.d. data, obtain a structural equivalence in the supports of Jacobian matrices. Our approach extends these efforts by addressing cases involving

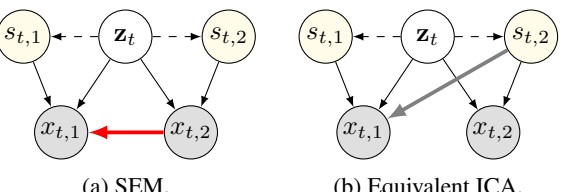

(a) SEM.      (b) Equivalent ICA.

Figure 4: **Equivalent SEM and ICA.** The red line in Fig. 4a indicates that information is transmitted from $x_{t,2}$ in the observed causal DAG, while the gray line in Fig. 4b equivalently represents that $x_{t,2}$ to $x_{t,1}$ because of transitivity.

latent variables, and make such equivalence closer to validity. In Result a), we demonstrate that for an SEM representing a DAG structure, there exists a nonlinear ICA model capable of representing the same data-generating process, an illustrative example is provided in Figure 4b. Subsequently, Result b) outlines the general relationship between SEM and ICA on their parameters, which naturally infers the corollary below.

**Corollary 3.6** *If $\mathcal{G}_{x_t}$ is a DAG, $g_m$ and $g$ are differentiable to $\mathbf{s}_t$ and $\mathbf{x}_t$, respectively, then $\mathbf{J}_{g_m}(\mathbf{s}_t)$, $\mathbf{J}_g(\mathbf{x}_t, \mathbf{s}_t)$ are invertible matrices, and $g_m$ and $g$ are bijective within their respective subspaces given* $\mathbf{z}_t$. *Given results above, observed Causal DAG $\mathcal{G}_{x_t}$ is represented by*

$$\mathbf{J}_g(\mathbf{x}_t) = \mathbf{I}_{d_x} - \mathbf{D}_{g_m}(\mathbf{s}_t)\mathbf{J}_{g_m}^{-1}(\mathbf{s}_t). \tag{8}$$

Through functional equivalence, we unveil that nonlinear DAG indicates bijective functions, eliminating the corresponding assumption in causal discovery under general nonlinear relationships (Monti et al., 2020; Reizinger et al., 2023), thus extending the theory to a broader range of applications. After that, we represent causal adjacency through the Jacobian matrix of nonlinear mixing functions, going beyond linear relations (Shimizu et al., 2006) and structural equivalence (Monti et al., 2020; Reizinger et al., 2023), allowing for interchangeable use in theories and implementations. The transformation on Jacobian matrix preserves the information of parameters,

thereby facilitating theoretical insights in Appendix B.5 and estimation methods in Section 4. Now the objective for learning observed causal DAG comes to clear:

*Identify $\mathbf{s}_t$ to derive the Jacobian matrix $\mathbf{J}_{g_m}(\mathbf{s}_t)$, and subsequently obtain $\mathbf{J}_g(\mathbf{x}_t)$.*

**Remark 3.7** *Theorem 3.5 and its corollaries hold true consistently, regardless of the absence of dependent noise (if $s_{t,i} = \epsilon_{x_{t,i}}$) or latent variables (if $\mathbf{pa}_L(x_{t,i}) = \varnothing$).*

Recalling the generation process of $\mathbf{s}_t$ 2.2, we have $s_{t,i} \perp\!\!\!\perp s_{t,j} \mid \mathbf{z}_t$ ($i \neq j$), suggesting the use of nonlinear ICA with auxiliary variables (Hyvarinen et al., 2019; Khemakhem et al., 2020) to identify $\mathbf{s}_t$ up to trivial indeterminacy.

**Theorem 3.8 (Identifiability of Observed Causal DAG)** *Suppose Assumption 3.4 holds true,*

*[Generation Variability] let $\mathbf{A}_{t,k} = \log p(\mathbf{s}_{t,k}|\mathbf{z}_t)$, assume that $\mathbf{A}_{t,k}$ is twice differentiable in $s_{t,k}$ and is differentiable in $z_{t,l}, l = 1, 2, ..., d_z$. Suppose there exists an estimated $\hat{g}_m$ of the function $g_m$: $\hat{\mathbf{x}}_t = \hat{g}_m(\hat{\mathbf{z}}_t, \hat{\mathbf{s}}_t)$. Let*

$$\mathbf{V}(t,k) := \left(\frac{\partial^2 \mathbf{A}_{t,k}}{\partial s_{t,k}\partial z_{t,1}}, \frac{\partial^2 \mathbf{A}_{t,k}}{\partial s_{t,k}\partial z_{t,2}}, \ldots, \frac{\partial^2 \mathbf{A}_{t,k}}{\partial s_{t,k}\partial z_{t,d_z}}\right), \mathbf{U}(t,k) = \left(\frac{\partial^3 \mathbf{A}_{t,k}}{\partial s_{t,k}\partial^2 z_{t,1}}, \frac{\partial^3 \mathbf{A}_{t,k}}{\partial s_{t,k}\partial^2 z_{t,2}}, \ldots, \frac{\partial^3 \mathbf{A}_{t,k}}{\partial s_{t,k}\partial^2 z_{t,d_z}}\right)^T, \quad (9)$$

*where for $k = 1, 2, \ldots, d_x$, $2d_x$ vector functions $\mathbf{V}(t,1), \ldots \mathbf{V}(t,d_x), \mathbf{U}(t,1), \ldots \mathbf{U}(t,d_x)$ are linearly independent. Then we attain ordered component-wise identifiability (Definition B.9), and thus $supp(\mathbf{J}_g(\mathbf{x}_t)) = supp(\mathbf{J}_{\hat{g}}(\hat{\mathbf{x}}_t))$, meaning the structure of observed causal DAG is identifiable.*

**Proof sketch.** The intuition of the proof involves the derivation of component-wise identifiability, with the same strategy used in nonlinear ICA with auxiliary variables (Hyvärinen & Morioka, 2017), where $\mathbf{z}_t$ obtained by Theorem. 3.2 serves as this side/auxiliary information. Next, to rule out the permutation indeterminacy, we apply Lemma 1 of LiNGAM (Shimizu et al., 2006), which leverages the structural constraints imposed by the DAG structure. Details are presented in B.5.

**Discussion of assumption.** The variable assumption 3.8 is widely used in nonlinear ICA (Hyvärinen et al., 2023; Yao et al., 2022), remaining relatively mild in the presence of heteroskedasticity in $\mathbf{s}_t$ given $\mathbf{z}_t$, as illustrated by (Yao et al., 2022). In practical climate science, it has been demonstrated that, within a given region, human activities ($s_{t,i}$) are strongly impacted by certain high-level climate latent variables ($\mathbf{z}_t$) (Abbass et al., 2022), following a process with sufficient changes, which is distinct from traditional parametric modeling (Lucarini et al., 2014). It is worth noting that the variability assumption made here is compatible with injective operator $L_{s_{t+1}|z_t}$ and $L_{s_{t-1}|s_{t+1}}$, implied by the corollary below.

**Corollary 3.9** *Under DAG constraints on $\mathcal{G}_{x_t}$, for all $t \in \mathcal{T}$, $L_{x_t|s_t}$ is injective.*

Please find the proof and explanation in Appendix B.6.

## 4 ESTIMATION METHODOLOGY

Following our theoretical analysis, we propose a estimation framework for **N**onparametrically doing **C**ausal **D**iscovery and causal representation **L**earning (**NCDL**), as illustrated in Fig. 5.

**Overall architecture.** According to the data generation process 2.2, we establish the Evidence Lower BOund (ELBO) as follows:

$$\mathcal{L}_{ELBO} = \mathbb{E}_{q(\mathbf{s}_{1:T}|\mathbf{x}_{1:T})}\left[\log p(\mathbf{x}_{1:T} \mid \mathbf{s}_{1:T}, \mathbf{z}_{1:T})\right] - \lambda_1 D_{kl}(q(\mathbf{s}_{1:T} \mid \mathbf{x}_{1:T})\|p(\mathbf{s}_{1:T} \mid \mathbf{z}_{1:T})) \\ - \lambda_2 D_{kl}(q(\mathbf{z}_{1:T} \mid \mathbf{x}_{1:T})\|p(\mathbf{z}_{1:T})), \quad (10)$$

Where $\lambda_1$ and $\lambda_2$ are hyperparameters, and $D_{kl}$ represents the Kullback-Leibler divergence. We set $\lambda_1 = 4 \times 10^{-3}$ and $\lambda_2 = 1.0 \times 10^{-2}$ to achieve the best performance. In Fig. 5, the `z-encoder`, `s-encoder` and `decoder` implemented by Multi-Layer Perceptrons (MLPs) are defined as:

$$\mathbf{z}_{1:T} = \phi(\mathbf{x}_{1:T}), \ \mathbf{s}_{1:T} = \eta(\mathbf{x}_{1:T}), \ \hat{\mathbf{x}}_{1:T} = \psi(\mathbf{z}_{1:T}, \mathbf{s}_{1:T}),$$

respectively, where the neural network $\phi$, `z-encoder` learns the latent variables through denoising, and `s-encoder` $\psi$ and `decoder` $\eta$ approximate invertible functions for encoding and reconstruction of nonlinear ICA, respectively.

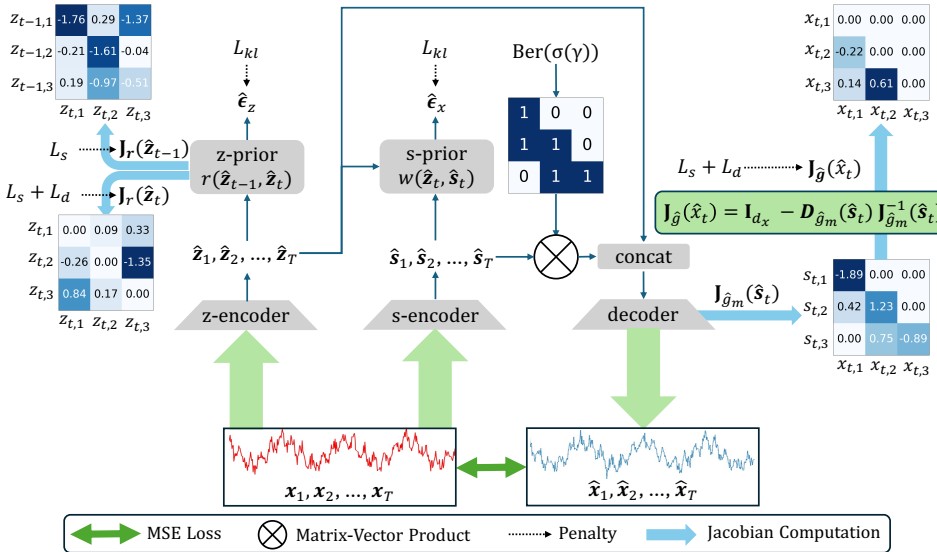

Figure 5: **The estimation procedure of NCDL.** The model framework includes two encoders: `z-encoder` for extracting latent variables $\mathbf{z}_t$, and `s-encoder` for extracting $\mathbf{s}_t$. A `decoder` reconstructs observations from these variables. Additionally, prior networks estimate the prior distribution using normalizing flow, target on learning causal structure based on Jacobian matrix. $L_s$ imposes a sparsity constraint and $L_d$ enforces the DAG structure on Jacobian matrix. $\mathcal{L}_{kl}$ enforces an independence constraint on the estimated noise by minimizing its KL divergence w.r.t. $\mathcal{N}(0, \mathbf{I})$.

**Prior estimation of $\mathbf{z}_t$ and $\mathbf{s}_t$.** We propose using the `s-prior` network and `z-prior` network to recover the independent noise $\hat{\epsilon}_{x_t}$ and $\hat{\epsilon}_{z_t}$, respectively, thereby estimating the prior distribution of latent variables $\hat{\mathbf{z}}_t$ and dependent noise $\hat{\mathbf{s}}_t$. Specifically, we first let $r_i$ be the $i$-th learned inverse transition function that take the estimated latent variables as input to recover the noise term, e.g., $\hat{\epsilon}_{z_{t,i}} = r_i(\hat{\mathbf{z}}_{t-1}, \hat{\mathbf{z}}_t)$. Each $r_i$ is implemented by MLPs. Sequentially, we devise a transformation $\kappa := \{\hat{\mathbf{z}}_{t-1}, \hat{\mathbf{z}}_t\} \rightarrow \{\hat{\mathbf{z}}_{t-1}, \hat{\epsilon}_{z_t}\}$, whose Jacobian can be formalized as $\mathbf{J}_\kappa = \begin{pmatrix} \mathbf{I} & 0 \\ \mathbf{J}_r(\hat{\mathbf{z}}_{t-1}) & \mathbf{J}_r(\hat{\mathbf{z}}_t) \end{pmatrix}$.
Then we have Eq. 11 derived from normalizing flow (Rezende & Mohamed, 2015).

$$\log p(\hat{\mathbf{z}}_t, \hat{\mathbf{z}}_{t-1}) = \log p(\hat{\mathbf{z}}_{t-1}, \hat{\epsilon}_{z_t}) + \log |\frac{\partial r_i}{\partial \hat{z}_{t,i}}|. \tag{11}$$

According to the generation process, the noise $\epsilon_{z_{t,i}}$ is independent of $\mathbf{z}_{t-1}$, allowing us to enforce independence on the estimated noise term $\hat{\epsilon}_{z_{t,i}}$ with $\mathcal{L}_{kl}$. Consequently, Eq. 11 can be rewritten as:

$$\log p(\hat{\mathbf{z}}_{1:T}) = p(\hat{\mathbf{z}}_1) \prod_{\tau=2}^{T} \left( \sum_{i=1}^{d_z} \log p(\hat{\epsilon}_{z_{\tau,i}}) + \sum_{i=1}^{d_z} \log |\frac{\partial r_i}{\partial \hat{z}_{\tau,i}}| \right), \tag{12}$$

where $p(\hat{\epsilon}_{z_{\tau,i}})$ is assumed to follow a Gaussian distribution. Similarly, we estimate the prior of $\mathbf{s}_t$ using $\hat{\epsilon}_{x_{t,i}} = w_i(\hat{\mathbf{z}}_t, \hat{\mathbf{s}}_t)$, and model the transformation between $\hat{\mathbf{s}}_t$ and $\hat{\mathbf{z}}_t$ as follows:

$$\log p(\hat{\mathbf{s}}_{1:T} \mid \hat{\mathbf{z}}_{1:T}) = \prod_{\tau=1}^{T} \left( \sum_{i=1}^{d_x} \log p(\hat{\epsilon}_{x_{\tau,i}}) + \sum_{i=1}^{d_x} \log \left| \frac{\partial w_i}{\partial \hat{s}_{\tau,i}} \right| \right). \tag{13}$$

Specifically, to ensure the conditional independence of $\hat{\mathbf{z}}_t$ and $\hat{\mathbf{s}}_t$, we using $\mathcal{L}_{kl}$ to minimize the KL divergence from the distributions of $\hat{\epsilon}_{x_t}$ and $\hat{\epsilon}_{z_t}$ to the distribution $\mathcal{N}(0, \mathbf{I})$, thereby promoting independence.

**Structure learning.** The variables $r_i$ and $w_i$ are designed to capture causal dependencies among latent and observed variables, respectively. We denote by $\mathbf{J}_r(\hat{\mathbf{z}}_{t-1})$ the Jacobian matrix of the function $r$, which implies the estimated time-lagged latent causal structure; $\mathbf{J}_r(\hat{\mathbf{z}}_t)$, which implies the estimation of instantaneous latent causal structure; and $\mathbf{J}_{\hat{g}}(\hat{\mathbf{x}}_t)$, which implies the estimated observed causal DAG. Considering the observed causal DAG, we first obtain the basic structure of the observed causal DAG by learning a binary mask $\mathcal{M} \sim \text{Ber}(\sigma(\gamma))$, where each edge $\mathcal{M}_{i,j}$ is an

independent Bernoulli random variable with parameter $\sigma(\gamma_{i,j})$. The Gumbel-Softmax technique is employed to learn $\gamma$ (Jang et al., 2017; Maddison et al., 2017), following Ng et al. (2022). Subsequently, we obtained $\mathbf{J}_{\hat{g}_m}(\hat{\mathbf{s}}_t)$ from the `decoder`, and compute the observed causal DAG $\mathbf{J}_{\hat{g}}(\hat{\mathbf{x}}_t)$ via Cor. 3.6. Notably, the entries of $\mathbf{J}_{\hat{g}}(\hat{\mathbf{x}}_t)$ vary with other variables such as $\hat{\mathbf{z}}_t$, resulting in a DAG that may change over time. For the latent structure, we directly compute $\mathbf{J}_r(\hat{\mathbf{z}}_{t-1})$ and $\mathbf{J}_r(\hat{\mathbf{z}}_t)$ from `z-prior` network as the time-lagged structure and instantaneous DAG in latent space, respectively.

To prevent redundant edges and cycles, a sparsity penalty $\mathcal{L}_s$ are imposed on each learned structure, and DAG constraint $\mathcal{L}_d$ are imposed on observed causal DAG and instantaneous latent causal DAG. Specifically, the Markov network structure for latent variables is computed as $\mathcal{M}(\mathbf{J}) = (\mathbf{I}+\mathbf{J})^\top(\mathbf{I}+\mathbf{J})$. Formally, we define the penalties as follows:

$$\sum \mathcal{L}_s = ||\mathcal{M}(\mathbf{J}_r(\hat{\mathbf{z}}_t))||_1 + ||\mathcal{M}(\mathbf{J}_r(\hat{\mathbf{z}}_{t-1}))||_1 + ||\mathbf{J}_{\hat{g}}(\hat{\mathbf{x}}_t)||_1; \quad \sum \mathcal{L}_d = \mathcal{D}(\mathbf{J}_{\hat{g}}(\hat{\mathbf{x}}_t)) + \mathcal{D}(\mathbf{J}_r(\hat{\mathbf{z}}_t)), \quad (14)$$

where $\mathcal{D}(A) = \text{tr}\left[(I + \frac{1}{m}A \circ A)^m\right] - m$ is the DAG constraint from (Yu et al., 2019), with $A$ being an $m$-dimensional matrix. $||\cdot||_1$ denotes the matrix $l_1$ norm. In summary, the overall loss function of the **NCDL** model is formalized as:

$$\mathcal{L}_{all} = \mathcal{L}_{ELBO} + \alpha \sum \mathcal{L}_s + \beta \sum \mathcal{L}_d, \quad (15)$$

where $\alpha = 1.0 \times 10^{-4}$ and $\beta = 5.0 \times 10^{-5}$ are hyperparameters. The discussion on hyperparameter selects is given in Appendix F.1.

## 5 EXPERIMENT

### 5.1 SYNTHETIC DATA

**Empirical study.** The evaluation metrics and their connections to our theorems is elaborated in Appendix F.1. We show performance on of the general nonlinear causal discovery and representation learning in Table 1, and investigate different dimensionalities of observed variables, including $d_x = \{3, 6, 8, 10, 100^*\}$ (* means add mask by simulated inductive bias, see detailed Appendix F.1). Our results on these metrics verify the effectiveness of our methodology under identifiabilty, and the result on $d_x = 100$ with inductive bias makes it scalable to high-dimensional data with prior knowledge of the elimination of some dependences provided by the physical law of climate (Ebert-Uphoff & Deng, 2012) or LLM (Long et al., 2023), supports our subsequent experiment on real-world data. The study on different $d_z$ can be found in Appendix F.1. Additionally, verification of our theoretical assumptions through an ablation study can be found in Appendix F.1.

| $d_z$ | $d_x$ | SHD ($\mathbf{J}_{\hat{g}}(\hat{\mathbf{x}}_t)$) | TPR | Precision | MCC ($\mathbf{s}_t$) | MCC ($\mathbf{z}_t$) | SHD ($\mathbf{J}_r(\hat{\mathbf{z}}_t)$) | SHD ($\mathbf{J}_r(\hat{\mathbf{z}}_{t-1})$) | $R^2$ |
|---|---|---|---|---|---|---|---|---|---|
| 3 | 3 | 0 | 1 | 1 | 0.9775 ($\pm$0.01) | 0.9721 ($\pm$0.01) | 0.27 ($\pm$0.05) | 0.26 ($\pm$0.03) | 0.90 ($\pm$0.05) |
| | 6 | 0.18 ($\pm$0.06) | 0.83 ($\pm$0.03) | 0.80 ($\pm$0.04) | 0.9583 ($\pm$0.02) | 0.9505 ($\pm$0.01) | 0.24 ($\pm$0.06) | 0.33 ($\pm$0.09) | 0.92 ($\pm$0.01) |
| | 8 | 0.29 ($\pm$0.05) | 0.78 ($\pm$0.05) | 0.76 ($\pm$0.04) | 0.9020 ($\pm$0.03) | 0.9601 ($\pm$0.03) | 0.36 ($\pm$0.11) | 0.31 ($\pm$0.12) | 0.93 ($\pm$0.02) |
| | 10 | 0.43 ($\pm$0.05) | 0.65 ($\pm$0.08) | 0.63 ($\pm$0.14) | 0.8504 ($\pm$0.07) | 0.9652 ($\pm$0.02) | 0.29 ($\pm$0.04) | 0.40 ($\pm$0.05) | 0.92 ($\pm$0.02) |
| | 100* | 0.17 ($\pm$0.02) | 0.80 ($\pm$0.05) | 0.81 ($\pm$0.02) | 0.9131 ($\pm$0.02) | 0.9565 ($\pm$0.02) | 0.21 ($\pm$0.01) | 0.29 ($\pm$0.10) | 0.93 ($\pm$0.03) |

Table 1: **Results on different observed dimensionality** $d_x$. We run simulations with 5 random seeds, selected based on the best-converged results to avoid local minima.

**Comparison with constraint-based methods on observed causal DAG.** To the best of our knowledge, no existing method employs a comparably general framework based on structural equation models. Therefore, we compare our approach against two constraint-based methods: FCI (Spantini et al., 2018) and CD-NOD (Huang et al., 2020), both of which are designed to discover causal DAGs while accounting for latent confounders. Additionally, we evaluate methods proposed for causal discovery on time-series data, including PCMCI (Runge et al., 2019), a climate-specific method that incorporates time-lagged and instantaneous effects, as well as LPCMCI (Gerhardus & Runge, 2020), which is proposed for causal discovery in observational time series in the presence of latent confounders and autocorrelation. As illustrated in Fig.6, NCDL demonstrates superior performance across different sample sizes, with further improvements observed as the sample size increases. We observe that FCI struggles when latent confounders are dependent on each other, often resulting in low recall. CD-NOD assumes pseudo-causal sufficiency, requiring latent confounders to be functions of surrogate variables, which is incompatible with general latent variable

settings. PCMCI ignores latent variables and underlying processes, while LPCMCI, despite considering latent variables, cannot handle latent processes and requires the absence of edges among latent confounders. These constraints collectively highlight the advantages of our approach in addressing such challenges. Additional experimental details can be found in AppendixF.1.

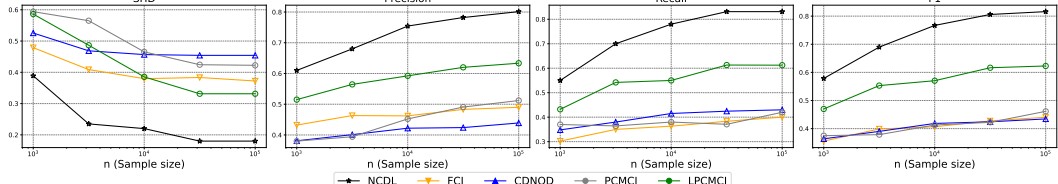

Figure 6: **Comparison with constraint-based methods.** We set $d_x = 6$ and $d_z = 3$. We run experiments using 5 different random seeds, and report the average performance on evaluation metrics.

**Comparison with temporal (causal) representation learning.**  We evaluate our method against the following compared methods including CaRiNG (Chen et al., 2024), TDRL (Yao et al., 2022), LEAP (Yao et al., 2021), SlowVAE (Klindt et al., 2020), PCL (Hyvarinen & Morioka, 2017), i-VAE (Khemakhem et al., 2020), TCL (Hyvarinen & Morioka, 2016), and methods handling instantaneous dependencies including iCITRIS (Lippe et al., 2022) and G-CaRL (Morioka & Hyvärinen, 2023) in Table 2. The dimensions are set to $d_z = 3$ and $d_x = 10$. The MCC and $R^2$ results for the **Independent** and **Sparse** settings demonstrate that our model achieves component-wise identifiability (Theorem 3.3). In contrast, other considered methods fail to recover latent variables, as they cannot properly address cases where the observed variables are causally-related. For the **Dense** setting, our approach achieves monoblock identifiability (Theorem 3.2) with the highest $R^2$, while other methods exhibit significant degradation because they are not specifically tailored to handle scenarios involving general noise in the generating function. These outcomes are consistent with our theoretical analysis.

| Setting | Metric | NCDL | iCITRIS | G-CaRL | CaRiNG | TDRL | LEAP | SlowVAE | PCL | i-VAE | TCL |
|---|---|---|---|---|---|---|---|---|---|---|---|
| **Independent** | MCC | **0.9811** | 0.6649 | 0.8023 | 0.8543 | 0.9106 | 0.8942 | 0.4312 | 0.6507 | 0.6738 | 0.5916 |
| | $R^2$ | **0.9626** | 0.7341 | 0.9012 | 0.8355 | 0.8649 | 0.7795 | 0.4270 | 0.4528 | 0.5917 | 0.3516 |
| **Sparse** | MCC | **0.9306** | 0.4531 | 0.7701 | 0.4924 | 0.6628 | 0.6453 | 0.3675 | 0.5275 | 0.4561 | 0.2629 |
| | $R^2$ | **0.9102** | 0.6326 | 0.5443 | 0.2897 | 0.6953 | 0.4637 | 0.2781 | 0.1852 | 0.2119 | 0.3028 |
| **Dense** | MCC | **0.6750** | 0.3274 | 0.6714 | 0.4893 | 0.3547 | 0.5842 | 0.1196 | 0.3865 | 0.2647 | 0.1324 |
| | $R^2$ | **0.9204** | 0.6875 | 0.8032 | 0.4925 | 0.7809 | 0.7723 | 0.5485 | 0.6302 | 0.1525 | 0.206 |

Table 2: **Experiments results on simulated data.** We consider three scenarios according to our theory: **Independent**: $z_{t,i}$ and $z_{t,j}$ are conditionally independent given $\mathbf{z}_{t-1}$; **Sparse**: $z_{t,i}$ and $z_{t,j}$ are dependent given $\mathbf{z}_{t-1}$, but the latent Markov network $\mathcal{G}_{z_t}$ and time-lagged latent structure are sparse; **Dense**: No sparsity restrictions on latent causal graph. Bold numbers indicate the best performance.

## 5.2 REAL-WORLD DATA

We use the CESM2 sea surface temperature dataset as our real-world data source for temperature forecasting and causal structure learning. Due to page limitations, detailed results and further information are provided in Appendix F.2.

## 6 CONCLUSION

We establish identifiability results for uncovering latent causal variables, latent Markov network, and observed causal DAG, especially in complex nonlinear systems such as climate science. Simulated experiments validate our theoretical findings, and real-world experiments offer causal insights for climate science.

For future work, we aim to address the issue of performance degradation in data with increasing dimensionality. A possible approach to tackling this challenge is to resort to the divide-and-conquer strategy, which partitions the high-dimensional problem into a set of overlapping subsets of variables with lower dimensionality, leveraging the prior knowledge of geographical information.

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

# APPENDIX FOR "IDENTIFICATION OF NONPARAMETRIC DYNAMIC CAUSAL MODEL AND LATENT PROCESS FOR CLIMATE ANALYSIS"

## CONTENTS

# A  NOTATION

| Index | Explanation | Value |
|---|---|---|
| $d_x$ | number of observed variables | $d_x \in \mathbb{N}^+$ |
| $d_z$ | number of latent variables | $d_z \in \mathbb{N}^+$ and $d_z \leq d_x$ |
| $t$ | time index | $t \in \mathbb{N}^+$ and $t \geq 3$ |
| $\mathcal{I}$ | index set of observed variables | $\mathcal{I} = \{1, 2, \ldots, d_x\}$ |
| $\mathcal{J}$ | index set of latent variables | $\mathcal{J} = \{1, 2, \ldots, d_z\}$ |

| Variable | Explanation | Value |
|---|---|---|
| $\mathcal{X}_t$ | support of observed variables in time-index $t$ | $\mathcal{X}_t \subseteq \mathbb{R}^{d_x}$ |
| $\mathcal{Z}_t$ | support of latent variables | $\mathcal{Z}_t \subseteq \mathbb{R}^{d_z}$ |
| $\mathbf{x}_t$ | observed variables in time-index $t$ | $\mathbf{x}_t \in \mathcal{X}_t$ |
| $\mathbf{z}_t$ | latent variables in time-index $t$ | $\mathbf{z}_t \in \mathcal{Z}_t$ |
| $\mathbf{s}_t$ | dependent noise of observations in time-index $t$ | $\mathbf{s}_t \in \mathbb{R}^{d_x}$ |
| $\boldsymbol{\epsilon}_{x_t}$ | independent noise for generating $\mathbf{s}_t$ in time-index $t$ | $\boldsymbol{\epsilon}_{x_t} \sim p_{\epsilon_x}$ |
| $\boldsymbol{\epsilon}_{z_t}$ | independent noise of latent variables in time-index $t$ | $\boldsymbol{\epsilon}_{z_t} \sim p_{\epsilon_z}$ |
| $\mathbf{z}_{t\backslash[i,j]}$ | latent variables except for $z_{t,i}$ and $z_{t,j}$ in time-index $t$ | / |

| Function | Explanation | Value |
|---|---|---|
| $p_a(\cdot \mid b)$ | density function of $a$ given $b$ | / |
| $p_b(a, \cdot \mid c)$ | joint density function of $(a, b)$ given $a$ and $c$ | / |
| $\mathbf{pa}(\cdot)$ | variable's parents | / |
| $\mathbf{pa}_O(\cdot)$ | variable's parents in observed space | / |
| $\mathbf{pa}_L(\cdot)$ | variable's parents in latent space | / |
| $\mathbf{an}_L(\cdot)$ | variable's ancestors in $\mathbf{z}_t$ | / |
| $g(\cdot)$ | generating function of SEM from $(\mathbf{z}_t, \mathbf{s}_t, \mathbf{x}_t)$ to $\mathbf{x}_t$ | $\mathbb{R}^{d_z + 2d_x} \to \mathbb{R}^{d_x}$ |
| $g_m(\cdot)$ | mixing function of ICA from $(\mathbf{z}_t, \mathbf{s}_t)$ to $\mathbf{x}_t$ | $\mathbb{R}^{d_z + d_x} \to \mathbb{R}^{d_x}$ |
| $h_z(\cdot)$ | invertible transformation from $\mathbf{z}_t$ to $\hat{\mathbf{z}}_t$ | $\mathbb{R}^{d_z} \to \mathbb{R}^{d_z}$ |
| $\pi(\cdot)$ | permutation function | $\mathbb{R}^{d_x} \to \mathbb{R}^{d_x}$ |
| $\mathrm{supp}(\cdot)$ | support matrix of Jacobian matrix | $\mathbb{R}^{d_x \times d_x} \to \{0,1\}^{d_x \times d_x}$ |

| Symbol | Explanation | Value |
|---|---|---|
| $\mathcal{G}_{x_t}$ | causal graph among observed variables (observed causal DAG) in $t$ | / |
| $\mathcal{G}_{z_t}$ | causal graph among latent variables in time-index $t$ | / |
| $A \to B$ | $A$ causes $B$ directly | / |
| $A \dashrightarrow B$ | $A$ causes $B$ indirectly | / |
| $\mathbf{J}_g(\mathbf{x}_t)$ | Jacobian matrix representing observed causal DAG | $\mathbf{J}_g(\mathbf{x}_t) \in \mathbb{R}^{d_x \times d_x}$ |
| $\mathbf{J}_g(\mathbf{x}_t, \mathbf{s}_t)$ | Jacobian matrix representing mixing structure from $(\mathbf{x}_t, \mathbf{s}_t)$ to $\mathbf{x}_t$ | $\mathbf{J}_g(\mathbf{x}_t, \mathbf{s}_t) \in \mathbb{R}^{d_x \times d_x}$ |
| $\mathbf{J}_{g_m}(\mathbf{s}_t)$ | Jacobian matrix representing mixing structure from $\mathbf{s}_t$ to $\mathbf{x}_t$ | $\mathbf{J}_{g_m}(\mathbf{s}_t) \in \mathbb{R}^{d_x \times d_x}$ |
| $\mathbf{J}_r(\mathbf{z}_{t-1})$ | Jacobian matrix representing latent time-lagged structure | $\mathbf{J}_r(\mathbf{z}_{t-1}) \in \mathbb{R}^{d_z \times d_z}$ |
| $\mathbf{J}_r(\mathbf{z}_t)$ | Jacobian matrix representing instantaneous latent causal graph | $\mathbf{J}_r(\mathbf{z}_t) \in \mathbb{R}^{d_z \times d_z}$ |

Table 3: List of notations, explanations and corresponding values.

# B  THEOREM PROOFS

## B.1  PROOF OF THEOREM 3.2

**Definition B.1** *(Diagonal Operator) Consider two random variable $a$ and $b$ with support $\mathcal{A}$ and $\mathcal{B}$, a function $f$ is defined on some support $\mathcal{A}$. The diagonal operator $D_{b|a}$ maps the function $f(a)$ to another function $D_{b|a} \circ f(b)$ defined by the pointwise multiplication of $f(x)$ by some function $p_{b|a}$ at a fixed point $b$, $b \in \mathcal{B}$,*

$$f(\mathcal{A}) \to f(\mathcal{B}): \quad D_{b|a} \circ f_a = g_{b|a}(\cdot \mid a) f_a \tag{16}$$

**Definition B.2** *(Completeness) A family of distribution $p(a|b)$ is complete if the only solution $p(a)$ to*

$$\int_A g(a) f_{a|b}(a|b)\, da = 0 \quad \text{for all } b \in \mathcal{B} \tag{17}$$

is $g(a) = 0$. In other words, no matter the range of an operator is on finite or infinite, it is complete if its null space[2] or kernel is a zero set. Completeness is always used to phrase the sufficient and necessary condition for injective linear operator (Newey & Powell, 2003; Chernozhukov et al., 2007)=.

**Theorem B.3** *(Theorem XV 4.5 in (Dunford & Schwartz, 1988) Part III) A bounded operator $T$ is a spectral operator if and only if it is the sum $T = S + N$ of a bounded scalar type operator $S$ and a quasi-nilpotent operator $N$ commuting with $S$. Furthermore, this decomposition is unique and $T$ and $S$ have the same spectrum and the same resolution of the identity.*

**Properties of linear operator**   We first outline useful properties of the linear operator to facilitate understanding of our proof:

i. *(Inverse)* If linear operator: $L_{b|a}$ exists a left-inverse $L_{b|a}^{-1}$, such $L_{b|a}^{-1} \circ L_{b|a} \circ p_a = p_a$ for all $a \in \mathcal{A}$. Analogously, if $L_{b|a}$ exists a right-inverse $L_{b|a}^{-1}$, such $L_{b|a} \circ L_{b|a}^{-1} \circ p_a = p_a$ for all $a \in \mathcal{A}$. If $L_{b|a}$ is bijective, there exists left-inverse and right-inverse which are the same.

ii. *(Injective)* $L_{b|a}$ is said to be an injective linear operator if its $L_{b|a}^{-1}$ is defined over the range of the operator $L_{b|a}$ (Kress et al., 1989). If so, under assumption (i), $L_{a|b}^{-1}$ exists and is densely defined over $\mathcal{F}(\mathcal{A})$. (Hu & Schennach, 2008).

iii. *(Composition)* Given two linear operators $L_{c|b} : \mathcal{F}(\mathcal{C}) \to \mathcal{F}(\mathcal{B})$ and $L_{c|a} : \mathcal{F}(\mathcal{A}) \to \mathcal{F}(\mathcal{C})$, with the function space supports defined uniformly on the range of supports for the domain spaces as characterized by $L_{b|a}$, it follows that $L_{c|a} = L_{c|b} \circ L_{b|a}$. Furthermore, the properties of linearity and associativity are preserved in the operation of linear operators. However, it is crucial to note the non-commutativity of these operators, i.e., $L_{c|b} L_{b|a} \neq L_{b|a} L_{c|b}$, indicating the significance of the order of application.

**Step 1: implications of d-separation.**   The definition of latent causal process indicates that $\mathbf{z}_t$ d-separates $\mathbf{x}_{t-1}, \mathbf{x}_t, \mathbf{x}_{t+1}$, which implies two limited feedbacks:

- $p(\mathbf{x}_{t-1} \mid \mathbf{x}_t, \mathbf{z}_t) = p(\mathbf{x}_{t-1} \mid \mathbf{z}_t)$
- $p(\mathbf{x}_{t+1} \mid \mathbf{x}_t, \mathbf{x}_{t-1}, \mathbf{z}_t) = p(\mathbf{x}_{t+1} \mid \mathbf{z}_t)$.

**Step 2: transformation in function space.**   The observed $p(\mathbf{x}_{t-1})$ and joint distribution $p(\mathbf{x}_{t+1}, \mathbf{x}_t, \mathbf{x}_{t-1})$ directly indicates $p(\mathbf{x}_{t+1}, \mathbf{x}_t \mid \mathbf{x}_{t-1})$, and then the processes of motions are established by noting that

$$
\begin{aligned}
p(\mathbf{x}_{t+1}, \mathbf{x}_t \mid \mathbf{x}_{t-1}) &= \int_{\mathcal{Z}_t} p(\mathbf{x}_{t+1}, \mathbf{x}_t, \mathbf{z}_t \mid \mathbf{x}_{t-1}) d\mathbf{z}_t \\
&= \int_{\mathcal{Z}_t} p(\mathbf{x}_{t+1} \mid \mathbf{x}_t, \mathbf{z}_t, \mathbf{x}_{t-1}) p(\mathbf{x}_t, \mathbf{z}_t \mid \mathbf{x}_{t-1}) d\mathbf{z}_t \\
&= \int_{\mathcal{Z}_t} p(\mathbf{x}_{t+1} \mid \mathbf{z}_t) p(\mathbf{x}_t, \mathbf{z}_t \mid \mathbf{x}_{t-1}) d\mathbf{z}_t \\
&= \int_{\mathcal{Z}_t} p(\mathbf{x}_{t+1} \mid \mathbf{z}_t) p(\mathbf{x}_t \mid \mathbf{z}_t, \mathbf{x}_{t-1}) p(\mathbf{z}_t \mid \mathbf{x}_{t-1}) d\mathbf{z}_t \\
&= \int_{\mathcal{Z}_t} p(\mathbf{x}_{t+1} \mid \mathbf{z}_t) p(\mathbf{x}_t \mid \mathbf{z}_t) p(\mathbf{z}_t \mid \mathbf{x}_{t-1}) d\mathbf{z}_t.
\end{aligned}
\tag{18}
$$

---

[2]The null space or kernel of an operator $L$ to be the set of all vectors which $L$ maps to the zero vector: null $L = \{v \in V : Lv = 0\}$.

We incorporate the integration over $\mathcal{X}_{t-1}$,

$$\int_{\mathcal{X}_{t-1}} p(\mathbf{x}_{t+1}, \mathbf{x}_t \mid \mathbf{x}_{t-1}) p(\mathbf{x}_{t-1}) d\mathbf{x}_{t-1} = \int_{\mathcal{X}_{t-1}} \int_{\mathcal{Z}_t} p(\mathbf{x}_{t+1} \mid \mathbf{z}_t) p(\mathbf{x}_t \mid \mathbf{z}_t) p(\mathbf{z}_t \mid \mathbf{x}_{t-1}) p(\mathbf{x}_{t-1}) d\mathbf{z}_t d\mathbf{x}_{t-1} \tag{19}$$

**Step 3: construct spectral decomposition.** By the definition of linear operator 3.1,

$$\int_{\mathcal{X}_{t-1}} p(\mathbf{x}_{t+1}, \mathbf{x}_t \mid \mathbf{x}_{t-1}) p(\mathbf{x}_{t-1}) d\mathbf{x}_{t-1} = [L_{x_t;x_{t+1}|x_{t-1}} \circ p](\mathbf{x}_{t+1}), \tag{20}$$

where $L_{x_t;x_{t+1}|x_{t-1}} = \int_{\mathcal{Z}_t} p_{x_{t-1}}(\mathbf{x}_t, \cdot \mid \mathbf{x}_{t-1}) p(\mathbf{x}_{t-1}) d\mathbf{x}_{t-1}$. Through the definition of diagonal operator B.1, we have

$$[L_{x_t;x_{t+1}|x_{t-1}} p](\mathbf{x}_{t+1}) = [L_{x_{t+1}|z_t} D_{x_t|z_t} L_{z_t|x_{t-1}} p](\mathbf{x}_{t+1}), \tag{21}$$

which implies the operator equivalence:

$$L_{x_t;x_{t+1}|x_{t-1}} = L_{x_{t+1}|z_t} D_{x_t|z_t} L_{z_t|x_{t-1}}. \tag{22}$$

Let's integrating out $\mathbf{x}_t$. First,

$$\int_{\mathbf{x}_t \in \mathcal{X}_t} L_{x_t;x_{t+1}|x_{t-1}} \, d\mathbf{x}_t = \int_{\mathbf{x}_t \in \mathcal{X}_t} L_{x_{t+1}|z_t} D_{x_t|z_t} L_{z_t|x_{t-1}} \, d\mathbf{x}_t$$

then we get

$$L_{x_{t+1}|x_{t-1}} = L_{x_{t+1}|z_t} L_{z_t|x_{t-1}}. \tag{23}$$

By the Assumption (ii), if the linear operator is injective, by the Lemma 1 in (Hu & Schennach, 2008), $L_{x_{t+1}|z_t}^{-1}$ exists and is densely defined over $f(\mathcal{X}_{t+1})$, then Eq. 23 can be written as

$$L_{x_{t+1}|z_t}^{-1} L_{x_{t+1}|x_{t-1}} = L_{z_t|x_{t-1}}. \tag{24}$$

Then the $L_{z_t|x_{t-1}}$ could be substituted by Eq. 24:

$$L_{x_t;x_{t+1}|x_{t-1}} L_{x_t|x_{t-1}}^{-1} = L_{x_{t+1}|z_t} D_{x_t|z_t} L_{x_{t+1}|z_t}^{-1}. \tag{25}$$

**Step 4: uniqueness of spectral decomposition** By Assumption (ii), the linear operator is bounded. Consequently, $L_{x_t;x_{t+1}|x_{t-1}} L_{x_t|x_{t-1}}^{-1}$ is also bounded, as established in Section XV.4 of Dunford & Schwartz (1988). Therefore, we can apply Theorem XV.4.3.5 in (Dunford & Schwartz, 1988) to demonstrate that the spectral decomposition of $L_{x_t;x_{t+1}|x_{t-1}} L_{x_t,x_{t-1}}^{-1}$ is unique. In particular, the operator $L_{x_{t+1}|z_t} D_{x_t|z_t} L_{x_{t+1}|z_t}^{-1}$ admits the unique spectral decomposition, corresponding eigenfunctions and eigenvalues.

This method leveraging the uniqueness of bounded linear operators discussed in (Dunford & Schwartz, 1988) are commonly utilized in works such as (Hu, 2008; Carroll et al., 2010; Hu & Shum, 2012; Hu & Shiu, 2018), where a spectral decomposition form similar to Eq. 25 is constructed.

Next, we address the indeterminacies associated with this uniqueness and give the solutions from the perspective of eigendecomposition.

**Indeterminacy 1: scaling ambiguity** Eigenfunctions corresponding to a given eigenvalue are not unique under scalar multiplication, as shown below:

$$L_{x_{t+1}|z_t} D_{x_t|z_t} L_{x_{t+1}|z_t}^{-1} = (cL_{x_{t+1}|z_t}) D_{x_t|z_t} (cL_{x_{t+1}|z_t})^{-1},$$

where $c$ is a non-zero constant. Thus, $cL_{x_{t+1}|z_t}$ is an equivalent alternative for the eigenfunction. Since the condition $\int_{\mathcal{X}_{t+1}} p_{x_{t+1}|z_t} dx_{t+1} = 1$ must hold, any arbitrary scaling would imply $\int_{\mathcal{X}_{t+1}} cp_{x_{t+1}|z_t} dx_{t+1} = c$. Setting $c = 1$ is the only way to maintain this normalization, thereby eliminating the scaling ambiguity.

**Indeterminacy 2: eigenvalue degeneracy**  When the matrix $D_{x_t|z_t}$ has repeated eigenvalues, eigenvalue degeneracy occurs and an eigenvalue has more than one corresponding eigenvector. For $\mathbf{z}_t, \mathbf{z}'_t \in \mathcal{Z}_t$ with $\mathbf{z}_t \neq \mathbf{z}'_t$, the probability distributions $p(\mathbf{x}_t \mid \mathbf{z}_t)$ and $p(\mathbf{x}_t \mid \mathbf{z}'_t)$ represent distinct elements within the set of eigenvalues. A mild assumption (iii) ensures that $p(\mathbf{x}_t \mid \mathbf{z}_t) \neq p(\mathbf{x}_t \mid \mathbf{z}'_t)$, thereby preventing the repetition of eigenvalues. After resolving this indeterminacy, we can obtain the complete sets of elements in $D_{x_t|z_t}$ and $D_{x_t|\hat{z}_t}$ as follows:

$$\{p(\mathbf{x}_t \mid \mathbf{z}_t) \mid \forall (\mathbf{x}_t, \mathbf{z}_t) \in \mathcal{X}_t \times \mathcal{Z}_t\} = \{p(\mathbf{x}_t \mid \hat{\mathbf{z}}_t) \mid \forall (\mathbf{x}_t, \hat{\mathbf{z}}_t) \in \mathcal{X}_t \times \hat{\mathcal{Z}}_t\}.$$

This indicates that the sets of eigenvalues in $D_{x_t|z_t}$ and $D_{x_t|\hat{z}_t}$ are identical across their corresponding ranges, which eliminates the eigenvalue degeneracy.

**Indeterminacy 3: ordering ambiguity**  Since the the unordered nature of set, $D_{x_t|\hat{z}_t}$ can be obtained from eigenvalues within $D_{x_t|z_t}$ assigned with an arbitrary order, e.g., all $\mathbf{z}_t$ exchange their rooms ($p(\mathbf{x}_t \mid \mathbf{z}_t)$ here) in the Hilbert's hotel ($D_{x_t|z_t}$ here). Evidently, the ordering permutation is surjective and injective, we can express this process as:

$$\hat{\mathbf{z}}_t = h_z(\mathbf{z}_t),$$

where $h_z$ is an invertible function.

Notably, in the spectral decomposition, resolving the ordering ambiguity presents significant challenges. Hu (2008) addresses this issue by assuming knowledge of the mapping from $p_{x_t}(\cdot \mid \mathbf{z}_t)$ to $\mathbf{z}_t$. This assumption may hold in specific scenarios, particularly in econometrics, where it is common to assume a paradigm such as $\mathbf{z}_t = g(\mathbf{z}_t, 0)$. For example, in a linear additive model with zero-mean noise (e.g., $\mathbf{x}_t = \mathbf{z}_t + \boldsymbol{\epsilon}$), the expected value can serve as a candidate for this mapping: $E[\mathbf{z}_t + \boldsymbol{\epsilon}] = \mathbf{z}_t$. However, identifying such a mapping is nearly impossible when dealing with an unknown, unrestricted general nonlinear function.

**Indeterminacy 4: dimensionality**  We provide a brief proof that the invertible function $h_z$ preserves the dimensionality, that is $d_{\hat{z}} = d_z$. We analyze two scenarios:

    i. $d_{\hat{z}} > d_z$: This implies that only $d_z$ components in $\hat{\mathbf{z}}_t$ are required to reconstruct the observations $\mathbf{x}_t$. Any variation in the remaining $d_{\hat{z}} - d_z$ components would not affect $\mathbf{x}_t$. Let $\hat{\mathbf{z}}_t, \hat{\mathbf{z}}_t$ then we can always find

$$p(\mathbf{x}_t \mid \mathbf{z}_{t,:d_{\hat{z}}-d_z}, \mathbf{z}_{t,d_{\hat{z}}-d_z:}) = p(\mathbf{x}_t \mid \mathbf{z}_{t,:d_{\hat{z}}-d_z}, \mathbf{z}'_{t,d_{\hat{z}}-d_z:}), \tag{26}$$

    which contradicts Assumption (iii).

    ii. $d_{\hat{z}} < d_z$: This suggests that only $d_{\hat{z}}$ dimensions are sufficient to describe $\mathbf{x}_t$, leaving $d_z - d_{\hat{z}}$ components constant, which violates that there are $d_z$ latent *variables*.

In summary, if dimensionality is not preserved, it contradicts the assumptions or the sufficiency of the latent representation.

## B.2    THEOREM 3.3

Once monoblock identifiability is achieved, this work can be further linked to existing research on identification of latent structure (Zhang et al., 2024; Li et al., 2024).

**Theorem B.4**  *Let* $\mathbf{c}_t \triangleq \{\mathbf{z}_{t-1}, \mathbf{z}_t\}$ *and* $\mathcal{M}_{\mathbf{c}_t}$ *be the variable set of two consecutive timestamps and the corresponding Markov network respectively. Suppose the following assumptions hold:*

- *A1 (Smooth and Positive Density): The probability function of the latent variables* $\mathbf{c}_t$ *is smooth and positive, i.e.,* $p_{\mathbf{c}_t}$ *is third-order differentiable and* $p_{\mathbf{c}_t} > 0$ *over* $\mathbb{R}^{2n}$.

- *A2 (Sufficient Variability): Denote* $|\mathcal{M}_{\mathbf{c}_t}|$ *as the number of edges in Markov network* $\mathcal{M}_{\mathbf{c}_t}$. *Let*

$$w(m) = \left( \frac{\partial^3 \log p(\mathbf{c}_t|\mathbf{z}_{t-2})}{\partial c_{t,1}^2 \partial z_{t-2,m}}, \ldots, \frac{\partial^3 \log p(\mathbf{c}_t|\mathbf{z}_{t-2})}{\partial c_{t,2n}^2 \partial z_{t-2,m}} \right) \oplus$$

$$\left( \frac{\partial^2 \log p(\mathbf{c}_t|\mathbf{z}_{t-2})}{\partial c_{t,1} \partial z_{t-2,m}}, \ldots, \frac{\partial^2 \log p(\mathbf{c}_t|\mathbf{z}_{t-2})}{\partial c_{t,2n} \partial z_{t-2,m}} \right) \oplus \left( \frac{\partial^3 \log p(\mathbf{c}_t|\mathbf{z}_{t-2})}{\partial c_{t,i} \partial c_{t,j} \partial z_{t-2,m}} \right)_{(i,j) \in \mathcal{E}(\mathcal{M}_{\mathbf{c}_t})}, \tag{27}$$

where $\oplus$ denotes the concatenation operation and $(i,j) \in \mathcal{E}(\mathcal{M}_{\mathbf{c}_t})$ denotes all pairwise indices such that $c_{t,i}, c_{t,j}$ are adjacent in $\mathcal{M}_{\mathbf{c}_t}$. For $m \in \{1, \dots, n\}$, there exist $4n + 2|\mathcal{M}_{\mathbf{c}_t}|$ different values of $\mathbf{z}_{t-2,m}$ as the $4n + 2|\mathcal{M}_{\mathbf{c}_t}|$ values of vector functions $w(m)$ are linearly independent.

Then for any two different entries $\hat{c}_{t,k}, \hat{c}_{t,l} \in \hat{\mathbf{c}}_t$ that are **not adjacent** in the Markov network $\mathcal{M}_{\hat{\mathbf{c}}_t}$ over estimated $\hat{\mathbf{c}}_t$,

(i) Each ground-truth latent variable $c_{t,i} \in \mathbf{c}_t$ is a function of at most one of $\hat{c}_{t,k}$ and $\hat{c}_{t,l}$.

(ii) For each pair of ground-truth latent variables $c_{t,i}$ and $c_{t,j}$ that are **adjacent** in $\mathcal{M}_{\mathbf{c}_t}$ over $\mathbf{c}_t$, they cannot be a function of $\hat{c}_{t,k}$ and $\hat{c}_{t,l}$ respectively.

**Definition B.5** *(Intimate Neighbor Set (Zhang et al., 2024)) Consider a Markov network $\mathcal{M}_Z$ over variables set $Z$, and the intimate neighbor set of variable $z_{t,i}$ is*

$$\Psi_{\mathcal{M}_{\mathbf{c}_t}}(c_{t,i}) \triangleq \{c_{t,j} \mid c_{t,j} \text{ is adjacent to } c_{t,i}$$
$$\text{and it is also adjacent to all other neighbors of } c_{t,i}, c_{t,j} \in \mathbf{c}_t \backslash \{c_{t,i}\}\}$$

**Theorem B.6** *(Component-wise Identification of Latent Variables with instantaneous dependencies.) Suppose that the observations are generated by Equation (2.2), and $\mathcal{M}_{\mathbf{c}_t}$ is the Markov network over $\mathbf{c}_t = \{\mathbf{z}_{t-1}, \mathbf{z}_t, \mathbf{z}_{t+1}\}$. Except for the assumptions A1 and A2 from Theorem B.4, we further make the following assumption:*

• *A3 (Latent Process Sparsity): For any $z_{t,i} \in \mathbf{z}_t$, the intimate neighbor set of $z_{t,i}$ is an empty set.*

*When the observational equivalence is achieved with the minimal number of edges of the estimated Markov network of $\mathcal{M}_{\hat{\mathbf{c}}_t}$, then we have the following two statements:*

*(i) The estimated Markov network $\mathcal{M}_{\hat{\mathbf{c}}_t}$ is isomorphic to the ground-truth Markov network $\mathcal{M}_{\mathbf{c}_t}$.*

*(ii) There exists a permutation $\pi$ of the estimated latent variables, such that $z_{t,i}$ and $\hat{z}_{t,\pi(i)}$ is one-to-one corresponding, i.e., $z_{t,i}$ is component-wise identifiable.*

**Proof sketch.** The detailed proofs, starting from $\hat{\mathbf{z}}_t = h_z(\mathbf{z}_t)$, follow a similar approach as illustrated in main results of (Zhang et al., 2024), where $\mathbf{z}_{t-1}$ can be considered as auxiliary variables, and (Li et al., 2024). The primary difference is that we recover $\mathbf{z}_t$ by using 3 measurements, while they use the invertibility assumption.

**Broader impact.** We present this result to demonstrate the furthest extent our identification can achieve. Moreover, it highlights the potential of monoblock identifiability from 3-measurement model: previous works of causal representation learning on temporal data (Li et al., 2024; Yao et al., 2022; 2021), data with multiple distribution Zhang et al. (2024), and nonlinear ICA with auxiliary variables Hyvarinen & Morioka (2016); Hyvarinen et al. (2019); Khemakhem et al. (2020) that are still able to achieve the same identifiability from causally-related and/or noise-contaminated observations, provided that our mild assumptions are met.

## B.3 PROOF OF THEOREM 3.5

**Definition B.7** *(Causal Order) An observed variable is in the $\tau$-th causal order if only observed variables in the $(\tau - 1)$-th causal order directly influence it. Specifically, we consider a latent variable $\mathbf{z}_t$ is in the 0-th causal order.*

### B.3.1 PROOF OF RESULT A)

For an observed variable $x_{t,i}$, we define the set $\mathcal{P}$ to include all variables in $\mathbf{x}_t$ involved in generating $x_{t,i}$, initialized as $\mathcal{P} = \mathbf{pa}_O(x_{t,i})$. The upper bound of the cardinality of $\mathcal{P}$ is given by $\mathcal{U}(|\mathcal{P}|)$, which satisfies $\mathcal{U}(|\mathcal{P}|) = d_x - 1$ initially. Let $\mathcal{Q}$ denote the set of latent variables, and define the separated set as $\mathcal{S}$, where $g_{s_i}(\mathbf{pa}_L(x_{t,i}), \epsilon_{x_{t,i}})$ is denoted by $s_{t,i}$. Initially, $\mathcal{S} = \{s_{t,i}\}$. We express $x_{t,i}$ as

$$x_{t,i} = g_i(\mathcal{P}, \mathcal{S}, \mathcal{Q}),$$

and traverse all $x_{t,j} \in \mathbf{x}_t$ in descending causal order $\tau_j$, performing the following operations:

1. Remove $x_{t,j}$ from $\mathcal{P}$ and apply Definition 2.2 to obtain

$$x_{t,i} = f_1\left(\mathcal{P} \setminus \{x_{t,j}\}, \mathcal{S}, \mathcal{Q}, \mathbf{pa}_O(x_{t,j}), \mathbf{pa}_L(x_{t,j}), s_{t,j}\right). \qquad (28)$$

   Then, update $\mathcal{P} \leftarrow (\mathcal{P} \setminus \{x_{t,j}\}) \cup \mathbf{pa}_O(x_{t,j})$ and $\mathcal{Q} \leftarrow \mathcal{Q} \cup \mathbf{pa}_L(x_{t,j})$. By Assumption 2.1, $x_{t,j}$ cannot reappear in the set of its ancestors, resulting in $\mathcal{U}(|\mathcal{P}|) \leftarrow \mathcal{U}(|\mathcal{P}|) - 1$.

2. Assumption 2.1 also ensures that a variable with a lower causal order does not appear in the generation of its descendants. Hence, $x_{t,j}$ cannot appear in the generation of its descendants, since their causal orders are larger than $\tau_j$. Similarly, $s_{t,j}$, which is involved in generating $x_{t,j}$, does not appear in the generation of its descendants. Thus, $s_{t,j} \notin \mathcal{S}$. Define the new separated set as $\mathcal{S} \leftarrow \mathcal{S} \cup \{s_{t,j}\}$, giving

$$x_{t,i} = f_2\left(\mathcal{P}, \mathcal{S}, \mathcal{Q}\right), \qquad (29)$$

   where the new cardinality is updated as $|\mathcal{S}| \leftarrow |\mathcal{S}| + 1$.

Given that $\mathcal{U}(|\mathcal{P}|) \geq |\mathcal{P}|$, $\mathcal{U}(|\mathcal{P}|)$ ensures that this iterative process can be performed until $|\mathcal{P}| = 0$. According to Definition 2.2, all the aforementioned functions are partially differentiable with respect to $\mathbf{s}_t$ and $\mathbf{x}_t$, or they are compositions of such functions. As a result, $\mathcal{Q} = \mathbf{an}_{z_t}(x_{t,i})$, and there exists a function $g_{m_i}$ such that

$$x_{t,i} = g_{m_i}(\mathbf{an}_{z_t}(x_{t,i}), \mathbf{s}_t).$$

Moreover, we observe that $\mathbf{s}_t$ is in fact the ancestors $\mathbf{an}_{\epsilon_{x_t}}(x_{t,i}) = \{\epsilon_{x_{t,j}} \mid s_{t,j} \in \mathcal{S}\}$, which are implied in this derivation process since $\epsilon_{x_{t,j}}$ is in one-to-one correspondence with $s_{t,j}$ through indexing.

### B.3.2 Proof of Result b)

**Bivariate case study.** Initially, we present a *bivariate* example $(x_{t,2} \rightarrow x_{t,1})$ for a better understanding:

$$\begin{cases} x_{t,1} &= g_1(\mathbf{z}_t, x_{t,2}, s_{t,1}) \\ x_{t,2} &= g_2(\mathbf{z}_t, s_{t,2}) \end{cases}, \quad \begin{cases} s_{t,1} &= g_{s_1}(\mathbf{z}_t, \epsilon_1) \\ s_{t,2} &= g_{s_2}(\mathbf{z}_t, \epsilon_2) \end{cases}. \qquad (30)$$

Since nonlinear function $g_{s_1}, g_{s_2}$ are mutable, $x_{t,1} = g_1(\mathbf{z}_t, g_2(\mathbf{z}_t, s_{t,2}), s_{t,1})$. Then

$$\mathbf{J}_g(\mathbf{x}_t) = \begin{bmatrix} 0 & \frac{\partial x_{t,1}}{\partial x_{t,2}} \\ 0 & 0 \end{bmatrix}, \quad \mathbf{J}_{g_m}(\mathbf{s}_t) = \begin{bmatrix} \frac{\partial x_{t,1}}{\partial s_{t,1}} & \frac{\partial x_{t,1}}{\partial x_{t,2}} \cdot \frac{\partial x_{t,2}}{\partial s_{t,2}} \\ 0 & \frac{\partial x_{t,2}}{\partial s_{t,2}}, \end{bmatrix}, \quad \mathbf{D}_{g_m}(\mathbf{s}_t) = \begin{bmatrix} \frac{\partial x_{t,1}}{\partial s_{t,1}} & 0 \\ 0 & \frac{\partial x_{t,2}}{\partial s_{t,2}}, \end{bmatrix},$$

$$\qquad (31)$$

which satisfies $\mathbf{J}_g(\mathbf{x}_t)\mathbf{J}_{g_m}(\mathbf{s}_t) = \mathbf{J}_{g_m}(\mathbf{s}_t) - \mathbf{D}_{g_m}(\mathbf{s}_t)$.

**General multivariate case.** Considering the mixing function $g_m$, and the functional relation $s_{t,j} \rightarrow x_{t,i}$, corresponding $[\mathbf{J}_{g_m}(\mathbf{s}_t)]_{i,j}$, where $i, j$ indicates the row and column index of the Jacobian matrix, respectively.

**For the elements $i \neq j$:** If there is a directed functional relationship $x_{t,j} \rightarrow x_{t,i}$, the corresponding element of the Jacobian matrix is $\frac{\partial x_{t,i}}{\partial x_{t,j}}$. If the relationship is indirect: $x_{t,j} \dashrightarrow x_{t,i}$, then for each $x_{t,k} \in \mathbf{pa}_O(x_{t,i})$, there must exist either an indirect-direct path $x_{t,j} \dashrightarrow x_{t,k} \rightarrow x_{t,i}$ or a direct-direct path $x_{t,j} \rightarrow x_{t,k} \rightarrow x_{t,i}$. By the chain rule, the directed dependence from $s_{t,j}$ to $x_{t,i}$ can only be expressed as the sum of effects through each component of $\mathbf{pa}_O(x_{t,i})$ and itself, allowing $[\mathbf{J}_{g_m}(\mathbf{s}_t)]_{i,j}$ to be decomposed as:

$$[\mathbf{J}_{g_m}(\mathbf{s}_t)]_{i,j} = \sum_{x_{t,k} \in \mathbf{pa}_O(x_{t,i})} \frac{\partial x_{t,i}}{\partial x_{t,k}} \cdot \frac{\partial x_{t,k}}{\partial s_{t,j}}. \qquad (32)$$

For each $x_{t,k} \notin \mathbf{pa}_O(x_{t,i})$, $\frac{\partial x_{t,i}}{\partial x_{t,k}} = 0$, thus Eq. 32 could be rewritten as

$$\begin{aligned} [\mathbf{J}_{g_m}(\mathbf{s}_t)]_{i,j} &= \sum_{x_{t,k} \in \mathbf{pa}_O(x_{t,i})} \frac{\partial x_{t,i}}{\partial x_{t,k}} \cdot \frac{\partial x_{t,k}}{\partial s_{t,j}} + \sum_{x_{t,k} \notin \mathbf{pa}_O(x_{t,i})} \frac{\partial x_{t,i}}{\partial x_{t,k}} \cdot \frac{\partial x_{t,k}}{\partial s_{t,j}} \\ &= \sum_{k=1}^{d_x} \frac{\partial x_{t,i}}{\partial x_{t,k}} \cdot \frac{\partial x_{t,k}}{\partial s_{t,j}} \\ &= \sum_{k=1}^{d_x} [\mathbf{J}_g(\mathbf{x}_t)]_{i,k} \cdot [\mathbf{J}_{g_m}(\mathbf{s}_t)]_{k,j}. \end{aligned} \qquad (33)$$

**For the elements** $i = j$: For each $x_{t,k} \in \mathbf{pa}_O(x_{t,i})$, since the DAG structure ensures $x_{t,i}$ would not appear in the set of ancestors of $x_{t,i}$, then $s_{t,k}$ also would not appear in this set due to its one-to-one relation to $x_{t,i}$, giving that $\frac{\partial x_{t,k}}{\partial s_{t,i}} = 0$. Then we have

$$
\begin{aligned}
[\mathbf{J}_{g_m}(\mathbf{s}_t)]_{i,i} &= \frac{\partial x_{t,i}}{\partial s_{t,i}} + 0 \\
&= \frac{\partial x_{t,i}}{\partial s_{t,i}} + \sum_{k=1}^{d_x} [\mathbf{J}_g(\mathbf{x}_t)]_{i,k} \cdot [\mathbf{J}_{g_m}(\mathbf{s}_t)]_{k,i} \\
&= \frac{\partial x_{t,i}}{\partial s_{t,i}} + \sum_{k=1}^{d_x} [\mathbf{J}_g(\mathbf{x}_t)]_{i,k} \cdot [\mathbf{J}_{g_m}(\mathbf{s}_t)]_{k,i}.
\end{aligned}
\tag{34}
$$

Since if $k = i$, $[\mathbf{J}_g(\mathbf{x}_t)]_{i,k} = 0$; otherwise, if $k \neq i$, $[\mathbf{J}_{g_m}(\mathbf{s}_t)]_{k,i} = 0$. Defining $\mathbf{D}_{g_m}(\mathbf{s}_t) = \text{diag}(\frac{\partial x_{t,1}}{\partial s_{t,1}}, \dots, \frac{\partial x_{t,d_x}}{\partial s_{t,d_x}})$, Finally, we get

$$
\mathbf{J}_g(\mathbf{x}_t)\mathbf{J}_{g_m}(\mathbf{s}_t) = \mathbf{J}_{g_m}(\mathbf{s}_t) - \mathbf{D}_{g_m}(\mathbf{s}_t).
\tag{35}
$$

### B.4 PROOF OF COROLLARY 3.6

Eq. 35 states that
$$
(\mathbf{I}_{d_x} - \mathbf{J}_g(\mathbf{x}_t))\mathbf{J}_{g_m}(\mathbf{s}_t) = \mathbf{D}_{g_m}(\mathbf{s}_t).
\tag{36}
$$
From DAG 2.1 and Assumption 3.4, $\mathbf{J}_g(\mathbf{x}_t)$ represents a DAG structure and can thus be permuted into a lower triangular form using identical row and column permutations. As a result, $\mathbf{I}_{d_x} - \mathbf{J}_g(\mathbf{x}_t)$ is an invertible matrix for all $\mathbf{x}_t \in \mathcal{X}_t$. Consequently, $(\mathbf{I}_{d_x} - \mathbf{J}_g(\mathbf{x}_t))^{-1}\mathbf{D}_{g_m}(\mathbf{s}_t)$ must be invertible, which implies that $\mathbf{J}_{g_m}(\mathbf{s}_t)$ is an invertible matrix.

Additionally, we have
$$
\text{supp}\,(\mathbf{I}_{d_x} - \mathbf{J}_g(\mathbf{x}_t)) = \text{supp}\,(\mathbf{J}_g(\mathbf{x}_t, \mathbf{s}_t))
\tag{37}
$$
because the diagonal entries of $\mathbf{J}_g(\mathbf{x}_t, \mathbf{s}_t)$ are non-zero. Therefore, $\mathbf{J}_g(\mathbf{x}_t, \mathbf{s}_t)$ is also invertible because of the property of a permuted lower triangle.

By exchanging the positions of $\mathbf{J}_g(\mathbf{x}_t)\mathbf{J}_{g_m}(\mathbf{s}_t)$ and $\mathbf{D}_{g_m}(\mathbf{s}_t)$, and then multiplying on the right by $\mathbf{I}_{d_x} - \mathbf{J}_g(\mathbf{x}_t)$, the desired result is obtained directly.

**Remark B.8** *Consider equation on Cor. 3.6, there might be confusion regarding whether the diagonal elements of $\mathbf{J}_g(\mathbf{x}_t)$ are zero. Since $\mathbf{J}_{g_m}(\mathbf{s}_t)$ can be rearranged into a lower triangular matrix through row and column permutations, the diagonal elements of its inverse are simply the reciprocals of its diagonal entries. Consequently, the diagonal entries of $\mathbf{D}_{g_m}(\mathbf{s}_t)\mathbf{J}_{g_m}(\mathbf{s}_t)^{-1}$ are all equal to 1.*

### B.5 PROOF OF THEOREM 3.8

**Definition B.9** *(Ordered Component-wise Identifiability)* *Variables $\mathbf{s}_t \in \mathbb{R}^{d_x}$ and $\hat{\mathbf{s}}_t \in \mathbb{R}^{d_x}$ are identified component-wise if there exists a permutation $\pi$, such that $\hat{s}_{t,i} = h_{s_i}(s_{t,\pi(i)})$ with invertible function $h_{s,i}$ and $\pi(i) = i$.*

**Explanation.** Ordered Component-wise Identifiability implies that the estimated value $\hat{s}_{t,i}$ contains complete information about $s_{t,i}$ while being entirely independent of $s_{t,j}$ for $j \neq \pi(i)$. Notably, the permutation $\pi(\cdot)$ is an identity function, distinguishing this concept from the permutation indeterminacy commonly encountered in nonlinear ICA (Hyvarinen et al., 2019).

**Lemma B.10** *(Lemma 1 in LiNGAM (Shimizu et al., 2006))* *For any invertible lower triangular matrix, a permutation of rows and columns of it has only non-zero entries in the diagonal if and only if the row and column permutations are equal.*

Let $(\hat{\mathbf{z}}_t, \hat{\mathbf{s}}_t, \hat{g}_m)$ be the estimations of $(\mathbf{z}_t, \mathbf{s}_t, g_m)$. By Theorem a),
$$
\mathbf{x}_t = g_m(\mathbf{z}_t, \mathbf{s}_t); \quad \hat{\mathbf{x}}_t = \hat{g}_m(\hat{\mathbf{z}}_t, \hat{\mathbf{s}}_t)
\tag{38}
$$

Suppose we are able to reconstruct observations successfully then $\mathbf{x}_t = \hat{\mathbf{x}}_t$. By the Thm. 3.2, $\hat{\mathbf{z}}_t = h_z(\mathbf{z}_t)$ tells us that

$$p(\mathbf{x}_t \mid \hat{\mathbf{z}}_t) = p(\mathbf{x}_t \mid h_z(\mathbf{z}_t)) = p(\mathbf{x}_t \mid \mathbf{z}_t), \tag{39}$$

since $h_z$ is invertible. Now we show that how to convert it to the quantified relationship between $\mathbf{s}_t$ and $\hat{\mathbf{s}}_t$. By the Eq.38,

$$p(\mathbf{x}_t \mid \mathbf{z}_t) = p(g_m(\mathbf{s}_t, \hat{\mathbf{z}}_t) \mid \mathbf{z}_t); \quad p(\mathbf{x}_t \mid \hat{\mathbf{z}}_t) = p(\hat{g}_m(\hat{\mathbf{s}}_t, \hat{\mathbf{z}}_t) \mid \hat{\mathbf{z}}_t). \tag{40}$$

Then by Eq. 39, we have

$$p(g_m(\mathbf{s}_t, \mathbf{z}_t) \mid \mathbf{z}_t) = p(\hat{g}_m(\hat{\mathbf{s}}_t, \hat{\mathbf{z}}_t) \mid \hat{\mathbf{z}}_t). \tag{41}$$

By the defination of partial Jacobian matrix,

$$[\mathbf{J}_{g_m}(\mathbf{s}_t)]_{i,j} = \frac{\partial x_{t,i}}{\partial s_{t,j}} = \frac{g_{m_i}(\mathbf{s}_t, \mathbf{z}_t)}{\partial s_{t,j}}, \tag{42}$$

which is also set up for $\hat{g}_m$. Thm b) has shown that $\mathbf{J}_{g_m}(\mathbf{s}_t)$ and $\mathbf{J}_{\hat{g}_m}(\hat{\mathbf{s}}_t)$ are invertible matrices, with the change of variables formula,

$$\frac{1}{|\mathbf{J}_{g_m}(\mathbf{s}_t)|} p(\mathbf{s}_t \mid \mathbf{z}_t) = \frac{1}{|\mathbf{J}_{\hat{g}_m}(\hat{\mathbf{s}}_t)|} p(\hat{\mathbf{s}}_t \mid \mathbf{z}_t). \tag{43}$$

We define $h_s := g_m^{-1} \circ \hat{g}_m$, then its correspinding Jacobian matrix $|\mathbf{J}_{h_s}(\hat{\mathbf{s}}_t)| = \frac{|\mathbf{J}_{\hat{g}_m}(\hat{\mathbf{s}}_t)|}{|\mathbf{J}_{g_m}(\mathbf{s}_t)|}$. Obviously $\hat{\mathbf{s}}_t = h_s(\mathbf{s}_t)$, and

$$p(\hat{\mathbf{s}}_t \mid \mathbf{z}_t) = \frac{1}{|\mathbf{J}_{h_s}(\hat{\mathbf{s}}_t)|} p(\mathbf{s}_t \mid \mathbf{z}_t)$$

$$\log p(\hat{\mathbf{s}}_t \mid \hat{\mathbf{z}}_t) = \log p(\mathbf{s}_t \mid \mathbf{z}_t) - \log |\mathbf{J}_{h_s}(\hat{\mathbf{s}}_t)|. \tag{44}$$

Since for any $(i, j, t) \in \mathcal{J} \times \mathcal{J} \times \mathcal{T}$, we have $s_{t,i} \perp\!\!\!\perp s_{t,j} \mid \mathbf{z}_t$. By (Lin, 1997),

$$\frac{\partial^2 \log p(\hat{\mathbf{s}}_t \mid \hat{\mathbf{z}}_t)}{\partial \hat{s}_{t,i} \partial \hat{s}_{t,j}} = 0. \tag{45}$$

To see what it implies, we show second-order partial derivative of $\log p(\hat{\mathbf{s}}_t \mid \hat{\mathbf{z}}_t)$ w.r.t. $(\hat{s}_{t,i}, \hat{s}_{t,j})$ is

$$\frac{\partial \log p(\hat{\mathbf{s}}_t \mid \hat{\mathbf{z}}_t)}{\partial \hat{s}_{t,i}} = \sum_{k=1}^{n} \frac{\partial \mathbf{A}_{t,k}}{\partial s_{t,k}} \cdot \frac{\partial s_{t,k}}{\partial \hat{s}_{t,i}} - \frac{\partial \log |\mathbf{J}_{h_s}(\hat{\mathbf{s}}_t)|}{\partial \hat{s}_{t,i}} = \sum_{k=1}^{n} \frac{\partial \mathbf{A}_{t,k}}{\partial s_{t,k}} \cdot [\mathbf{J}_{h_s}(\hat{\mathbf{s}}_t)]_{k,i} - \frac{\partial \log |\mathbf{J}_{h_s}(\hat{\mathbf{s}}_t)|}{\partial \hat{s}_{t,i}},$$

$$\frac{\partial^2 \log p(\hat{\mathbf{s}}_t \mid \hat{\mathbf{z}}_t)}{\partial \hat{s}_{t,i} \partial \hat{s}_{t,j}} = \sum_{k=1}^{n} \left( \frac{\partial^2 \mathbf{A}_{t,k}}{\partial s_{t,k}^2} \cdot [\mathbf{J}_{h_s}(\hat{\mathbf{s}}_t)]_{k,i} \cdot [\mathbf{J}_{h_s}(\hat{\mathbf{s}}_t)]_{k,j} + \frac{\partial \mathbf{A}_{t,k}}{\partial s_{t,k}} \cdot \frac{\partial [\mathbf{J}_{h_s}(\hat{\mathbf{s}}_t)]_{k,i}}{\partial \hat{s}_{t,j}} \right) - \frac{\partial^2 \log |\mathbf{J}_{h_s}(\hat{\mathbf{s}}_t)|}{\partial \hat{s}_{t,i} \partial \hat{s}_{t,j}}. \tag{46}$$

Therefore, for each value $z_{t,l}, l \in \mathcal{J}$, its partial derivative w.r.t. $z_{t,l}$ is always 0. That is,

$$\frac{\partial^3 \log p(\hat{\mathbf{s}}_t \mid \hat{\mathbf{z}}_t)}{\partial \hat{s}_{t,i} \partial \hat{s}_{t,j} \partial z_{t,l}} = \sum_{k=1}^{n} \left( \frac{\partial^3 \mathbf{A}_{t,k}}{\partial s_{t,k}^2 \partial z_{t,l}} \cdot [\mathbf{J}_{h_s}(\hat{\mathbf{s}}_t)]_{k,i} \cdot [\mathbf{J}_{h_s}(\hat{\mathbf{s}}_t)]_{k,j} + \frac{\partial^2 \mathbf{A}_{t,k}}{\partial s_{t,k} \partial z_{t,l}} \cdot \frac{\partial [\mathbf{J}_{h_s}(\hat{\mathbf{s}}_t)]_{k,i}}{\partial \hat{s}_{t,j}} \right) \equiv 0, \tag{47}$$

where we have made use of the fact that entries of $\mathbf{J}_{h_s}(\hat{\mathbf{s}}_t)$ do not depend on $z_{t,l}$.

By Assumption 3.8, since each term in the equation is linearly independent, maintaining the equality requires setting $[\mathbf{J}_{h_s}(\hat{\mathbf{s}}_t)]_{k,i} \cdot [\mathbf{J}_{h_s}(\hat{\mathbf{s}}_t)]_{k,j} = 0$ for $i \neq j$. This implies that each row of $\mathbf{J}_{h_s}(\hat{\mathbf{s}}_t)$ contains at most one non-zero entry, corresponding to an unnormalized permutation matrix.

**DAG eliminates permutation indeterminacy of ICA.** Next, we show that the structure of a DAG inherently avoids such a permutation (Shimizu et al., 2006; Reizinger et al., 2023). We leverage the following properties:

1. The inverse of a lower triangular matrix remains a lower triangular matrix.

2. A matrix representing a DAG can always be permuted into a lower-triangular form using appropriate row and column permutations.

3. Corollary 3.6 of functional equivalence, which states that:

$$\mathbf{J}_{g^L}(\mathbf{x}_t) = \mathbf{I}_{d_x} - \mathbf{D}_{g_m^L}(\mathbf{s}_t)\mathbf{J}_{g_m^L}^{-1}(\mathbf{s}_t); \quad \mathbf{J}_g(\mathbf{x}_t) = \mathbf{I}_{d_x} - \mathbf{D}_{g_m}(\mathbf{s}_t)\mathbf{J}_{g_m}^{-1}(\mathbf{s}_t) \tag{48}$$

where $\mathbf{J}_{g^L}(\mathbf{x}_t)$ and $\mathbf{J}_{g_m^L}(\mathbf{s}_t)$ are (strictly) lower triangular matrices obtained by permuting $\mathbf{J}_g(\mathbf{x}_t)$ and $\mathbf{J}_{g_m}(\mathbf{s}_t)$, respectively. $\mathbf{D}_{g_m^L}(\mathbf{s}_t)$ is the diagonal matrix extracted from $\mathbf{J}_{g_m^L}(\mathbf{s}_t)$. Consequently, we can express the relationship between $\mathbf{J}_{g_m}(\mathbf{s}_t)$ and $\mathbf{J}_{g_m^L}(\mathbf{s}_t)$ as follows:

$$\mathbf{J}_{g^L}(\mathbf{x}_t) = \mathbf{P}_{d_x}\mathbf{J}_g(\mathbf{x}_t)\mathbf{P}_{d_x}^\top \implies \mathbf{J}_{g_m}(\mathbf{s}_t) = \mathbf{P}_{d_x}\mathbf{J}_{g_m^L}(\mathbf{s}_t)\mathbf{D}_{g_m^L}^{-1}(\mathbf{s}_t)\mathbf{P}_{d_x}^\top\mathbf{D}_{g_m}(\mathbf{s}_t), \tag{49}$$

where $\mathbf{P}_{d_x}$ is the Jacobian matrix of a permutation function on the $d_x$-dimensional vector. Consequently, by $\mathbf{J}_{g_m}(\mathbf{s}_t) = \mathbf{J}_{\hat{g}_m}(\hat{\mathbf{s}}_t)\mathbf{J}_{h_s}(\mathbf{s}_t)$, we obtain

$$\mathbf{J}_{\hat{g}_m}(\hat{\mathbf{s}}_t) = \mathbf{P}_{d_x}\mathbf{J}_{g_m^L}(\mathbf{s}_t)\mathbf{D}_{g_m^L}^{-1}(\mathbf{s}_t)\mathbf{P}_{d_x}^\top\mathbf{D}_{g_m}(\mathbf{s}_t)\mathbf{J}_{h_s}^{-1}(\mathbf{s}_t), \tag{50}$$

Using Lemma B.10, we obtain $\mathbf{P}_{d_x}\mathbf{D}_{g_m^L}^{-1}(\mathbf{s}_t)\mathbf{P}_{d_x}^\top\mathbf{D}_{g_m}(\mathbf{s}_t)\mathbf{J}_{h_s}(\hat{\mathbf{s}}_t) = \mathbf{I}_{d_x}$, which implies $\mathbf{J}_{h_s}^{-1}(\mathbf{s}_t) = \mathbf{D}_{g_m}^{-1}(\mathbf{s}_t)\mathbf{D}_{g_m^L}(\mathbf{s}_t)$, a diagonal matrix. Consequently, $\mathbf{J}_{\hat{g}_m}(\hat{\mathbf{s}}_t)$ and $\mathbf{J}_{g_m}(\mathbf{s}_t)$ have the same support, meaning $\mathbf{J}_{\hat{g}}(\hat{\mathbf{x}}_t)$ and $\mathbf{J}_g(\mathbf{x}_t)$ share the same support as well, according to Corollary 3.6. Thus, by Assumption 3.4, the structure of observed causal DAG is identifiable.

**Remark B.11** *Since the Jacobian matrix changes w.r.t other variables in the function, the scaling of $\mathbf{J}_g(\mathbf{x}_t)$ is also not invariant. However, due to the support stability (changes in the modulus of non-zero elements in a lower triangular matrix do not alter the support of this matrix inverse), the support of $\mathbf{I}_{d_x} - \mathbf{D}_{g_m}(\mathbf{s}_t)\mathbf{J}_{g_m}^{-1}(\mathbf{s}_t)$ remains invariant under scaling indeterminacy of $\mathbf{J}_{g_m}(\mathbf{s}_t)$, which is resulted by that the identification of $\mathbf{s}_t$ can only be achieved in a component-wise manner.*

## B.6 Proof of Corollary 3.9

Since the transformation from $\mathbf{s}_t$ to $\mathbf{x}_t$ is invertible and deterministic, the probability density function for $\mathbf{x}_t$ can be expressed as:

$$p(\mathbf{x}_t) = \begin{cases} \frac{1}{|\mathbf{J}_{g_m}(\mathbf{s}_t)|}p(\mathbf{s}_t), & \mathbf{x}_t = g_m(\mathbf{s}_t) \\ 0, & \mathbf{x}_t \neq g_m(\mathbf{s}_t) \end{cases}$$

Hence, the conditional probability can be represented using the Dirac delta function:

$$p(\mathbf{x}_t \mid \mathbf{s}_t) = \delta(\mathbf{x}_t - g_m(\mathbf{s}_t)).$$

By recalling Def. 3.1, we can rewrite $p(\mathbf{x}_t)$ in terms of the linear operator $L_{x_t|s_t}$ acting on $p_{s_t}$:

$$p(\mathbf{x}_t) = L_{x_t|s_t} \circ p_{s_t}(\mathbf{x}_t) = \int_{\mathcal{S}_t} \delta(\mathbf{x}_t - g_m(\mathbf{s}_t))p(\mathbf{s}_t)\,d\mathbf{s}_t.$$

We consider $p(\mathbf{x}_t \mid \mathbf{s}_t)$ as an infinite-dimensional vector, and the operator $L_{x_t|s_t}$ as an infinite-dimensional matrix:

$$L_{x_t|s_t} = [\delta(\mathbf{x}_t - g_m(\mathbf{s}_t))]_{\mathbf{x}_t \in \mathcal{X}_t}^\top.$$

By Corollary 3.6, since $\mathbf{J}_{g_m}(\mathbf{s}_t)$ is invertible, for any two different points $\mathbf{s}_t, \mathbf{s}'_t \in \mathcal{S}_t$ ($\mathbf{s}_t \neq \mathbf{s}'_t$), we have $g_m(\mathbf{s}_t) \neq g_m(\mathbf{s}'_t)$. This implies that the supports of $\delta(\mathbf{x}_t - g_m(\mathbf{s}_t))$ and $\delta(\mathbf{x}_t - g_m(\mathbf{s}'_t))$ are disjoint. Thus, $[\delta(\mathbf{x}_t - g_m(\mathbf{s}_t))]_{\mathbf{x}_t \in \mathcal{X}_t}^\top$ forms an infinite-dimensional permutation matrix, ensuring:

$$\text{null } [\delta(\mathbf{x}_t - g_m(\mathbf{s}_t))]_{\mathbf{x}_t \in \mathcal{X}_t}^\top = \{0^{(\infty)}\},$$

which denotes the definition of completeness as stated in Definition B.2, indicating that $L_{x_t|s_t}$ is injective.

**Explanation.** By the nature of distribution transformation, we can decouple the linear operator mentioned in Assumption (ii) in two parts: deterministic measurements (operator between $\mathbf{s}_t$ and $\mathbf{x}_t$) and noised measurements:

$$L_{x_{t+1}|z_t} = L_{x_{t+1}|s_{t+1}} \circ L_{s_{t+1}|z_t}; \quad L_{x_{t-1}|x_{t+1}} = L_{x_{t-1}|s_{t-1}} \circ L_{s_{t-1}|s_{t+1}} \circ L_{s_{t+1}|x_{t+1}} \tag{51}$$

This corollary demonstrates that the observed causal DAG does not affect the overall injectivity of the linear operator, thereby relaxing Assumption (ii). Consequently, we can focus solely on the completeness of the noisy measurement processes $L_{s_{t+1}|z_t}$ and $L_{s_{t-1}|s_{t+1}}$. These formulations, as defined in Def. 2.2, have been extensively studied in prior works (D'Haultfoeuille, 2011; Hu & Shiu, 2018; Mattner, 1993), suggesting that the completeness conditions are not difficult to satisfy. Furthermore, this result ensures that the generation variability and transition variability assumptions made here are compatible with the injectivity of $L_{s_{t+1}|z_t}$ and $L_{s_{t-1}|s_{t+1}}$, aiming to ensure that $g_s$ and $\epsilon_{x_t}$ provide sufficient variability to $\mathbf{z}_t$.

## C  ASSUMPTION DISCUSSIONS

**Injectivity of linear operator**   It is currently difficult to formalize precise conditions for injectivity or completeness. Specifically, if $p(a \mid b)$ can be expressed as $p_\epsilon(a - b)$, such as in linear additive noise models, then $L_{b|a}$ is injective if and only if the Fourier transform of $p_\epsilon$ is non-vanishing everywhere (Mattner, 1993). For instance, the Fourier transform of a Gaussian distribution is $\hat{p}_\epsilon(k) = \exp\left(-\frac{\sigma^2 k^2}{2}\right)$. However, this condition is quite restrictive as it imposes stringent smoothness and decay properties on $p_\epsilon$, which are not always observed in real-world distributions. For example, the Laplace distribution has a Fourier transform that decays to zero, while the uniform distribution has zeros at regular intervals, failing to meet the requirement of non-vanishing everywhere in $\mathbb{R}$. Similar results for more general distribution families can be found in (D'Haultfoeuille, 2011).

In contrast, conditional heteroscedasticity can substantially relax these strong requirements on $\epsilon$. This is because completeness demands linear independence of all $p_b(\cdot \mid a)$ over the infinite space of $a \in \mathcal{A}$. When $p(a)$ changes, $p(b)$ undergoes a non-trivial variation, ensuring that the operator $L_{b|a}$ remains "non-singular." In our context, the effects of latent climate variables on the noise term, such as human activities, are significant. Consequently, completeness holds since Corollary 3.9 guarantees that the relationships in $\mathbf{x}_t$ do not disrupt these conditions.

**Discussion of functional faithfulness.**   Functional faithfulness corresponds to the **edge minimality** Zhang (2013); Lemeire & Janzing (2013); Peters et al. (2017) for the Jacobian matrix $\mathbf{J}_g(\mathbf{x}_t)$ representing the nonlinear SEM $\mathbf{x}_t = g(\mathbf{x}_t, \mathbf{z}_t, \boldsymbol{\epsilon}_{x_t})$, where $\frac{\partial x_{t,j}}{\partial x_{t,i}} = 0$ implies no causal edge, and $\frac{\partial x_{t,j}}{\partial x_{t,i}} \neq 0$ indicates causal relation $x_{t,i} \to x_{t,j}$. This assumption is fundamental to ensuring that Jacobian matrix reflects the true causal graph. If our functional faithfulness is violated, the results can be misleading, but in theory (classical) faithfulness Spirtes et al. (2001) is generally possible as discussed in Lemeire & Janzing (2013) (2.3 Minimality). As a weaker version of it, edge minimality holds the same property. If needed, violations of faithfulness can be testable except in the triangle faithfulness situation Zhang (2013). As opposed to classical faithfulness Spirtes et al. (2001), for example, this is not an assumption about the underlying world. It is a convention to avoid redundant descriptions.

**Extension to multiple time lags.**   The theoretical results are still valid when extending beyond a first-order Markov process. For example, in a second-order Markov process, 6 measurements are needed, with each consecutive pair of observed variables representing a unique measurement group. The central latent variables d-separate these groups, ensuring monoblock identifiability. Other identifiability properties are similarly preserved, depending on the model's structure and functional form.

**Surrogates of dependent noise.**   Conditional independence of $\mathbf{s}_t$ is the primary key for discovering observed causal DAG. In scientific applications where the noise $\mathbf{s}_t$ cannot be presumed to be dependent on $\mathbf{z}_t$ or other things, one may resort to alternative constraints, such as structural sparsity (Zheng et al., 2022; Zheng & Zhang, 2023) or multi-domain frameworks (Hyvarinen et al., 2019; Khemakhem et al., 2020; Hyvärinen et al., 2023; Zhang et al., 2024), refrain from imposing generation variability. Nevertheless, these modifications inevitably constrain the generality of the methodology, which limits its applicability to climate science.

## D  BROADER IMPACT

**Broader applications of 3-measurement model.**   In contrast to directly defining $\mathbf{x}_{t-1}$, $\mathbf{x}_t$ and $\mathbf{x}_{t+1}$ as measurements of $\mathbf{z}_t$ are widely used in nonparametric identification in economic (Hu & Schennach, 2008; Carroll et al., 2010), which anchors the 'reference frame' to latent variables and allows for variability in the measurement process, our definition pertains to fixing the reference frame to observations, which ensures an invariant measurement process while the latent variables vary, which is commonplace to encounter such varying in causal mechanisms in practice (arising from heterogeneous data or time series), thereby marrying it to numerous causal representation learning tasks, including climate analysis (Brouillard et al.; Yao et al., 2024), video understanding (Yao et al.,

2022; Chen et al., 2024), natural language processing (Yan et al., 2024; Rajendran et al., 2024), and multi-view data such as 3D vision (Yao et al., 2023), multimodality (Morioka & Hyvarinen, 2023).

Crucially, in our case where climate data (Rasp et al., 2020; Kaltenborn et al., 2023) is typical a discrete time-series data, $\mathcal{T}$ would be a subset of $\mathbb{N}^+$, similar to the tasks in (Hyvarinen & Morioka, 2017; Yao et al., 2022; Lippe et al., 2022), but $\mathcal{T}$ is not restricted to time-series which can be a discrete set aims at indexing measurement on arbitrary dimension. For the supervised learning (Hyvarinen et al., 2019; Zhang et al., 2013), it can represent class labels or domain variables. For economic measurements, it can index the measuring objective such as group in the investigation of self-reported education (Kane et al., 1999).

**Generalization of identifiability.** The findings above lay the groundwork for our subsequent analysis by elucidating the relationship between the estimated latent variables $\hat{\mathbf{z}}_t$ and the true variables $\mathbf{z}_t$, derived from nonparametric observations under 3 different measurement settings. These results generalize to broader causal discovery tasks that involve hidden confounders (Cai et al., 2023; Kong et al., 2023; Spirtes et al., 2013; Huang et al., 2022; Li et al., 2023), if the completeness of linear operators holds true. Moreover, the approach is not restricted to identifying latent variables from causally-related observations, but allows the identification in noisy observations beyond linear additive models (Khemakhem et al., 2020; Hälvä et al., 2021; Gassiat et al., 2020) and replies why, loosely speaking, the invertible neural network is not necessary in practices with the past of the components in the time series, as mentioned in (Hyvarinen & Morioka, 2017; Hyvarinen et al., 2019). The analysis here can also be extended to discrete cases, where the linear operator may be finite, differing only in the conditions stated in Assumption (i).

**Connections to nonlinear ICA.** Non-invertible methods (Chen et al., 2024) leverage temporal context information to recover latent variables (in our case, $\hat{\mathbf{z}} = h_z(\mathbf{z})$). Similarly, our approach does not rely on additional invertibility assumptions for the mixing functions (Zheng et al., 2022; Kong et al., 2023; Lachapelle et al., 2024) and utilizes contextual information, such as the temporal structure, to provide more general results, even in the presence of general nonclassical noise.

# E    EXTENDED RELATED WORK

**Causal discovery algorithms in climate.** A prominent approach for causal discovery in climate analysis is PCMCI (Runge et al., 2019), which is specifically designed for linearly dependent time-series data. PCMCI effectively captures time-lagged dependencies and instantaneous relationships. Subsequently, (Runge, 2020) extended this method to handle nonlinear scenarios. However, these methods do not account for latent variables, which limits their ability to accurately model real-world climate systems. Recently, several causal representation learning methods inspired by climate science have been developed. For example, (Brouillard et al.) assumes single-node structures to achieve identifiability, while (Yao et al., 2024) employs an ODE-based approach to gain insights into climate-zone classification. Nevertheless, these approaches still overlook the dependencies among observed variables.

**Comparisons with Jacobian-based methods.** Nonlinear causal discovery methods often leverage the Jacobian matrix or its byproduct to identify DAGs and ensure identifiability. For example, LiNGAM (Shimizu et al., 2006) uses a mixing matrix in linear settings, while (Lachapelle et al., 2019) and (Rolland et al., 2022) apply the Jacobian to nonlinear models for acyclicity constraints. In dynamical systems, (Atanackovic et al., 2024) adopt a Bayesian approach using Jacobians of SEMs, and (Zheng et al., 2023) learn Markov structures using the Jacobian of the data generation process. Jacobian properties also support identifiability in IMA (Gresele et al., 2021) and causal models with non-i.i.d. data (Reizinger et al., 2023), while (Liu et al., 2024) handle mixed models with score-based method. Table 4 summarizes these methods against our approach.

| Method | $f$ | Data | $J$ | CD | CRL | Identifiability |
|---|---|---|---|---|---|---|
| (Shimizu et al., 2006) | Linear | Non-Gaussian | $J_{f^{-1}}$ | ✓ | × | ✓ |
| (Lachapelle et al., 2019) | Additive | Gaussian | $J_{f^{-1}}$ | ✓ | × | × |
| (Gresele et al., 2021) | IMA | All | $J_f$ | × | × | ✓ |
| (Zheng et al., 2023) | Sparse | All | $J_f$ | × | × | ✓ |
| (Rolland et al., 2022) | Additive | Gaussian | $J_{\nabla_x \log p(x)}$ | ✓ | × | × |
| (Atanackovic et al., 2024) | Cyclic (ODE) | All | $J_f$ | ✓ | × | × |
| (Reizinger et al., 2023) | All | Assums. 2, F. 1 | $J_{f^{-1}}$ | ✓ | × | ✓ |
| (Liu et al., 2024) | Mixed | Gaussian | $J_{\nabla_x \log p(x)}$ | ✓ | × | Partial |
| **Ours** | All | All | $J_{f^{-1}}$ | ✓ | ✓ | ✓ |

Table 4: Comparison of different methods based on their property in function type ($f$), data, Jacobian ($J$), causal discovery (CD), causal represnetation learning (CRL), and achievement of identifiability.

## F  EXPERIMENT DETAILS

### F.1  ON SIMULATION DATASET

**Evaluation metrics.** Due to the nature of monoblock identifiability (Theorem 3.2), we use the coefficient of determination $R^2$ between the estimated variables $\hat{\mathbf{z}}_t$ and the true variables $\mathbf{z}_t$, where $R^2 = 1$ indicates perfect alignment. We employ kernel regression with a Gaussian kernel to estimate the nonlinear mapping. For recovering latent components in Theorem 3.3), we apply the Spearman Mean Correlation Coefficient (MCC). We use the Structural Hamming Distance (SHD) to evaluate similarity of learned latent and observed causal structure. Specifically, considering the indeterminacy of the permutation of identified latent variables, we align the instantaneous latent causal structure $\mathbf{J}_r(\hat{\mathbf{z}}_t)$ and time-lagged latent causal structure $\mathbf{J}_r(\hat{\mathbf{z}}_{t-1})$ by permuting the learned adjacency matrices to match the ground truth. As a surrogate metric to learn observed causal DAG, we evaluated recovery of $\mathbf{s}_t$ using unpermuted MCC, corresponding to the identification strategy shown in Theorem 3.8. The recovered observed and latent causal DAG are further evaluated using SHD, divided by the number of possible structures. Based on such evaluation process, we also report TPR (recall), precision, and F1 among the comparisons with constraint-based methods.

**Simulation process** As defined in Def 2.2, under the **Independent** setting for the latent temporal process and dependent noise variable $\mathbf{s}_t$, we use the generation process from (Yao et al., 2022). For observational causal relations, we randomly generate lower triangular matrices and apply equal row and column permutations to obtain a mixing structure, which is combined with an MLP network generated from $\mathbf{z}_t$ and $\mathbf{s}_t$ using LeakyReLU units. This simulation is also used for **Sparse** and **Dense** settings, controlling graph degree after removing diagonals. Each independent noise is sampled from normal distributions.

**Baselines implementation details.** We utilized publicly available implementations for TDRL, CaRiNG, and iCRITIS, covering the majority of the employed methods. As G-CaRL's code was not released, we re-implemented it based on the details provided in the original paper. Additionally, since the iCRITIS setup was originally designed for image inputs, we adapted it by replacing its encoder and decoder components with a Variational Autoencoder with same hyperparameters in NCDL.

**Mask by inductive bias.** Continuous optimization faces challenges like local minima (Ng et al., 2022; Maddison et al., 2017), making it difficult to scale to higher dimensions. However, incorporating prior knowledge on the low probability of certain dependencies (Spirtes et al., 2001; Runge et al., 2019) enables us to compute a mask. To validate this approach using physical laws as observed DAG initialization  F.2 in climate data, we mask $\frac{3}{4}$ of the lower triangular elements in a simulation with $d_x = 100$, a ratio much lower than in real-world applications.

**Comparison with constraint-based methods.** Constraint-based methods rely on Conditional Independence (CI) tests and do not require a specified form of structural equation models (SEMs), making them a nonparametric approach. However, CI test-based methods generally return equivalence classes of graphs rather than a unique solution. For example, algorithms such as FCI produce

Partial Ancestral Graphs (PAGs), and CD-NOD similarly yields equivalence classes. In our implementation, we utilize the `Causal-learn` package (Zheng et al., 2024) for FCI and CD-NOD and the `Tigramite` library (Runge et al., 2019) for PCMCI and LPCMCI. We employ a near-optimal configuration for these methods to facilitate fair comparisons. The details for each method are as follows:

(i) **FCI**: We use Fisher's Z conditional independence test. For the obtained PAG, we enumerate all possible adjacency matrices and select the one closest to the ground truth by minimizing the Structural Hamming Distance (SHD).

(ii) **CD-NOD**: We concatenate the time indices $[1, 2, \ldots, T]$ of the simulated data into the observed variables and only consider the edges that exclude the time index. We use kernel-based CI test since it demonstrates superior performance here. We consider all obtained equivalence classes and select the result that minimizes SHD relative to the ground truth.

(iii) **PCMCI**: We use partial correlation as the metric of conditional independent test. We enforce no time-lagged relationships in PCMCI and run it to focus exclusively on contemporaneous (instantaneous) causal relationships. In the `Tigramite` library, this can be achieved by setting the maximum time lag $\tau_{\max}$ to zero. This effectively disables the search for lagged causal dependencies. We select contemporary relationships as the ultimate result.

(iv) **LPCMCI**: Similarly to PCMCI, we use partial correlation as the metric of conditional independent test, and select the contemporary relationships as the causal graph obtained.

**Study on dimension of latent variables.** We fix $d_x = 6$ and vary $d_z = \{2, 3, 4\}$ as shown in Table 5. The results indicate that both the Markov network and time-lagged structure are identifiable for lower dimensions. However, as the latent dimension increases, there is a decline in identifiability of the latent structure, highlighting ongoing challenges in the continuous optimization of latent process identification (Zhang et al., 2024; Li et al., 2024). Nevertheless, monoblock identifiability ($R^2$) remains satisfied across all settings.

| $d_x$ | $d_z$ | SHD ($\mathcal{G}_{x_t}$) | TPR | Precision | MCC ($\mathbf{s}_t$) | MCC ($\mathbf{z}_t$) | SHD ($\mathcal{G}_{z_t}$) | SHD ($\mathcal{M}_{lag}$) | $R^2$ |
|---|---|---|---|---|---|---|---|---|---|
| | 2 | 0.12 ($\pm$0.04) | 0.86 ($\pm$0.02) | 0.85 ($\pm$0.04) | 0.9864 ($\pm$0.01) | 0.9741 ($\pm$0.03) | 0.15 ($\pm$0.03) | 0.21 ($\pm$0.05) | 0.95 ($\pm$0.01) |
| 6 | 3 | 0.18 ($\pm$0.06) | 0.83 ($\pm$0.02) | 0.80 ($\pm$0.04) | 0.9583 ($\pm$0.02) | 0.9505 ($\pm$0.01) | 0.24 ($\pm$0.06) | 0.33 ($\pm$0.09) | 0.92 ($\pm$0.01) |
| | 4 | 0.23 ($\pm$0.02) | 0.80 ($\pm$0.06) | 0.74 ($\pm$0.01) | 0.9041 ($\pm$0.02) | 0.8931 ($\pm$0.03) | 0.33 ($\pm$0.03) | 0.48 ($\pm$0.05) | 0.91 ($\pm$0.02) |

Table 5: **Results on different latent dimensions.** We run simulations with 5 random seeds, selected based on the best-converged results to avoid local minima.

**Assumption ablation study.** We further validate our identifiability theory using $d_z = 3$ and $d_x = 6$. In simulating data, we remove conditionsthat are nontrivial to our theories, including:

(i) **A** (Def. 2.1): Ensuring $\mathbf{z}_t$ conditionally independent and replacing the transition function with an orthogonal matrix, violating the 3-measurement Hu & Schennach (2008).

(ii) **B** (Assumption (ii)): Violating the injectivity of linear operators using $g_s$: $\mathbf{z}_t = \mathbf{z}_{t-1} + \boldsymbol{\epsilon}_{z_t}$, where $\boldsymbol{\epsilon}_{z_t} \sim$ Uniform$(0, 1)$ is a typical violation Mattner (1993) for injectivity of $L_{z_t|z_{t-1}}$, as well as $L_{x_{t-1}|x_{t+1}}$.

(iii) **C** (Assumption 3.8): Violating the generation variability assumption with $\mathbf{s}_t = q(\mathbf{z}_t) + \boldsymbol{\epsilon}_{x_t}$, where $\boldsymbol{\epsilon}_{x_t} \sim \mathcal{N}(0, \mathbf{I}_{d_x})$ and $q$ is a mixing process, results in a linear additive Gaussian model without heteroscedasticity. Such a setup significantly reduces variability, as discussed in Yao et al. (2022)..

| Setting | MCC ($\mathbf{s}_t$) | $R^2$ |
|---|---|---|
| **A** | 0.6328 | 0.34 |
| **B** | 0.7563 | 0.67 |
| **C** | 0.7052 | 0.85 |

Table 6: **Assumption ablation study.** We mainly present these result to verify the necessity of our assumptions.

As shown in Table 6, without these assumptions, we cannot achieve monoblock identifiability or identify the observed causal DAG, leading to a decrease in $R^2$ and MCC ($\mathbf{s}_t$). In summary, these

results further confirm the validity of our theory and the necessity of these assumptions, especially under challenging conditions where they are disrupted.

**Hyperparameter sensitivity** We also test the hyperparameter sensitivity of NCDL with respect to the sparsity and DAG penalty, as these hyperparameters have a significant influence on the performance of structure learning. In this experiment, we set $d_z = 3$ and $d_z = 6$. As shown in Fig. 7, the results demonstrate robustness across different settings, although the performance of structure learning is particularly sensitive to the sparsity constraint. Notably, an excessively large DAG penalty at the beginning of training can result in a loss explosion or the failure of convergence.

| $\alpha$ | $1 \times 10^{-5}$ | $5 \times 10^{-5}$ | $1 \times 10^{-4}$ | $5 \times 10^{-4}$ | $1 \times 10^{-3}$ | $1 \times 10^{-2}$ |
|---|---|---|---|---|---|---|
| SHD | 0.23 | 0.22 | 0.18 | 0.27 | 0.32 | 0.67 |
| $\beta$ | $1 \times 10^{-5}$ | $5 \times 10^{-5}$ | $1 \times 10^{-4}$ | $5 \times 10^{-4}$ | $1 \times 10^{-3}$ | $1 \times 10^{-2}$ |
| SHD | 0.37 | 0.18 | 0.20 | / | / | / |

Table 7: **Hyperparameter sensitivity.** We run experiments using 5 different random seeds for data generation and estimation procedures, reporting the average performance on evaluation metrics. "/" means loss explosion.

## F.2 ON REAL-WORLD DATASET

**CESM2 Pacific SST.** CESM2 dataset employs monthly Sea Surface Temperature (SST) data generated from a 500-year pre-2020 control run of the CESM2 climate model. The dataset is restricted to oceanic regions, excluding all land areas, and retains its native gridded structure to preserve spatial correlations. It encompasses 6000 temporal steps, representing monthly SST values over the designated period. Spatially, the dataset comprises a grid with 186 latitude points and 151 longitude points, resulting in 28086 spatial variables, including 3337 land points where SST is undefined, and 24749 valid SST observations. To accommodate computational constraints, a downsampled version of the data, reduced to 84 grid points ($6 \times 14$), is utilized. This subset is specifically chosen to facilitate the investigation of oceanic temperature dynamics and underlying climate mechanisms.

**WeatherBench (Rasp et al., 2020).** WeatherBench is a benchmark dataset specifically tailored for data-driven weather forecasting. We specifically selected wind direction data for visualization comparisons within the same time period, maintaining the original 350,640 timestamps. Wind system is considered as the dominating factor resulting in potential instantaneous causal relationships among the temperature in different regions.

**Initialization observed DAG.** We incorporate the Spatial Autoregressive (SAR) model as a prior in the continuous optimization of the causal DAG structure matrix $\hat{G}_{x_t}$ to mitigate local minima and improve optimization stability, convergence rate, and computational complexity. The SAR model, commonly used in geography, economics, and environmental science, captures spatial dependencies defined as $\mathbf{X} = \mathbf{Z}\beta + \lambda\mathbf{W}\mathbf{X} + \mathbf{E}$, where $\mathbf{W}$ is the spatial weights matrix, and $\mathbf{E}$ is a disturbance term. Setting $\beta = 0$ results in a pure SAR model:

$$\mathbf{X} = \lambda\mathbf{W}\mathbf{X} + \mathbf{E}.$$

We define $\mathbf{W}$ based on Euclidean distances constrained by $\mathcal{M}_{loc}$, where $[\mathcal{M}_{loc}]_{i,j} = 1\{\|s_2 - s_1\|_2 \leq 50\}$, with $s_1$ and $s_2$ representing the locations of two regions. The rationale is that regions cannot be instantaneously connected if they are separated by a large physical distance. This configuration captures potential instantaneous causal effects only between spatially adjacent regions within a specified distance threshold of 50 units.

Linear regression coefficients capture inter-variable relationships and are used in causality research to construct an initial causal structure, reducing the search space. We build a sparse adjacency matrix by regressing $\mathbf{x}_{1:T,i}$ on $\mathbf{b} \cdot [\mathcal{M}_{loc}]_i \cdot \mathbf{x}_{1:T,[d_x]\backslash i}$, where $\mathbf{b} \cdot [\mathcal{M}_{loc}]_i$ represents the regressed coefficients, corresponding to the off-diagonal elements of $i$-th row of the initial matrix $\mathcal{M}_{init}$.

**Computation of the observed causal DAG in climate system.** In addition to using the mask gradient-based method for mask estimation, we compute the causal DAG by multiplying the mask with the Jacobian matrix derived from 6 data points to explicitly capture the dynamic causal mechanisms. The causal structure is defined as:

$$\hat{\mathcal{G}}_{x_t} = \mathrm{supp}(\hat{\mathcal{M}}_{x_t} * \mathbf{J}_{\hat{g}}(\mathbf{x}_t) * \mathcal{M}_{init})$$

A threshold of 0.15 is applied to obtain the final binary adjacency matrix. To compute the partial Jacobian $\mathbf{J}_{\hat{g}}(\mathbf{x}_t)$ with respect to $\mathbf{s}_t$ while holding $\mathbf{z}_t$ constant, set `requires_grad=False` for $\mathbf{z}_t$, and use `autograd.functional.jacobian` in PyTorch.

**Implications of not including time-lag effects in the observed space.** In this paper, we assume time-lagged effects are fully captured by the latent variables, as the temporal resolution of CESM2 data is relatively coarse with **1 month** interval. The temperature interactions by the wind system (observed causal DAG) occurs over a relatively short timescale, unlike the continuous and long-term processes of high-level latent variables (e.g., oceanic circulation patterns or gradual atmospheric pressure changes). Empirically, we also found time-delayed dependence (autocorrelation) in the CESM2 data is very small. Thus, we prefer to interpret causal effects through wind system as being completed within each time step.

**Real-world data experiments.** Our analysis yields two main results: *(i)* temperature forecasting, which demonstrates the effectiveness of the learned representations, and *(ii)* visualization of the inferred causal graph across regions, validated against contemporaneous wind patterns. As summarized in Table 8, our approach surpasses existing time-series forecasting models in precision, due to existing temporal causal representation learning cannot handle cases where observations are causally-related and the generating function is non-invertible, restricting their usability in real-world climate data. Computational cost of NCDL with other methods in the experiments can be seen in paragraph. 8. To compare against the estimated observed causal DAG, we use wind data from (Rasp et al., 2020) for the same period. In Fig. 7, the inferred causal structures closely correspond to actual wind patterns over the sea surface, accurately capturing the overall spatial dynamics and corroborating prior findings. Moreover, regions near coastlines exhibit denser causal connections, suggesting potential influences from anthropogenic activities or topographic features, thereby enriching our understanding of the underlying mechanisms governing the climate system.

|  |  | NCDL (Ours) | | TDRL | | CARD | | FITS | | MICN | | iTransformer | | TimesNet | | Autoformer | |
| Dataset | Len | MSE | MAE | MSE | MAE | MSE | MAE | MSE | MAE | MSE | MAE | MSE | MAE | MSE | MAE | MSE | MAE |
|---|---|---|---|---|---|---|---|---|---|---|---|---|---|---|---|---|---|
| CESM2 | 96 | 0.410 | 0.483 | 0.439 | 0.507 | **0.409** | 0.484 | 0.439 | 0.508 | 0.417 | 0.486 | 0.422 | 0.491 | 0.415 | 0.486 | 0.959 | 0.735 |
| CESM2 | 192 | **0.412** | 0.487 | 0.440 | 0.508 | 0.422 | 0.493 | 0.447 | 0.515 | 1.559 | 0.984 | 0.425 | 0.495 | 0.417 | 0.497 | 1.574 | 0.972 |
| CESM2 | 336 | **0.413** | 0.485 | 0.441 | 0.505 | 0.421 | 0.497 | 0.482 | 0.536 | 2.091 | 1.173 | 0.426 | 0.494 | 0.423 | 0.499 | 1.845 | 1.078 |

Table 8: **The MSE and MAE results for different prediction lengths in temperature forecasting.** Lower values indicate better forecasting performance. Bold numbers represent the best performance among the models, while underlined numbers denote the second-best performance.

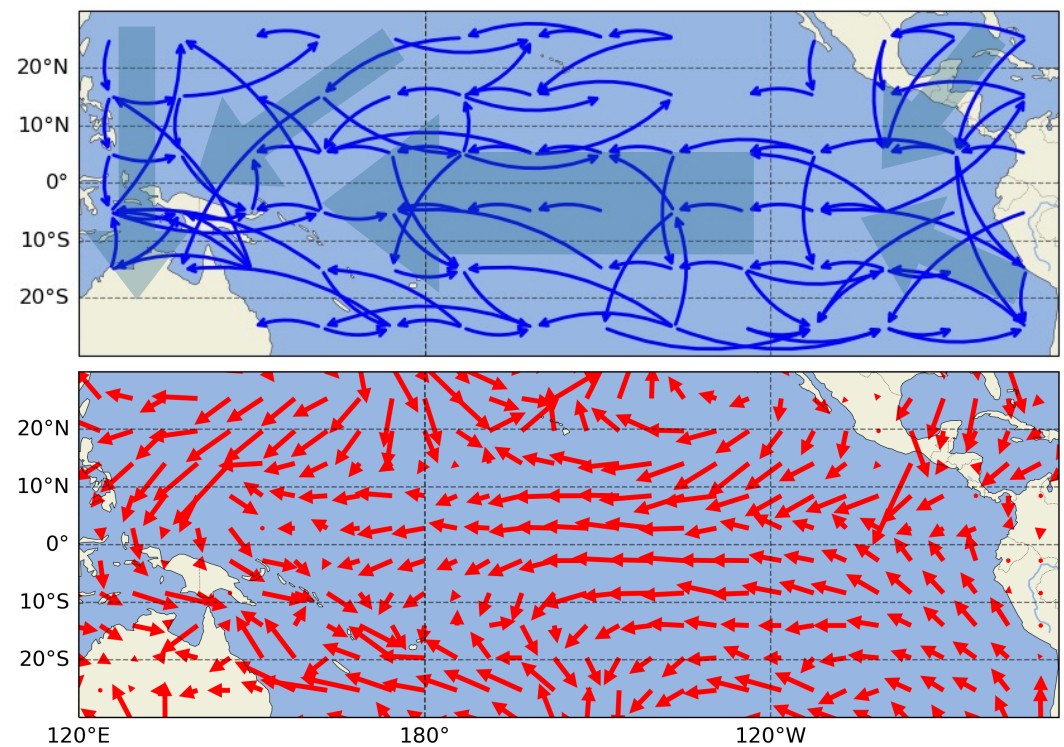

Figure 7: **Comparison of observed causal DAG obtained by NCDL and wind system.** Top: Visualization of learned instantaneous causal graph. Bottom: Visualization of the wind system. The blue arrows indicate the causal adjacency, while the red arrows represent the wind direction in the respective area. Notably, NCDL effectively identifies the underlying causal graph, showing a high degree of overlap with the real-world system. For instance, it captures the westward trend in the central sea region. However, in the sea/land interaction zones, the learned causal edges appear disorganized, losing clear patterns and becoming much denser than in other regions. Our result suggests that causal relationships are more intricate in these areas beyond wind system, likely due to the influence of human activities Vautard et al. (2019) and other factors arising from soil-atmosphere/cloud-temperature interactions, and land–sea warming contrasts Boé & Terray (2014).

**Runtime and computational efficiency.** We report the computational cost of the different methods considered. The comparison considers metrics including training time, memory usage, and corresponding performance MSE in forecasting task. Note that inference time is not included in the comparison, as our work focuses on causal structure learning through continuous optimization rather than constraint-based methods.

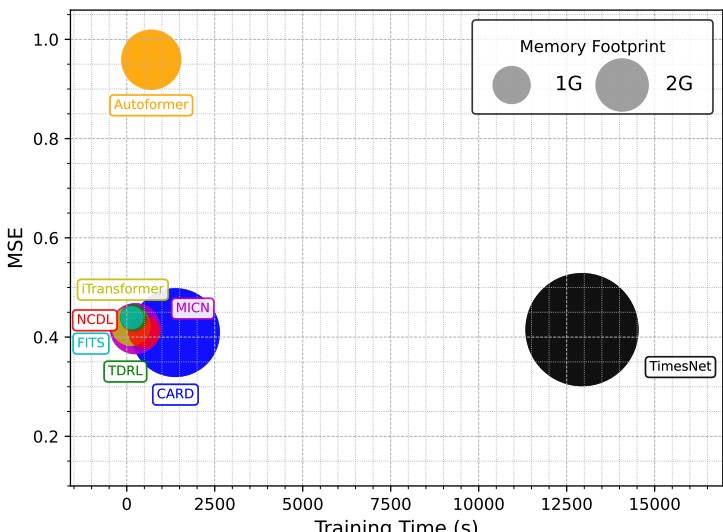

Figure 8: **Comparison of computational cost.** Different colors represent different methods, while the size of the circles corresponds to memory usage. The prediction length is set to 96.

Fig. 8 shows that our NCDL method simultaneously learns the causal structure while achieving the lowest MSE, highlighting the importance of building a transparent and interpretable model. Furthermore, NCDL exhibits similar training time and memory usage compared to mainstream time-series forecasting models in the lightweight track.

# G    DISCUSSIONS OF ALLOWING TIME-LAGGED CAUSAL RELATIONSHIPS IN OBSERVED SPACE

In this section, we demonstrate that our proposed framework is compatible with the consideration of time-lagged effects, with providing **potential solutions**.

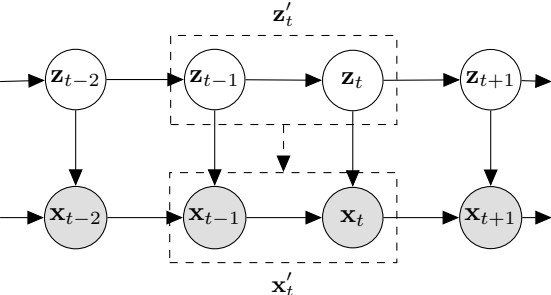

Figure 9: **4-measurement model with time-lagged effects in observed space.** $\mathbf{x}_t$ could be considered as the directed (dominating) measurement of $\mathbf{z}_t$, and $\mathbf{x}_{t-2}$, $\mathbf{x}_{t-1}$ and $\mathbf{x}_{t+1}$ provide indirect measurements of $\mathbf{z}_t$. For identifying the time-lagged causal relationships in observed space, we consider $\mathbf{z}_t' = (\mathbf{z}_{t-1}, \mathbf{z}_t)$ as the new latent variables, and $\mathbf{x}_t' = (\mathbf{x}_{t-1}, \mathbf{x}_t)$ as the new observed variables, to apply our *functional equivalence* b).

## G.1 PHASE I: IDENTIFYING LATENT VARIABLES FROM TIME-LAGGED CAUSALLY-RELATED OBSERVATIONS

For the identification of latent variables, we adopt the strategy outlined in (Carroll et al., 2010; Hu & Shum, 2012) to construct an spectral decomposition. We extend this approach to develop a proof strategy that establishes monoblock identifiability, as stated in Theorem 3.2.

We begin by defining the 4-measurement model, which includes time-series data with time-lagged effects in the observed space as a special case.

**Definition G.1 (4-Measurement Model)** $\mathbf{Z} = \{\mathbf{z}_{t-2}, \mathbf{z}_{t-1}, \mathbf{z}_t, \mathbf{z}_{t+1}\}$ *represents latent variables in four continuous time steps, respectively. Similarly,* $\mathbf{X} = \{\mathbf{x}_{t-2}, \mathbf{x}_{t-1}, \mathbf{x}_t, \mathbf{x}_{t+1}\}$ *are observed variables that directly measure* $\mathbf{z}_{t-2}, \mathbf{z}_{t-1}, \mathbf{z}_t, \mathbf{z}_{t+1}$ *using the same generating functions* g. *The model is defined by the following properties:*

- *The transformation within* $\mathbf{z}_{t-2}, \mathbf{z}_{t-1}, \mathbf{z}_t, \mathbf{z}_{t+1}$ *is not measure-preserving.*

- *Joint density of* $\mathbf{x}_{t-2}, \mathbf{x}_{t-1}, \mathbf{x}_t, \mathbf{x}_{t+1}, \mathbf{z}_t$ *is a product measure w.r.t. the Lebesgue measure on* $\mathcal{X}_{t-2} \times \mathcal{X}_{t-1} \times \mathcal{X}_t \times \mathcal{X}_{t+1} \times \mathcal{Z}_t$ *and a dominating measure* μ *is defined on* $\mathcal{Z}_t$.

- ***Limited feedback***: $p(\mathbf{x}_t \mid \mathbf{x}_{t-1}, \mathbf{z}_t, \mathbf{z}_{t-1}) = p(\mathbf{x}_t \mid \mathbf{x}_{t-1}, \mathbf{z}_t)$.

- *The distribution over* $(\mathbf{X}, \mathbf{Z})$ *is Markov and faithful to a directed acyclic graph (DAG).*

Limited feedback explicitly assumes that future events do not cause past events and excludes instantaneous effects from $\mathbf{x}_t$ to $\mathbf{z}_t$. As illustrated in Fig. 9, $\mathbf{x}_{t-2}, \mathbf{x}_{t-1}, \mathbf{x}_t, \mathbf{x}_{t+1}$ are defined as different measurements of $\mathbf{z}_t$, forming a temporal structure characteristic of a typical 4-measurement model. Under the data-generating process depicted in Fig. 9, and based on the assumption of limited feedback, we propose the following framework:

$$
\begin{aligned}
p(\mathbf{x}_{t-1}, \mathbf{x}_t, \mathbf{x}_{t+1}, \mathbf{x}_{t+2}) &= \int_{\mathcal{Z}_t} p(\mathbf{x}_{t+1} \mid \mathbf{x}_t, \mathbf{z}_t) p(\mathbf{x}_t \mid \mathbf{x}_{t-1}, \mathbf{z}_t) p(\mathbf{x}_{t-1}, \mathbf{x}_{t-2}, \mathbf{z}_t) dz_t \\
&= \int_{\mathcal{Z}_t} p(\mathbf{x}_{t+1} \mid \mathbf{x}_t, \mathbf{z}_t) p(\mathbf{x}_t, \mathbf{x}_{t-1}, \mathbf{z}_t) p(\mathbf{x}_{t-2} \mid \mathbf{z}_t, \mathbf{x}_{t-1}) dz_t.
\end{aligned}
\tag{52}
$$

**Discussion of achieving monoblock identifiability.** Comparing Eq. 52 with Eq. 18, which represents the foundational result for proving monoblock identifiability under the 3-measurement model, we extend the identification strategy from (Carroll et al., 2010; Hu & Shum, 2012) to the 4-measurement model. This forms the critical step in our identification process. We adopt assumptions analogous to those in (Carroll et al., 2010; Hu & Shum, 2012) and Theorem 3.2, and suppose the followings:

- (i) The joint distribution of $(\mathbf{X}, \mathbf{Z})$ and their all marginal and conditional densities are bounded and continuous.
- (ii) The linear operators $L_{x_{t+1}|x_t,z_t}$ and $L_{x_{t-2},x_{t-1},x_t,x_{t+1},z_t}$ are injective for bounded function space.
- (iii) For all $\mathbf{z}_t, \mathbf{z}'_t \in \mathcal{Z}_t$ $(\mathbf{z}_t \neq \mathbf{z}'_t)$, the set $\{\mathbf{x}_t : p(\mathbf{x}_t|\mathbf{z}_t) \neq p(\mathbf{x}_t|\mathbf{z}'_t)\}$ has positive probability.

hold true. Similar to the proof of our monoblock identifiability (Theorem B.1), except for the conditional independence introduced by the temporal structure, the key assumptions include an injective linear operator to enable the recovery of the density function of latent variables and distinctive eigenvalues to prevent eigenvalue degeneracy. The primary difference is the property **limited feedback**, where we can adopt the strategy in (Carroll et al., 2010) to construct a unique spectral decomposition, where $(\mathbf{x}_{t-2}, \mathbf{x}_{t-1}, \mathbf{x}_t, \mathbf{x}_{t+1}, \mathbf{z}_t)$ correspond to $(X, S, Z, Y, X^*)$, respectively.

Following this, we apply the key steps of our identification process as detailed in the Appendix B.1. Ultimately, we can establish that the block $(\mathbf{z}_t, \mathbf{x}_t)$ is identifiable up to an invertible transformation:

$$(\hat{\mathbf{z}}_t, \hat{\mathbf{x}}_t) = h_{x,z}(\mathbf{z}_t, \mathbf{x}_t). \tag{53}$$

where $h_{x,z} : \mathbb{R}^{d_x+d_z} \to \mathbb{R}^{d_x+d_z}$ is a invertible function. Since the observation $\mathbf{x}_t$ is known and suppose $\hat{\mathbf{x}}_t = \mathbf{x}_t$, this relationship indeed represents an invertible transformation between $\hat{\mathbf{z}}_t$ and $\mathbf{z}_t$

as
$$\hat{\mathbf{z}}_t = h_z(\mathbf{z}_t). \tag{54}$$
With an additional assumption of a sparse latent Markov network, we achieve component-wise identifiability of the latent variables, as stated in Theorem B.4 in appendix, leveraging the proof strategies of (Zhang et al., 2024; Li et al., 2024). These results are stronger than those in (Carroll et al., 2010).

## G.2 Phase II: Identifying Time-Lagged Observed Causal DAG

**Unified modeling across neighboring time points.** In the presence of time-lagged effects in the observed space, such as $\mathbf{x}_{t-1} \to \mathbf{x}_t$, alongside the causal DAG within $\mathbf{x}_t$, as depicted in Figure 9, we show that by introducing an expanded set of latent variables $\mathbf{z}'_t = (\mathbf{z}_{t-1}, \mathbf{z}_t)$ and an expanded set of observed variables $\mathbf{x}'_t = (\mathbf{x}_{t-1}, \mathbf{x}_t)$, the property of functional equivalence is preserved. Moreover, identifiability continues to hold, and, broadly speaking, it becomes more accessible due to the incorporation of Granger causality principles in time-series data (Freeman, 1983), if we assume that future events cannot influence or cause past events.

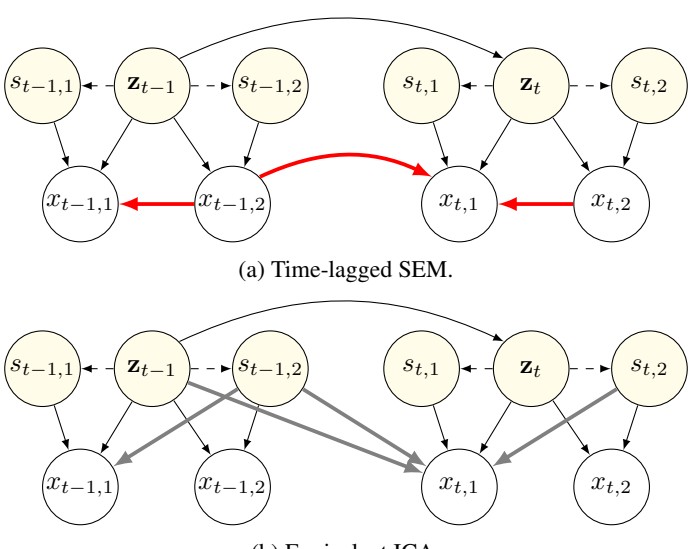

(a) Time-lagged SEM.

(b) Equivalent ICA.

Figure 10: **Equivalent time-lagged SEM and ICA in the case with time-lagged causal relationships in observed space.** The red lines in Fig. 10a indicate that information are transmitted by the instantaneous and the time-lagged observed causal DAGs, while the gray lines in Fig. 10b represent that the information transitions are equivalent to originating from contemporary $\mathbf{s}_t$ and previous $(\mathbf{z}_t, s_{t-1,2})$ within the mixing structure.

**Functional equivalence in presence of time-lagged effects.** As shown in Fig. 10, we show that, if we consider the time-lagged causal relationship in observed space, it still can be processed with the technique as in our paper proposed, through considering time-lagged causal relationships as a part of observed causal DAG, by reformulating $\mathbf{z}'_t = (\mathbf{z}_{t-1}, \mathbf{z}_t)$, $\mathbf{x}'_t = (\mathbf{x}_{t-1}, \mathbf{x}_t)$ and $\mathbf{s}'_t = (\mathbf{s}_{t-1}, \mathbf{s}_t)$, to apply the Theorem b). Specifically, the time-lagged effects from $\mathbf{x}_{t-2}$ can be considered as side information, which does not make difference to causal relationships from $\mathbf{x}_{t-1}$ to $\mathbf{x}_t$ and its corresponding ICA form.

## G.3 Estimation Methodology

**Slided window.** Building on the analysis above, we aggregate two adjacent time-indexed observations into a single new observation. By employing a sliding window with a step size of 1, we obtain $T - 1$ new observations along with their corresponding latent variables, thereby aligning with the estimation methodology described in Section 4.

**Structure pruning.** For structure learning, given the assumption that future climate cannot cause past climate, we can mask $\frac{1}{4}$ of elements in the causal adjacency matrix during implementation, as depicted in Fig. 11. Compared with the original implementation, the masking simplifies the difficulty of optimization by reducing the degrees of freedom in the graph.

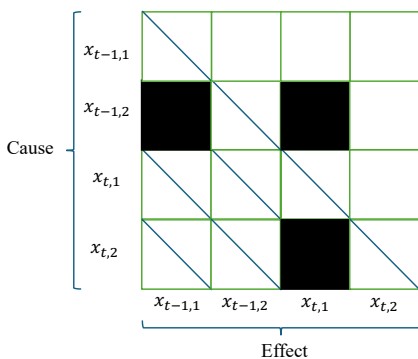

(a) Causal adjacency matrix of SEM.

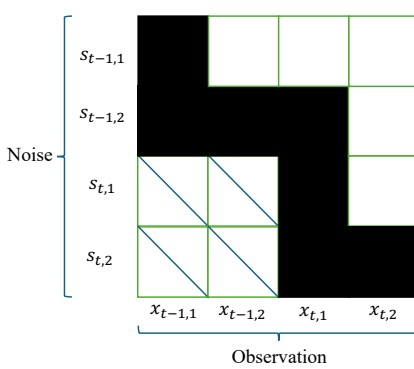

(b) Mixing matrix of equivalent ICA.

Figure 11: **Interpreting Fig. 10 with causal adjacency matrix of the SEM and the mixing matrix of the equivalent ICA.** The diagonal lines indicate masked elements, as future events cannot cause past events, and self-loops are not permitted. Black blocks represent the presence of a causal relationship or functional dependency in the generating function $g_m$, while white blocks indicate the absence of such a relationship.

