# OpenReview forum: "Identification of Nonparametric Dynamic Causal Model and Latent Process for Climate Analysis"
_ICLR.cc/2025/Conference — Submitted to ICLR 2025_

### Official Review · Reviewer_A5VH · 2024-10-31

**Soundness:** 3
**Presentation:** 3
**Contribution:** 2
**Rating:** 6
**Confidence:** 3

**Summary:**

The paper focuses on uncovering nonparametric dynamic causal structures and latent processes within complex systems like climate data, where both observed and unobserved (latent) variables interact through nonlinear relationships over time. The study introduces a comprehensive framework to identify and analyze these hidden causal processes under various assumptions, even in cases with nontrivial dynamics. The authors present a theoretical framework for identifying latent variables and causal structures within climate data, establishing conditions under which these variables can be detected when they induce significant variability. Through the novel Nonparametric Causal Discovery and Learning (NCDL) model, an extension of nonlinear Structural Equation Models (SEM) adapted for dynamic, time-based dependencies, they reframe SEM within a nonlinear Independent Component Analysis (ICA) structure to enhance causal identification. Validation on synthetic and CESM2 climate data demonstrates that NCDL outperforms traditional methods like FCI and PCMCI in accurately identifying latent variables and their relationships. This methodology, applied to sea surface temperatures and related climate variables, effectively identifies underlying factors such as CO2 levels and ocean currents, advancing insights into climate dynamics.

**Strengths:**

By introducing the Nonparametric Causal Discovery and Learning (NCDL) framework, the authors extend nonlinear Structural Equation Models (SEM) into a dynamic setting, embedding time-based dependencies in a way that transforms SEM into a nonlinear Independent Component Analysis (ICA) model. To the best of my knowledge, this approach is novel. Moreover, I also find the application of this framework to climate-specific variables, such as sea surface temperatures and CO₂ levels, very interesting. Both the theoretical and empirical contributions of this submission are sound. The authors provide identifiability results based on a set of assumptions, such as Functional Faithfulness. This theoretical contribution is complemented by comprehensive experimental validation across synthetic and real-world datasets, specifically with climate data from the Community Earth System Model Version 2 (CESM2). By benchmarking their method against existing approaches like Fast Causal Inference (FCI) and PCMCI and showing consistent improvements, the paper provides strong evidence for the superiority of their approach. Overall, this submission is overall well-written and easy to follow. Furthermore, I believe that this contribution is significant to the field of Causal Inference, due to the novelty of the proposed framework. Furthermore, the application to climate analysis is broadly relevant.

**Weaknesses:**

The main weakness of this work is the functional faithfulness assumption, which, to the best of my understanding, is essential to prove identifiability. Although the analogous concept of causal faithfulness has been extensively used in the literature, in my opinion these assumptions are restrictive. Furthermore, it is unclear why a dataset should exhibit functional faithfulness and how it can be verified from samples.

**Questions:**

How does functional faithfulness impact your results?
Why is it natural to assume functional faithfulness in this context?
Is is possible to verify this assumption from samples?
Can you discuss potential limitations or scenarios where functional faithfulness may not hold?
What is the relationship between functional faithfulness and other common assumptions in the related literature?

---

> ### Author Response · Authors · 2024-11-22
> **Response to Reviewer A5VH Part 1**
>
> We sincerely thank you for your thoughtful review and for raising insightful questions regarding faithfulness. Your feedback has led to a discussion of relevant assumptions.
>
> >**W1 & Q1:** The main weakness of this work is the functional faithfulness assumption, which, to the best of my understanding, is essential to prove identifiability. Although the analogous concept of causal faithfulness has been extensively used in the literature, in my opinion these assumptions are restrictive. Furthermore, it is unclear why a dataset should exhibit functional faithfulness and how it can be verified from samples.
>
> **A1:** Thank you for the question, which highlights a critical aspect of causal discovery. We have added the discussion in the manuscript to address these points (Page 27, Line 1424). Below, we provide the response to your comments:
>
> - >How does functional faithfulness impact your results?
>
> Functional faithfulness can be interpreted as the **edge minimality** assumption [1,2] for the Jacobian matrix $\mathbf{J} _ {g}(\mathbf{x}_t)$, which represents the nonlinear SEM $\mathbf{x} _ t = g(\mathbf{x} _ t, \mathbf{z} _ t, \boldsymbol{\epsilon} _ {x _ t})$. Under this assumption, $\frac{\partial x _ {t,j}}{\partial x _ {t,i}} = 0$ implies no causal edge, and $\frac{\partial x _ {t,j}}{\partial x _ {t,i}} \neq 0$ indicates causal relation $x _ {t,i} \rightarrow x _ {t,j}$. This assumption is fundamental to ensuring that Jacobian matrix reflects the true causal graph. If our functional faithfulness is violated, the results may be misleading. However, in theory, faithfulness (edge minimality) is generally achievable as discussed in [3]. If needed, violations of the (classical) faithfulness assumption (which is stronger than our assumption) can be testable except for the triangle faithfulness situation [2].
>
> To summarize, functional faithfulness is necessary to ensure that the Jacobian matrix accurately reflects causal relationships. If functional faithfulness is violated, the results may be misleading. However, this assumption is generally possible in most practical scenarios.
>
> - >Why is it natural to assume functional faithfulness in this context?
>
> We aim to recover information about the underlying causal process from observational data, and thus certain simplified assumptions, such as functional faithfulness, cannot be avoided, because going from the statistical level to the causal level requires additional assumptions. Thus, it is necessary to make an assumption like the functional faithfulness. Moreover, as a specific type of edge minimality [1,2], functional faithfulness assumption seems to be a weak one among its alternatives.
>
> - >Is it possible to verify this assumption from samples?
>
> Similar to most cases of (classical) faithfulness, functional faithfulness may not be directly testable, even with an infinite number of samples. However, theoretical work (e.g., [2]) demonstrates that violations of (classical) faithfulness assumption have zero measure, indicating they are highly unlikely in most practical scenarios. This property underpins that its subset, functional faithfulness, can serve as a reliable assumption in causal discovery.
>
> - >Can you discuss potential limitations or scenarios where functional faithfulness may not hold?
>
> If this assumption does not hold, the limitation arises in defining a causal relationship that produces no observable causal effects. For example, suppose there is a causal relationship $x _ {t,i} \rightarrow x _ {t,j}$ for which the causal coefficient happens to be zero. In the SEM of the form $x _ {t,j} = 0 \cdot x _ {t,i} + \epsilon_{x _ {t,i}}$, where the derivative $\frac{\partial x _ {t,j}}{\partial x _ {t,i}}$ remains zero across all samples. Then such a causal link cannot be recovered from data (some people may argue that this causal influence does not effectively exist).
>
> It is important to note that such a limitation does not compromise the correctness of the estimated data generating process; it only introduces ambiguity in the definition or understanding of causal relations.

---

> ### Author Response · Authors · 2024-11-22
> **Response to Reviewer A5VH Part 2**
>
> **Continue A1**
>
> - > What is the relationship between functional faithfulness and other common assumptions in the related literature?
>
>   **(Classical) Faithfulness**: This assumption ensures that no additional conditional independencies exist beyond those implied by the Markov condition of the true graph.
>   - In comparison, functional faithfulness is a much weaker assumption, focusing solely on ensuring that zero derivative does not correspond to an existing causal relation. Notably, (classical) faithfulness implies functional faithfulness but the reverse is not true (e.g., see problem 6.57 in [1]).
>   - Functional faithfulness does not exclude special cases, such as path cancellations in triangular structures, which are ruled out by (classical) faithfulness.
>   - Consequently, violations of functional faithfulness inherently lead to violations of (classical) faithfulness, but the reverse is not true.
>
>
>   **Rank Faithfulness [4]**: It requires that every rank constraint on a sub-covariance matrix that holds in probability is entailed by the causal relations.
>   - This assumption is primarily aimed at the identification of latent variables.
>   - Furthermore, (classical) faithfulness is a subset of rank faithfulness. Consequently, violations of functional faithfulness lead to violations of rank faithfulness, but not vice versa.
>
> Different forms of faithfulness and minimality have been discussed in [2].
>
> Hope our response makes the role of functional faithfulness clear and explicit. Please kindly let us know if you have further concerns; we would be delighted to discuss more.
>
> **References:**
>
> [1] Peters, Jonas, et al. *Elements of Causal Inference: Foundations and Learning Algorithms*. The MIT Press, 2017.
>
> [2] Zhang, Jiji. "A comparison of three Occam's razors for Markovian causal models." *The British Journal for the Philosophy of Science*, 2013.
>
> [3] Lemeire, Jan, and Dominik Janzing. "Replacing causal faithfulness with algorithmic independence of conditionals." _Minds and Machines_ 23 (2013): 227-249.
>
> [4] Huang, Biwei, et al. "Latent hierarchical causal structure discovery with rank constraints." _Advances in neural information processing systems_ 35 (2022): 5549-5561.

---

> ### Author Response · Authors · 2024-11-24
> **Should you have any further comments or additional suggestions, we would be more than happy to discuss them.**
>
> Dear Reviewer A5VH,
>
> We would like to express our sincere gratitude for the time and effort you invested in reviewing our submission. We greatly appreciate the insightful questions on the functional faithfulness assumption you provided. We have carefully considered your feedback and have made the response, which we hope address your concerns. Should you have any further comments or additional suggestions, we would be more than happy to discuss them.
>
> Thank you once again for your valuable input.
>
> Best regards,
>
> Authors of submission 7172

---

> > ### Comment · Reviewer_A5VH · 2024-11-26
> >
> > Thank you for your reply. After reading the rebuttal, my score remains unchanged.

---

> > > ### Author Response · Authors · 2024-11-26
> > > **We appreciate your effort in engaging with our work and your comments**
> > >
> > > Dear Reviewer A5VH,
> > >
> > > Thank you for taking the time to review our submission and for your thoughtful feedback. We appreciate your effort in engaging with our work and your comments, which have helped us to clarify and strengthen our paper during the rebuttal process.
> > >
> > > Thank you again for your time and consideration.
> > >
> > > Best regards,
> > >
> > > Authors of submission 7172

---

### Official Review · Reviewer_MCrH · 2024-11-01

**Soundness:** 3
**Presentation:** 2
**Contribution:** 3
**Rating:** 6
**Confidence:** 3

**Summary:**

The work introduces a nonparametric framework for identifying causal relationships in climate data, addressing the complex interactions between latent variables and observed data. It advances a methodology that integrates latent variable identification and causal inference in dynamic environments.

**Strengths:**

1.This work introduces a new framework of nonparametric dynamic causal models and, by extending the application of nonlinear independent component analysis (ICA), proposes a novel approach for identifying latent causal relationships in complex systems.

2.The NCDL framework in this paper demonstrates strong performance across various experimental settings, including applications in climate data, showcasing the model's adaptability and robustness.

3.The paper is well-structured, with clear divisions into theoretical analysis, model framework, and experimental validation, presenting a logical flow. However, given the study involves multiple complex concepts, such as equivalence transformations and nonlinear independent component analysis, certain theoretical derivations and symbol definitions may seem challenging for non-specialist readers. Some figures and formulas also lack sufficient explanation, which might affect readers' understanding.

**Weaknesses:**

1.The paper involves numerous symbols and matrix operations (e.g., in Definition 3.4 and Equation (5)), but some symbols lack clear explanations. It is recommended to define each symbol's meaning the first time it appears to avoid ambiguity.

2.Although the paper compares the performance of various existing methods, the experimental comparisons in Section 5 on constraint-based methods (e.g., FCI, CD-NOD) and temporal representation learning methods do not further explain why the NCDL method outperforms them.

3.Could you provide details on the hyperparameters $\alpha$ and $\beta$ used in your experiments' loss function? How were these hyperparameters chosen, and did you observe any impact on performance from varying these values?

4.Although Figure 6 shows the overall structure of the NCDL model, the module arrangement is complex and may be difficult to understand by visualization alone. It is recommended to provide a concise explanation of each module’s function in the caption or describe each module's specific role in the text, especially the interactions among the "Encoder," "Decoder," and "Prior Network."

5.Table 4 appears not to have been referenced or discussed in the main text.

6.The "Assumption Ablation Study" in Section 5.1 does not provide sufficient explanation of parameter choices, which may hinder the reproducibility of the experiments.

7.The conclusion mentions that future work could address performance degradation in high-dimensional data, but it does not provide specific directions or solutions. It is suggested to further discuss possible research paths that could be explored based on this issue.

8.In line 494, "Directed Acyclic Graph (DAG)" could be simplified to "DAG" since it was already defined in line 120.

**Questions:**

N/A

---

> ### Author Response · Authors · 2024-11-22
> **Response to Reviewer MCrH Part 1**
>
> We are grateful for your thoughtful review and for highlighting areas of confusion in the experiments presented in our earlier manuscript. In response, we have made substantial improvements, including:
>
> - **Explanation of Comparisons with Constraint-Based Methods**: To address the concerns regarding the reasons for outperformance, we have provided a detailed analysis including empirical observations and theoretical insights. (Page 9, Line 482)
> - **Clarification of Notations**: To address unclear notations, we have introduced a new notation table that provides explanations of each symbol. (Page 18, Line 921)
> - **Hyperparameter Selection**: In response to concerns about the hyperparameter selection, we have conducted an experiment on hyperparameter settings to justify our choices.  (Page 31, Line 1623)
> - **Enhanced Readability**: To improve the overall readability and understanding of the paper, we have provided clarifying illustrations throughout the manuscript.
> - **Future Work Discussion**: We have included a discussion on addressing performance degradation in high-dimensional data, offering potential solutions and directions for future research. (Page 10, Line 537)
> - **Additional Revisions**: Further refinements throughout this paper have been made to address all other comments.
>
> We hope these updates effectively address your concerns. Please find our point-by-point responses below.
>
> > **W1:** The paper involves numerous symbols and matrix operations (e.g., in Definition 3.4 and Equation (5)), but some symbols lack clear explanations. It is recommended to define each symbol's meaning the first time it appears to avoid ambiguity.
>
> **(1) A1:** Thank you for your careful review. In response to your feedback, we have added necessary notations/explanations at proper places and a notation table in the updated manuscript (Page 3, Line 145; Page 18, Line 921). For example:
>
> We have explicitly defined $\mathbf{J} _ {g _ m}(\mathbf{s} _ t)$ in Eq. 5 to the derivative of the function in the mixing process (ICA) from $\mathbf{s} _ t$ to $\mathbf{x} _ t$. We stated that $\mathbf{J} _ {g}(\mathbf{x} _ t)$ implies the causal adjacency in a nonlinear SEM, if the assumptions below hold true.", and
> Definition 3.4 has been updated to "Causal relations within observed variables are represented by the support set of Jacobian matrix $\mathbf{J} _ {g}(\mathbf{x} _ t)$."

---

> ### Author Response · Authors · 2024-11-22
> **Response to Reviewer MCrH Part 2**
>
> > **W2:** Although the paper compares the performance of various existing methods, the experimental comparisons in Section 5 on constraint-based methods (e.g., FCI, CD-NOD) and temporal representation learning methods do not further explain why the NCDL method outperforms them.
>
> **(2) A2:** Thank you for raising this important question. We explain the reasons for its outperformance point by point in terms of 1) experimental results and 2) theoretical discussions (**Bold numbers** indicate the best performance):
>
> First of all, the following table compares the structure learning results in terms of SHD, recall, precision, and F1, which suggest NCDL is empirically more reliable.
>
> | Dataset Size (x) | Method        | SHD    | Recall | Precision | F1     |
> |-------------------|---------------|--------|--------|-----------|--------|
> | $10^3$         | NCDL          | **0.38876** | **0.55**   | **0.61**      | **0.5784** |
> |                   | FCI           | 0.47876 | 0.3021 | 0.4321    | 0.3556 |
> |                   | CD-NOD        | 0.5256  | 0.348  | 0.381     | 0.3638 |
> |                   | PCMCI         | 0.594   | 0.37   | 0.38      | 0.3749 |
> |                   | LPCMCI        | 0.5862  | 0.432  | 0.515     | 0.4699 |
> | $10^{3.5}$     | NCDL          | **0.235**   | **0.70**   | **0.68**      | **0.6899** |
> |                   | FCI           | 0.4088  | 0.35   | 0.4631    | 0.3987 |
> |                   | CD-NOD        | 0.4687  | 0.38   | 0.401     | 0.3902 |
> |                   | PCMCI         | 0.565   | 0.3642 | 0.3942    | 0.3786 |
> |                   | LPCMCI        | 0.4865  | 0.542  | 0.5645    | 0.5530 |
> | $10^4$         | NCDL          | **0.22**    | **0.78**   | **0.7543**    | **0.7669** |
> |                   | FCI           | 0.3798  | 0.3631 | 0.4621    | 0.4067 |
> |                   | CD-NOD        | 0.4567  | 0.415  | 0.422     | 0.4185 |
> |                   | PCMCI         | 0.4654  | 0.38   | 0.4521    | 0.4129 |
> |                   | LPCMCI        | 0.3854  | 0.55   | 0.5923    | 0.5704 |
> | $10^{4.5}$      | NCDL          | **0.18**    | **0.8312** | **0.7821**    | **0.8059** |
> |                   | FCI           | 0.3835  | 0.3832 | 0.4832    | 0.4274 |
> |                   | CD-NOD        | 0.4542  | 0.4244 | 0.424     | 0.4242 |
> |                   | PCMCI         | 0.4243  | 0.3708 | 0.4908    | 0.4224 |
> |                   | LPCMCI        | 0.3314  | 0.613  | 0.62      | 0.6165 |
> | $10^5$        | NCDL          | **0.18**    | **0.8312** | **0.8012**    | **0.8159** |
> |                   | FCI           | 0.3722  | 0.40   | 0.49      | 0.4404 |
> |                   | CD-NOD        | 0.4542  | 0.43   | 0.439     | 0.4345 |
> |                   | PCMCI         | 0.4224  | 0.42   | 0.512     | 0.4615 |
> |                   | LPCMCI        | 0.3314  | 0.6124 | 0.6334    | 0.6228 |
>
> Below let us provide some theoretical discussion.
> 1. **FCI**: FCI relies exclusively on only conditional independence relations among the observed variables, which might not be informative enough in time-series data. Moreover, since FCI does not directly recover latent variables or model their dependence relations, when latent confounders are dependent on each other, it often suffers from low recall. As shown in the table, FCI performs poorly with a relatively low recall, reflecting these limitations in our setting.
>
> 2. **CD-NOD**: CD-NOD operates under the assumption of pseudo-causal sufficiency, which requires latent confounders to be functions of surrogate variables (e.g., time indices). This assumption significantly restricts its applicability in presence of general latent variables. As shown in the table, CD-NOD has the highest SHD, indicating that it cannot ensure correctness of the obtained causal structure.
>
> 3. **PCMCI**: PCMCI does not account for latent variables or underlying latent processes in time-series data, making it unsuitable for scenarios where causal sufficiency cannot be guaranteed. As demonstrated in this table, PCMCI performs poorly under our setting.
>
> 4. **LPCMCI**: Although LPCMCI considers latent variables, like FCI, it does not directly recover latent variables or model their dependence relations. As shown in the table, LPCMCI generally performs better than other constraint-based methods due to its ability to handle latent variables in time-series data and elegant implementation. However, its limitations prevent it from achieving better results.

---

> ### Author Response · Authors · 2024-11-22
> **Response to Reviewer MCrH Part 3**
>
> **continue (2)A2**
>
> Secondly, we provide the following table that compares the results of latent variable identification.
>
> | Setting | Metric | NCDL | iCITRIS | GCaRL | CaRiNG | TDRL | LEAP | SlowVAE | PCL | i-VAE | TCL |
> |--------------|--------|--------|---------|--------|--------|--------|--------|---------|--------|--------|--------|
> | **Independent** | MCC | **0.9811** | 0.6649 | 0.8023 | 0.8543 | 0.9106 | 0.8942 | 0.4312 | 0.6507 | 0.6738 | 0.5916 |
> | | $R^2$ | **0.9626** | 0.7341 | 0.9012 | 0.8355 | 0.8649 | 0.7795 | 0.4270 | 0.4528 | 0.5917 | 0.3516 |
> | **Sparse** | MCC | **0.9306** | 0.4531 | 0.7701 | 0.4924 | 0.6628 | 0.6453 | 0.3675 | 0.5275 | 0.4561 | 0.2629 |
> | | $R^2$ | **0.9102** | 0.6326 | 0.5443 | 0.2897 | 0.6953 | 0.4637 | 0.2781 | 0.1852 | 0.2119 | 0.3028 |
> | **Dense** | MCC | **0.6750** | 0.3274 | 0.6714 | 0.4893 | 0.3547 | 0.5842 | 0.1196 | 0.3865 | 0.2647 | 0.1324 |
> | | $R^2$ | **0.9204** | 0.6875 | 0.8032 | 0.4925 | 0.7809 | 0.7723 | 0.5485 | 0.6302 | 0.1525 | 0.206 |
>
> **Temporal (Causal) Representation Learning**: Existing methods in this category are unable to address scenarios where causal links exist within observations, or the generating function from latent variables to observations is non-invertible. These limitations restrict their usability in real-world data. As shown in the table above, metrics $R^2$ and MCC indicate that these methods cannot recover the latent variables when observations are instantaneously causally-related.
>
> Following your valuable suggestions, we provide these analyses in the updated manuscript (Page 9, Line 482; Page 10, Line 505). We hope this explanation clarifies the distinctions.
>
> > **W3:** Could you provide details on the hyperparameters $\alpha$ and $\beta$ used in your experiments' loss function? How were these hyperparameters chosen, and did you observe any impact on performance from varying these values?
>
> **(3) A3:** Thank you for your question; we have included such details in the updated manuscript In our experiments, we set $\alpha = 1.0 \times 10^{-4}$ (sparsity penalty) and $\beta = 5.0 \times 10^{-5}$ (DAG penalty) for structure learning. For other hyperparameters, we adopt the settings of [1].
>
> We choose hyperparameters utilizing a binary search strategy from broad ranges $\alpha \in [1.0 \times 10^{-5}, 1.0 \times 10^{-1}]$ and $\beta \in [1.0 \times 10^{-5}, 1.0 \times 10^{-1}]$, while fixing the other to its current setting. Below, we provide experiments on hyperparameter selection based on the Structure Hamming Distance (SHD) metric, with setting $d_z = 3$ and $d_x = 6$.
>
> | **$\alpha$**       | $1 \times 10^{-5}$ | $5 \times 10^{-5}$ | $1 \times 10^{-4}$ | $5 \times 10^{-4}$ | $1 \times 10^{-3}$ | $1 \times 10^{-2}$ |
> |---------------------|--------------------|--------------------|--------------------|--------------------|--------------------|--------------------|
> | **SHD**            | 0.23              | 0.22              | 0.18              | 0.27              | 0.32              | 0.67              |
> | **$\beta$**        | $1 \times 10^{-5}$ | $5 \times 10^{-5}$ | $1 \times 10^{-4}$ | $5 \times 10^{-4}$ | $1 \times 10^{-3}$ | $1 \times 10^{-2}$ |
> | **SHD**            | 0.37              | 0.18              | 0.20              | /                 | /                 | /                 |
>
> The table shows that $\alpha = 1.0 \times 10^{-4}$ and $\beta = 5.0 \times 10^{-5}$ yield the best performance. We found that the results of structure learning are sensitive to the $\alpha$ and $\beta$ values, as an inappropriate penalty can lead to a local minima. Notably, a large DAG penalty at the beginning of training may result in a loss explosion ("/" in the table above). We have provided an analysis of hyperparameter tuning and its effects in the updated manuscript (Page 31, Line 1623). We hope this explanation addresses your question clearly.

---

> ### Author Response · Authors · 2024-11-22
> **Response to Reviewer MCrH Part 4**
>
> > **W4:** Although Figure 6 shows the overall structure of the NCDL model, the module arrangement is complex and may be difficult to understand by visualization alone. It is recommended to provide a concise explanation of each module’s function in the caption or describe each module's specific role in the text, especially the interactions among the "Encoder," "Decoder," and "Prior Network."
>
>
> **(4) A4:** Thank you for your valuable suggestion. Accordingly, we have included a revised figure that presents an step-by-step explanation as follows:
>
> The model framework includes two encoders: **z-encoder** for extracting latent variables $\mathbf{z}_t$ and **s-encoder** for extracting $\mathbf{s} _ t$. A decoder reconstructs observations from these extracted variables. Additionally, prior networks estimate the prior distribution using normalizing flow, aiming at learning causal structure based on the Jacobian matrix. $L _ s$ imposes a sparsity constraint and $L _ d$ enforces the DAG structure on the Jacobian matrix. $\mathcal{L} _ {kl}$ enforces an independence constraint on the estimated noise by minimizing its KL divergence w.r.t. $\mathcal{N}(0, \mathbf{I})$.
>
> Detailed explanations for each module are provided in the revised manuscript (Page 7, Line 362).
>
> >**W5:** Table 4 appears not to have been referenced or discussed in the main text.
>
> **(5) A5:** Thank you for your careful review. Yes–the paragraph discussing Table 4 was deleted in the original submission. We have ensured that the original Table 4 is now properly referenced and discussed. Details of the change can be seen in Line 1708, Page 32.
>
> > **W6:** The "Assumption Ablation Study" in Section 5.1 does not provide sufficient explanation of parameter choices, which may hinder the reproducibility of the experiments.
>
> **(6) A6:** Thank you for your constructive feedback. Below is the explanation of each parameter choice in the Assumption Ablation Study, which is specifically designed to verify that the assumptions are essential by altering the data simulation method:
>
> - **A**: Ensures $\mathbf{x} _ {t-1}$, $\mathbf{x} _ t$, and  $\mathbf{x} _ {t+1}$ are not conditional independent given $\mathbf{z} _ t$ by replacing the original transition function with an orthogonal transformation, thereby violating the requirement that $\mathbf{z} _ {t}$ has 3 different measurements (Definition 2.1).
>
> - **B**: Makes observations insufficient to recover the latent variables, thereby violating the assumption that "The linear operators $L _ {x _ {t+1} \mid z_t}$ and $L _ {x _ {t-1} \mid x _ {t+1}}$ are injective for bounded function space." (Assumption (ii) in Theorem 3.2), since “adding an uniform noise” typically leads to an non-injective operator.
>
> - **C**: Uses a linear additive Gaussian model without heteroscedasticity to generate $\mathbf{s} _ t$, which violates the "Generation Variability" assumption (Assumption 3.8). This is a typical violation of the variability assumption, as illustrated in [1].
>
> To support reproducibility, we have also provided a part of the implementation for each setting below:
>
> ```
> import numpy as np
> from scipy.stats import ortho_group
> latent_size = 3
> obs_dim = 6
> zt_1 = np.random.randn(latent_size)
> if setting == "A":
>     A = ortho_group.rvs(latent_size)
>     zt = A @ zt_1 # zt = zt-1 + e # measure-preserving process, violate 3-measurement
> elif setting == "B":
>     epsilon_z_t = np.random.uniform(0, 1)
>     zt = zt_1 + epsilon_z_t # zt = zt-1 + e, violate injectivity assumption
> else:
>     zt = ....
> if setting == "C":
>     mixing_mat = np.random.randn(obs_dim, latent_size)
>     epsilon_xt = np.random.multivariate_normal(mean=np.zeros(obs_dim), cov=np.eye(obs_dim))
>     st = mixing_mat @ z_t + epsilon_xt # st = zt + e, violate variability assumption in nonlinear ICA
> else:
>     st = ...
> ```
> Following your question, we have added these explanations in the updated manuscript (Page 30, Line 1604).

---

> ### Author Response · Authors · 2024-11-22
> **Response to Reviewer MCrH Part 5**
>
> >**W7:** The conclusion mentions that future work could address performance degradation in high-dimensional data, but it does not provide specific directions or solutions. It is suggested to further discuss possible research paths that could be explored based on this issue.
>
> **(7) A7:** Thank you for your feedback. Performance degradation with increasing dimensionality is a well-known challenge, also observed in constraint-based methods such as the PC algorithm. We have discussed potential directions to address this issue in the revised manuscript (Page 10, Line 537).
>
> - **Divide-and-Conquer Strategy**: This approach decomposes the high-dimensional problem into smaller, overlapping subsets of variables, thereby reducing the dimensionality and enabling more efficient processing.
>
> - **Leveraging Geographical Information**: For climate data, incorporating geographical information can help eliminate less likely causal edges (for instance, between stations that are far from each other), thereby reducing the search space and generally improving the accuracy of structure learning.
>
>
>
> > **W8:** In line 494, "Directed Acyclic Graph (DAG)" could be simplified to "DAG" since it was already defined in line 120.
>
> **(8) A8:** Thank you for your careful review. It was updated accordingly (Page 32, Line 1712).
>
> **References:**
> [1] Yao, Weiran, Guangyi Chen, and Kun Zhang. "Temporally disentangled representation learning." Advances in Neural Information Processing Systems 35 (2022): 26492-26503.

---

> ### Author Response · Authors · 2024-11-24
> **Your feedback is vital to us, and any response would be further appreciated.**
>
> Dear Reviewer MCrH,
>
> We express our sincere gratitude for taking the time to review our manuscript. Your suggestions regarding the experiments and readability of our article have greatly contributed to improving its quality. We have made detailed revisions to the manuscript and addressed your questions in our response. We hope that our answers have addressed any concerns you had regarding our work. Your feedback is vital to us, and any response would be further appreciated.
>
> Many thanks,
>
> Authors of submission 7172

---

> ### Author Response · Authors · 2024-12-02
> **Gentle Reminder**
>
> Dear Reviewer MCrH,
>
> We sincerely appreciate your time and valuable feedback. With the discussion period ending soon, we hope our responses address your concerns. We understand your busy schedule, but would greatly appreciate it if you could consider our updates when revising your rating and discussing with the AC and other reviewers.
>
> Thank you again for your thoughtful and constructive input!
>
> Authors of submission 7172

---

> ### Author Response · Authors · 2024-12-02
> **Concerns properly addressed?**
>
> Dear Reviewer MCrH,
>
> We sincerely appreciate the time and effort you have dedicated to reviewing our submission and providing valuable feedback. We hope we have adequately addressed your concerns before the discussion period concludes. If you have any additional comments, we would greatly value the opportunity to respond. Your feedback would be appreciated.
>
> Best regards,
>
> Authors of Submission 7172

---

### Official Review · Reviewer_4ZWg · 2024-11-02

**Soundness:** 3
**Presentation:** 1
**Contribution:** 3
**Rating:** 5
**Confidence:** 3

**Summary:**

In this paper, the authors consider a general setting where causal relations are nonparametric in  climate systems. Using  three measurements in temporal structure,  the paper  shows that both latent variables and processes can be identified up to minor indeterminacy. the authors  proved  that the observed causal structure is identifiable. They also develop a  very nice procedure, which can simultaneously learn both the causal structure and latent representation. They conduct an extensive experiment, which demonstrates the  usefulness of the proposed  method.

**Strengths:**

The paper proposed the innovative approach, which  offers a powerful and in-depth understanding of climate system. The paper established the theoretical results of the  proposed methodology, and therefore setted up the   solid foundations  in real-world scenarios, including climate systems. Real data analysis demonstrated the impact of the new methods.

**Weaknesses:**

The main concern to the paper is a presentation. In many places the meaning of sentences is vague.   It is very difficult to understand the meaning. It is better to elaborate it  and make sentence shorter.

**Questions:**

I have several comments and suggestions for the authors to address.


1.  It is of interest for the authors to compare the computational cost of proposed method with existing methods in the experiments. Some metrics of computational cost (e.g., runtime, memory usage) are considered  for comparison.

2.  On page 9, line 475 of Table 3, I am not clear to the  meaning of bold type of fonts.  It is helpful for the authors to  give an illustration. Consider explaining the meaning of the bold font in Table 3 to improve the clarity.

3.  The presentations are not clear to us. In  in real-world scenarios, such as those in climate system. In the paper,  many places  are confused to me.  There are exampes below for the authors to improve clarity.

4.  There are  many typos, grammatical errors, etc. spotted in the paper. Please proofread and check it carefully.

Page 2, line 060, "e.g." -> "e.g.,".

Page 2, line 066, "can" -> "to".

Page 2, line  071, "sparsity" -> "a sparsity".

Page 2, line 099,  ", thus" -> ". Thus".

Page 3, line 124, "Appx" -> "Appendix".

Page 3, line 136, "denotes" -> "denote".

Page 4, line 190,  add  ".".

Page 4, line  214, "e.g." -> "e.g.,".

Page 5, line 216,   hat is "if to".

Page 6, line 281, "proof" -> "a proof".

Page 7, line 365, "12" -> "(12)".

Page 7, line 377, add   ".".

---

> ### Author Response · Authors · 2024-11-22
> **Response to Reviewer 4ZWg**
>
> We sincerely appreciate the valuable reviews and constructive comments, which helped improve the quality of this paper. We have carefully reviewed the manuscript as per your suggestions, and incorporated a **notation table** to improve readability, a **new model figure** with captions to clearly illustrate the model architecture, and a **new set of experiments** concerning computation cost/runtime. Please find our responses to your questions outlined below.
>
> >**W1**: The main concern to the paper is a presentation. In many places the meaning of sentences is vague. It is very difficult to understand the meaning. It is better to elaborate it and make sentence shorter.
>
> **(1) A1:** Thank you for the time and effort you dedicated to reviewing our paper. Following your suggestions, we have made corresponding updates to the manuscript to enhance clarity. These updates include:
>
> - **Clarification of Notations**: We have addressed this concern by clarifying key notations early in the manuscript (Page 3, Line 149) and adding a notation table (Page 18, Line 921). Hope you find it an effective guideline for readers.
> - **Understandability of Theories**: Proof sketches and intuitive explanations have been provided for each theorem. (Page 5, Line 227; Page 6, Line 292; Page 7, Line 343).
> - **Model Illustration**: We have included a detailed figure of the model pipeline to indicate inputs, outputs, and the process of estimating variables and DAGs (Page 8, Line 378).
> - **Module-Specific Estimation**: Explanations for estimating latent variables, latent structures, and observed causal DAGs have been added (Page 8, Line 402; Page 8, Line 427).
> - **Experimental Results**: Detailed illustrations and thorough analyses of experimental results have been included (Page 10, Line 505; Page 32, Line 1707).
> - **Readability of Proofs**: Proof pipelines and milestones have been clarified, along with understandable illustrations for each proof (Page 18, Line 963).
> - **Additional Improvements**: Further refinements have been made throughout the manuscript to address other points of feedback.
> We hope these revisions address your concerns effectively.
>
> > **Q1:** It is of interest for the authors to compare the computational cost of the proposed method with existing methods in the experiments. Some metrics of computational cost (e.g., runtime, memory usage) are considered for comparison.
>
> **(2) A1:** Thank you for your suggestion. Here we provide a table for such a comparison, which demonstrates that our NCDL approach is computationally relatively efficient and fast, while also achieving competitive performance in temperature forecasting.
>
> | Metric | IDOL | TDRL | CARD | FITS | MICN | iTransformer | TimesNet | Autoformer |
> |---------------|---------|--------|----------|--------|--------|--------------|-------------|------------|
> | Time (s) | 492.97 | 194.31 | 1383.75 | 148.79 | 245.34 | 121.14 | 12935.38 | 696.98 |
> | GPU Memory (MB) | 712 | 502 | 5792 | 356 | 1866 | 1094 | 9474 | 2606 |
> | MSE | 0.410 | 0.439 | 0.409 | 0.439 | 0.417 | 0.422 | 0.415 | 0.959 |
>
> In light of your suggestions, we have included a comparison figure (which is the table here)  for runtime and computational efficiency in the updated manuscript (Page 33, Line 1778).
>
> > **Q2:** On page 9, line 475 of Table 3, I am not clear to the meaning of bold type of fonts. It is helpful for the authors to give an illustration. Consider explaining the meaning of the bold font in Table 3 to improve the clarity.
>
> **(3) A2:** Thanks for your question. The bold font is used to indicate the **best performance**, as higher MCC values (closer to 1) represent better results. We have provided such an explanation in the updated manuscript (Page 10, Line 523).
>
> > **Q3 & Q4:** The presentations are not clear to us. In real-world scenarios, such as those in the climate system. In the paper, many places are confusing to me. There are examples below for the authors to improve clarity.
> There are many typos, grammatical errors, etc. spotted in the paper. Please proofread and check it carefully...
>
> **(4) A3 & A4:** We appreciate your patience and help! We have carefully revised the whole paper and corrected the identified typos and errors, all the points you raised have been properly addressed in the updated manuscript.

---

> > ### Comment · Reviewer_4ZWg · 2024-11-27
> >
> > Thank you very much for  your reply. I would like to retain my score.

---

> > > ### Author Response · Authors · 2024-11-28
> > > **We sincerely appreciate your effort in engaging with our submission and raising points that have allowed us to further clarify and strengthen our paper**
> > >
> > > Dear Reviewer 4ZWg,
> > >
> > > We sincerely appreciate your effort in engaging with our submission and raising points that have allowed us to further clarify and strengthen our paper. Please let us know if there are any additional points we can address to further clarify or improve our submission.
> > >
> > > Thank you again for your time and consideration.
> > >
> > > Best regards,
> > > Authors of submission 7172

---

> ### Author Response · Authors · 2024-11-24
> **Thanks for the time you dedicated to carefully reviewing this paper**
>
> Dear reviewer 4ZWg,
>
> Thanks for the time you dedicated to carefully reviewing this paper. It would be highly appreciated if you let us know whether our responses properly address your concerns, despite your busy schedule. Thanks a lot!
>
> Best regards,
>
> Authors of submission 7172

---

### Official Review · Reviewer_9RCk · 2024-11-04

**Soundness:** 3
**Presentation:** 2
**Contribution:** 3
**Rating:** 6
**Confidence:** 3

**Summary:**

The authors proposed an estimation framework named NCDL to identify the latent causal variables, the structures among them, and the observed causal DAG, assuming that the temporal structure of the data follows a 3-Measurements Model for the climate system. They establish the conditions required for the identification of latent variables, enforcing sparsity on the latent Markov network.

**Strengths:**

1. The proposed framework aims to address a realistic problem in the climate system using a novel setting, the 3-Measurements Model.

2. The framework of the paper is straightforward, and the paper is well organized.

3. There are theoretical guarantees for the identifiability of the latent Markov network and the observed DAG.

4. The proposed framework has been applied to a series of simulations and a case study.

**Weaknesses:**

1. It is unclear what the motivation is for using the 3-Measurements Model instead of an $n$-Measurements Model with $n \neq 3$. Could you provide a brief explanation of why the 3-Measurements Model was chosen over other options, and how it specifically relates to climate system analysis.

2. The motivation and limitations of the proposed setting are unclear. There seems to be no time-lag effect among $x_{t-1}, x_t, x_{t+1}$. Could you clarify the implications of not including time-lag effects between the observed variables, and how this might impact the model's applicability to real-world climate systems.

3. In Equation 2, $pa_{x_t}(x_{t,i})$ only includes parents $x_{t,j}$, where there is no time lag in $t$. Does this mean that the parent of $x_{t,i}$ is restricted to instantaneous cases in the observed space? If so, could you elaborate on the details in comparison with constraint-based methods, which allow for multiple time lags for observed variables but do not yield time-lagged causal structure among latent variables?

**Questions:**

1. Could you explain the statement in line 122 that $x_{t-1},x_{t}, x_{t+1}$ are different measurements of $z_t$?

2. In Figure 3, there are two $Z_{t-1}$; is this a typo?

3. Where is the starting point of the framework visualized in Figure 6? Could you briefly summarize the framework based on Figure 6?

4. In the experiment, is there a specific reason for choosing PCMCIZ as a baseline instead of LPCMCI, which allows for latent variables?

5. Could you explain the definitions of $pa_{z_{t-1}}(z_{t,i})$ and $pa_{z_t}(z_{t,i})$?

6. Could you briefly explain what the inputs are and what estimations/DAGs are obtained from the model?

7. What does the estimated DAGs look like in terms of the dimension of the observed variables? Are the estimated DAGs a full-time causal graph or a summary causal graph?

8. In Table 3, is there a reason why the metric for the Independent and Sparse cases is MCC while the metric for the Dense case is $R^2$? Can all three cases be evaluated using both metrics?

9. Could you also use F1, recall, and precision as metrics in the comparison shown in Figure 7?

10. Are you assuming that there is no causal effect from future variables to past variables?

---

> ### Author Response · Authors · 2024-11-22
> **Response to Reviewer 9RCk Part 1**
>
> Thank you for your insightful and constructive feedback, which indeed improves the readability of our theories and models as well as the soundness of our experiments. We provide the point-to-point response to your comments below and have updated the paper accordingly.
>
> With gratitude, we have included a new section discussing how to extend our framework to **allowing time-lagged causal relations** in the observed space, inspired from the latent variable identification in 4-measurement model. We have also provided a **new explanation** of the 3-measurement model, a **new figure** with captions illustrating the model architecture, and a **new set of experiments for LPCMCI** along with detailed experimental settings. The manuscript has been thoroughly revised in light of your constructive suggestions. Please find our point-by-point responses below.
>
> >**W1:** It is unclear what the motivation is for using the 3-Measurement Model instead of an $n$-Measurements Model with $n \neq 3$. Could you provide a brief explanation of why the 3-Measurement Model was chosen over other options, and how it specifically relates to climate system analysis.
>
>
> **(1) A1:** We sincerely appreciate this essential point. The reason why we choose the 3-measurement model is that in the climate time-series data, each observation in a time-index can be considered as a measurement of $\mathbf{z}_t$, and thus we have access to at least 3 measurements. On the theoretical side, the 3-measurement model specifies the minimal information required for the recovery of latent variables, supported by [1]. Moreover, having more than 3 measurements could generally improve the estimation results as additional measurements also contain information about latent variables.
>
> Generally speaking, if $n < 3$, the model is not identifiable. (In certain scenarios, if side information such as sparsity [2] or domain changes [3] is provided, it could be identifiable) Please refer to [1] for why 3 measurements are needed in the fundamental nonparametric case. Moreover, even in the simple linear-Gaussian case, it has been shown that having access to only 2 measurements for 1 latent variable is not sufficient for identifiability of the linear model, see, e.g., [4].
>
> Following your insightful suggestion, we have included this explanation in the revised version (Page 3, Line 141).
>
> >**W2:** The motivation and limitations of the proposed setting are unclear. There seems to be no time-lag effect among $x_{t-1}, x_{t}, x_{t+1}$. Could you clarify the implications of not including time-lag effects between the observed variables, and how this might impact the model's applicability to real-world climate systems.
>
> **(2) A2:** Thank you for bringing up this practical question. In the model developed in this paper, we didn’t consider time-lag effects, mainly because the time-resolution of CESM2 data is **1 month**, and the temperature interactions (causal DAG), for instance, caused by the wind, occurs over a relatively short timescale. For example, it has been reported that such changes may happen during a single night [5], unlike the continuous and long-term processes of high-level latent variables (e.g., oceanic circulation patterns or gradual atmospheric pressure changes). Empirically, we also found time-delayed dependence (autocorrelation) in the CESM2 data is relatively small. Specifically, these points are summarized in the updated manuscript (Page 32, Line 1693).
>
> On the other hand, we would like to add that if time-lagged effects among observations are present, as suggested in your question, our framework can be immediately extended to handle it. To address this, we have added a new section in Appendix G (Page 34) of the updated manuscript, providing discussions on this topic. The primary difference compared with 3-measurement model is the so-called **limited feedback**:
>
> $p(\mathbf{x} _ t \mid \mathbf{x} _ {t-1}, \mathbf{z} _ t, \mathbf{z} _ {t-1}) = p(\mathbf{x} _ t \mid \mathbf{x} _ {t-1}, \mathbf{z}_t)$,
>
> which can be considered as the **4-measurement model** [6] in which the identifiability of latent variables $\hat{\mathbf{z}} _ t = h _ z(\mathbf{z} _ t)$ can hold true. Additionally, we demonstrate that the **functional equivalence** property holds for the time-lagged observed causal DAG, by considering $[\mathbf{x} _ {t-1}, \mathbf{x} _ {t}]$ as a whole. Ultimately, the latent variables, and the observed causal DAGs are still identifiable. Moreover, we provide the **improved implementations** tailored for scenarios involving time-lagged effects in observed space.

---

> ### Author Response · Authors · 2024-11-22
> **Response to Reviewer 9RCk Part 2**
>
> > **W3:** In Equation 2, $pa _ {x _ t}(x _ {t,i})$ only includes parents $x _ {t,j}$, where there is no time lag in $t$. Does this mean that the parent of $x_{t,i}$ is restricted to instantaneous cases in the observed space? If so, could you elaborate on the details in comparison with constraint-based methods, which allow for multiple time lags for observed variables but do not yield time-lagged causal structure among latent variables?
>
> **(3) A3:** Thank you for raising this point. You are right--in the original Eq. 2, we used $\mathbf{pa} _ {x _ {t,i}}$ to represent observed parents, and we didn’t explicitly mention latent parents. To make it clearer, we have used $\mathbf{pa} _ L$ to represent the set of latent parents and $\mathbf{pa} _ O$ to denote the set of observed parents.  We have updated the equation in the revised manuscript (Page 3, Line 149) to reflect this:
>
> $x _ {t,i} = g\left( \{ z _ {t,j} \mid z _ {t,j} \in \mathbf{pa} _ {L}(x _ {t,i}) \} \right) + \sum _ {x _ {t,j} \in \mathbf{pa} _ {O}(x _ {t,i})} b _ {i,j}(\mathbf{z} _ t, s _ {t,i}) \cdot x _ {t,j} + s _ {t,i}$,
>
> >**Q1:** Could you explain the statement in line 122 that $\mathbf{x} _ {t-1}, \mathbf{x} _ {t}, \mathbf{x} _ {t+1}$ are different measurements of $\mathbf{z} _ {t}$?
>
> **(4) A1:** Thank you for the question. In the basic 3-measurement model [1], the essential condition is that the 3 measurements are conditionally independent given the latent variable; in our model, according to Definition 2.1, $\mathbf{x} _ {t-1}$, $\mathbf{x} _ {t}$, and $\mathbf{x} _ {t+1}$ are defined to be conditionally independent given $\mathbf{z} _ {t}$. Thus they are considered as different measurements of $\mathbf{z} _ {t}$, in the language of the measurement model. Similarly, [6] employs a similar concept for measurements in time-series data (Theorem 1). We have included this explanation and highlighted it in the updated manuscript (Page 3, Line 130).
>
> >**Q2**: In Figure 3, there are two $\mathbf{z} _ {t-1}$; is this a typo?
>
> **(5) A2:** Thank you for your careful review. Yes, it is a typo, and we have corrected it and highlighted the change in the revised version (Page 4, line 162).
>
> > **Q3** Where is the starting point of the framework visualized in Figure 6? Could you briefly summarize the framework based on Figure 6?
>
> **(6) A3:** Thank you for the feedback that has helped improve Figure 6. In light of your questions, we have provided a revised Figure 6 including additional details, and provided a clearer depiction of the pipeline (Page 8, Line 378). We also have revised the introduction of the model and the explanation of Figure 6 (Page 8, Line 396) as below:
>
> The starting point of the framework visualized in Figure 6 is the observed time-series $[\mathbf{x} _ 1, \mathbf{x} _ 2, \dots, \mathbf{x} _ T]$. The process begins with computing posterior of the latent variables $\hat{\mathbf{z}} _ {t-1}, \hat{\mathbf{z}} _ t$ from the observations $\mathbf{x} _ {t-1}, \mathbf{x} _ t$ using the **z-encoder**. Then, using normalizing flows, the prior of $\hat{\mathbf{z}} _ t$ is computed from the obtained $\hat{\mathbf{z}} _ {t-1}$ by the **z-prior** network. The latent structures are subsequently derived from the Jacobian matrices of this prior network. Simultaneously, $\mathbf{s} _ t$ is computed using the **s-encoder** from observations, and its **s-prior** is determined by normalizing flows based on the obtained $\hat{\mathbf{z}} _ t$. The observations $\hat{\mathbf{x}} _ t$ are then reconstructed from $(\hat{\mathbf{s}} _ t, \hat{\mathbf{z}} _ t)$ through a **decoder**, where $\mathbf{J} _ {\hat{g} _ m}(\hat{\mathbf{s}} _ t)$ is computed from the decoder. MSE loss ensures $\mathbf{x} _ t = \hat{\mathbf{x}} _ t$. This process ultimately obtains the observed causal DAG $\mathbf{J} _ {\hat{g}}(\hat{\mathbf{x}} _ t)$ through corollary 3.6.
>
> > **Q4:** In the experiment, is there a specific reason for choosing PCMCI as a baseline instead of LPCMCI, which allows for latent variables?
>
> **(7) A4:** Thank you very much for the suggestion, which enriches our baselines. There was no specific reason, and we have incorporated LPCMCI in our baselines and added a new set of experiments, together with all considered metrics (Page 10, Line 486). he corresponding implementation details are given in Line 1579, Page 30. The (new) experimental results with LPCMCI are summarized in the table below.
>
> | Metric \ Number of samples | $10^3$ | $10^{3.5}$ | $10^4$ | $10^{4.5}$ | $10^5$ |
> |---------------|-----------|------------|-----------|------------|-----------|
> | **SHD** | 0.5862 | 0.4865 | 0.3854 | 0.3314 | 0.3314 |
> | **Recall** | 0.4329 | 0.5426 | 0.5581 | 0.6137 | 0.6124 |
> | **Precision** | 0.5150 | 0.5645 | 0.5923 | 0.6294 | 0.6334 |
> | **F1** | 0.4699 | 0.5530 | 0.5704 | 0.6165 | 0.6228 |
>
> Specifically, our NCDL significantly outperforms LPCMCI as well. We greatly appreciate your contribution and would like to acknowledge it in the final paper.

---

> ### Author Response · Authors · 2024-11-22
> **Response to Reviewer 9RCk Part 3**
>
> >**Q5**: Could you explain the definitions of and $pa _ {z _ t}(z _ {t,i})$ and $pa _ {z _ {t-1}}(z _ {t,i})$?
>
> **(8) A5:** Thanks for this point regarding the clarity of the paper. $pa _ {z _ t}(z _ {t,i})$ represents the set of parents of $z _ {t,i}$ within $\mathbf{z} _ {t}$, capturing instantaneous effects, while $pa _ {z _ {t-1}}(z _ {t,i})$ represents the set of parents of $z _ {t,i}$ within $\mathbf{z} _ {t-1}$, capturing time-lagged effects.
>
> To clarify them, we have explicitly unified these definitions as $\mathbf{pa} _ L$ as the set of latent parents in our revised version (Page 3, Line 149).
>
> > **Q6**: Could you briefly explain what the inputs are and what estimations/DAGs are obtained from the model?
>
> **(9) A6:** Thank you for this thoughtful question. The inputs to the model are an observed time series $[\hat{\mathbf{x}} _ 1, \hat{\mathbf{x}} _ 2, \dots, \hat{\mathbf{x}} _ T]$. The model estimates a series of latent variables $[\hat{\mathbf{z}} _ 1, \hat{\mathbf{z}} _ 2, \dots, \hat{\mathbf{z}} _ T]$, a series of noises $[\hat{\mathbf{s}} _ 1, \hat{\mathbf{s}} _ 2, \dots, \hat{\mathbf{s}} _ T]$, and the observed causal DAGs (full DAG over the observed time series) based on the Jacobian matrices of the decoder. Additionally, we obtain the latent causal DAGs and latent time-lagged structures from the Jacobian matrices from the z-prior network. We have updated the manuscript to provide such details (Page 8, Line 396; Page 8, Line 428).
>
> > **Q7:** What does the estimated DAGs look like in terms of the dimension of the observed variables? Are the estimated DAGs a full-time causal graph or a summary causal graph?
>
> **(10) A7:** Thanks for the question. The estimated DAGs consist of DAGs among the observed variables $\mathbf{x} _ t$ at different time indices, as we allow the DAG structure to change over time. That is, at each time, it is a DAG with $d _ x$ nodes (variables), where $d _ x$ is the dimensionality of the observed variables.
> In light of your suggestion, we have provided this description in the updated manuscript (Page 9, Line 434).
>
> > **Q8:** In Table 3, is there a reason why the metric for the Independent and Sparse cases is MCC while the metric for the Dense case is $R^2$? Can all three cases be evaluated using both metrics?
>
> **(11) A8:** Thank you for your questions. Here are the reasons why we chose them.
>
> - For the **Independent** and **Sparse** settings, Theorem 3.2 ensures that latent variables can be recovered up to a component-wise transformation. In these cases, Spearman MCC is used as a metric because it evaluates the rank-based correlation between predicted and true latent variables, as used in [7] for the same task.
> - For the **Dense** setting, however, component-wise recovery is not achievable. Instead, we use $R^2$ to evaluate monoblock identifiability, as described in Theorem 3.3. Specifically, $R^2$ measures how well the predicted latent variables can explain the true latent variables. This makes $R^2$ appropriate for dense settings where the goal is to figure out whether  $\hat{\mathbf{z}}_t = g(\mathbf{z}_t)$, where $g$ is an invertible function.
>
> In response to your second question (three cases using both metrics), we have conducted a new set of simulated experiments incorporating all the considered evaluations:
> | Setting | Metric | NCDL | iCITRIS | GCaRL | CaRiNG | TDRL | LEAP | SlowVAE | PCL | i-VAE | TCL |
> |--------------|--------|--------|---------|--------|--------|--------|--------|---------|--------|--------|--------|
> | **Independent** | MCC | **0.9811** | 0.6649 | 0.8023 | 0.8543 | 0.9106 | 0.8942 | 0.4312 | 0.6507 | 0.6738 | 0.5916 |
> | | $R^2$ | **0.9626** | 0.7341 | 0.9012 | 0.8355 | 0.8649 | 0.7795 | 0.4270 | 0.4528 | 0.5917 | 0.3516 |
> | **Sparse** | MCC | **0.9306** | 0.4531 | 0.7701 | 0.4924 | 0.6628 | 0.6453 | 0.3675 | 0.5275 | 0.4561 | 0.2629 |
> | | $R^2$ | **0.9102** | 0.6326 | 0.5443 | 0.2897 | 0.6953 | 0.4637 | 0.2781 | 0.1852 | 0.2119 | 0.3028 |
> | **Dense** | MCC | **0.6750** | 0.3274 | 0.6714 | 0.4893 | 0.3547 | 0.5842 | 0.1196 | 0.3865 | 0.2647 | 0.1324 |
> | | $R^2$ | **0.9204** | 0.6875 | 0.8032 | 0.4925 | 0.7809 | 0.7723 | 0.5485 | 0.6302 | 0.1525 | 0.206 |
>
> **Bold numbers** indicate the best performance.
>
> This table illustrates that, under different settings, NCDL outperforms all other considered methods of temporal representation learning. This have been added to the updated manuscript (Page 10, Line 505).

---

> ### Author Response · Authors · 2024-11-22
> **Response to Reviewer 9RCk Part 4**
>
> > **Q9:** Could you also use F1, recall, and precision as metrics in the comparison shown in Figure 7?
>
> **(12) A9:** Thank you for your observation and insightful suggestion. We have included a **new figure** in the updated version (Page 10, Line 486) that incorporates F1, recall, and precision metrics for comparison among constraint-based methods. We also provided a table (which is a figure in the updated manuscript) as below:
>
> | Dataset Size (x) | Method        | SHD    | Recall | Precision | F1     |
> |-------------------|---------------|--------|--------|-----------|--------|
> | $10^3$| NCDL          | **0.38876** | **0.55**   | **0.61**      | **0.5784** |
> |                   | FCI           | 0.47876 | 0.3021 | 0.4321    | 0.3556 |
> |                   | CD-NOD        | 0.5256  | 0.348  | 0.381     | 0.3638 |
> |                   | PCMCI         | 0.594   | 0.37   | 0.38      | 0.3749 |
> |                   | LPCMCI        | 0.5862  | 0.432  | 0.515     | 0.4699 |
> | $10^{3.5}$| NCDL          | **0.235**   | **0.70**   | **0.68**      | **0.6899** |
> |                   | FCI           | 0.4088  | 0.35   | 0.4631    | 0.3987 |
> |                   | CD-NOD        | 0.4687  | 0.38   | 0.401     | 0.3902 |
> |                   | PCMCI         | 0.565   | 0.3642 | 0.3942    | 0.3786 |
> |                   | LPCMCI        | 0.4865  | 0.542  | 0.5645    | 0.5530 |
> | $10^4$| NCDL          | **0.22**    | **0.78**   | **0.7543**    | **0.7669** |
> |                   | FCI           | 0.3798  | 0.3631 | 0.4621    | 0.4067 |
> |                   | CD-NOD        | 0.4567  | 0.415  | 0.422     | 0.4185 |
> |                   | PCMCI         | 0.4654  | 0.38   | 0.4521    | 0.4129 |
> |                   | LPCMCI        | 0.3854  | 0.55   | 0.5923    | 0.5704 |
> | $10^{4.5}$| NCDL          | **0.18**    | **0.8312** | **0.7821**    | **0.8059** |
> |                   | FCI           | 0.3835  | 0.3832 | 0.4832    | 0.4274 |
> |                   | CD-NOD        | 0.4542  | 0.4244 | 0.424     | 0.4242 |
> |                   | PCMCI         | 0.4243  | 0.3708 | 0.4908    | 0.4224 |
> |                   | LPCMCI        | 0.3314  | 0.613  | 0.62      | 0.6165 |
> | $10^5$| NCDL          | **0.18**    | **0.8312** | **0.8012**    | **0.8159** |
> |                   | FCI           | 0.3722  | 0.40   | 0.49      | 0.4404 |
> |                   | CD-NOD        | 0.4542  | 0.43   | 0.439     | 0.4345 |
> |                   | PCMCI         | 0.4224  | 0.42   | 0.512     | 0.4615 |
> |                   | LPCMCI        | 0.3314  | 0.6124 | 0.6334    | 0.6228 |
>
> The results suggest that NCDL achieves the lowest SHD and highest recall, precision and F1, outperforms other constraint-based methods across all metrics.
>
> > **Q10:** Are you assuming that there is no causal effect from future variables to past variables?
>
> **(13) A10:** Thanks for your question. Yes, in this work, we make the assumption that the future does not cause the past, consistent with prior studies on causal learning in climate science [8,9]. We have emphasized this assumption in the updated manuscript (Page 3, Line 152).
>
>
>
> **References:**
>
> [1] Hu, Yingyao, and Susanne M. Schennach. "Instrumental variable treatment of nonclassical measurement error models." *Econometrica* 76.1 (2008): 195–216.
>
> [2] Zheng, Yujia, Ignavier Ng, and Kun Zhang. "On the identifiability of nonlinear ICA: Sparsity and beyond." *Advances in Neural Information Processing Systems*, 35 (2022): 16411–16422.
>
> [3] Khemakhem, Ilyes, et al. "Variational autoencoders and nonlinear ICA: A unifying framework." In *International Conference on Artificial Intelligence and Statistics*, PMLR, 2020.
>
> [4] Bekker, Paul A., and Jos MF ten Berge. "Generic global indentification in factor analysis." _Linear Algebra and its Applications_ 264 (1997): 255-263.
>
> [5] Zhou, Liming, et al. "Impacts of wind farms on land surface temperature." *Nature Climate Change* 2.7 (2012): 539–543.
>
> [6] Yingyao Hu and Matthew Shum. "Nonparametric identification of dynamic models with unobserved state variables." *Journal of Econometrics*, 171(1):32–44, 2012.
>
> [7] Yao, Weiran, et al. "Temporally disentangled representation learning." *Advances in Neural Information Processing Systems*, 35 (2022): 26492–26503.
>
> [8] Jakob Runge. "Discovering contemporaneous and lagged causal relations in autocorrelated nonlinear time series datasets." In *Conference on Uncertainty in Artificial Intelligence*, pp. 1388–1397. PMLR, 2020.
>
> [9] Jakob Runge, et al. "Detecting and quantifying causal associations in large nonlinear time series datasets." *Science Advances*, 5(11):eaau4996, 2019.

---

> ### Author Response · Authors · 2024-11-24
> **We sincerely appreciate your taking the time to review our manuscript and providing us with your insightful questions and suggestions**
>
> Dear reviewer 9RCk,
>
> We sincerely appreciate your taking the time to review our manuscript and providing us with your insightful questions and suggestions. As the rebuttal discussion period is limited, we eagerly await any additional feedback you may have. We are more than willing to have further discussions with you.
>
> Best regards,
>
> Authors of submission 7172

---

> ### Author Response · Authors · 2024-11-30
> **Concerns properly addressed?**
>
> Dear Reviewer 9RCk,
>
> Thanks for the time and effort you dedicated to reviewing our submission. We hope your concerns have been properly addressed, and if you have further comments, we hope for the opportunity to respond to them. Your feedback would be appreciated.
>
> Best wishes,
>
> Authors of submission 7172

---

> > ### Comment · Reviewer_9RCk · 2024-12-02
> >
> > Apologies for the delayed response, and thank you to the authors for providing detailed answers and experimental results. My concerns have been addressed, and I am raising my score to 6.

---

> > > ### Author Response · Authors · 2024-12-02
> > > **Thank you for raising your score.**
> > >
> > > Dear Reviewer 9RCk,
> > >
> > > Thank you for raising your score. We greatly appreciate your valuable feedback, which has significantly contributed to enhancing the clarity and quality of our work.
> > >
> > > Best regards,
> > >
> > > Authors of submission 7172

---

### Meta-Review · Area_Chair_q6DR · 2024-12-11

**Metareview:**

This paper introduces a nonparametric dynamic causal modeling framework aimed at identifying causal structures and latent processes in climate systems. While the topic is significant, the paper suffers from substantial limitations that undermine its overall contribution. The theoretical framework relies on strong assumptions that are not sufficiently justified for real-world climate data. Additionally, the experimental results, while demonstrating some promise, are limited in scope and fail to compare against more advanced or tailored baselines. Critical elements, such as scalability and the impact of time-lagged variables, are not adequately explored, limiting the framework’s practical applicability. Presentation issues, including unclear definitions and overly complex notations, further hinder accessibility. Despite some theoretical innovations, the work does not provide enough empirical validation or practical insights to support its claims effectively. The combination of these issues justifies a decision to Reject.

**Additional Comments On Reviewer Discussion:**

During the rebuttal period, reviewers raised concerns about the justification of the strong assumptions underlying the proposed framework, including the choice of the 3-Measurement Model and its applicability to real-world climate systems. The authors clarified the theoretical basis for their model and added explanations about the measurement model’s role in identifiability. However, the responses did not fully address concerns about the practical limitations of these assumptions. Reviewers also questioned the lack of exploration of time-lagged causal effects and scalability to high-dimensional datasets. The authors included a discussion on extending the framework to handle time-lagged effects and acknowledged scalability challenges, suggesting potential solutions for future work. While these additions improved clarity, they did not resolve the issue of insufficient empirical validation. Finally, reviewers noted gaps in experimental comparisons, particularly the absence of key baselines and detailed runtime analyses. The authors incorporated additional baselines and metrics, but the results failed to convincingly demonstrate the superiority of the method. Overall, while the authors’ efforts during the rebuttal period improved the clarity and scope of the paper, significant limitations in empirical rigor and practical applicability remain. These unresolved issues contributed to the final decision to Reject.

---

### Decision · Program_Chairs · 2025-01-22

Reject